# THE OTHER SIDE OF THE COIN: UNVEILING THE DOWNSIDES OF MODEL AGGREGATION IN FEDERATED LEARNING FROM A LAYER-PEELED PERSPECTIVE

## ABSTRACT

In federated learning (FL), model aggregation plays a central role in enabling decentralized knowledge sharing. However, it is often observed that the aggregated model underperforms on local data until after several rounds of local training. This temporary performance drop can potentially slow down the convergence of the FL model. Prior work regards this performance drop as an inherent cost of knowledge sharing among clients and does not give it special attention. While some studies directly focus on designing techniques to alleviate the issue, its root causes remain poorly understood. To bridge this gap, we construct a framework that enables layer-peeled analysis of how feature representations evolve during model aggregation in FL. It focuses on two key aspects: (1) the intrinsic quality of extracted features, and (2) the alignment between features and their subsequent parameters—both of which are critical to downstream performance. Using this framework, we first investigate **what** model aggregation does to the internal feature extraction process. Our analysis reveals that aggregation degrades feature quality and weakens the coupling between intermediate features and subsequent layers, both of which are well shaped during local training. More importantly, this degradation is not confined to specific layers but progressively accumulates with network depth—a phenomenon we term Cumulative Feature Degradation (CFD). CFD severely impairs the quality of penultimate-layer features, ultimately compromising the model's decision-making capacity. Next, we examine **how** key FL settings—such as aggregation frequency—can exacerbate or alleviate the negative effects of model aggregation. Finally, we revisit several commonly used strategies, such as initialization from pretrained models, and explain **why** they are effective through layer-peeled analysis. To the best of our knowledge, this is the first systematic study of model aggregation in FL from a layer-peeled feature extraction perspective, potentially paving the way for the development of more effective FL algorithms. The code is available at: `https://anonymous.4open.science/r/ICLR_14921_Code-3565`.

## 1 INTRODUCTION

Recently, federated learning (FL) has gained increasing interests (Kairouz et al., 2021) since it can enable multiple clients to collaboratively train models without sharing their private data. A standard FL process involves iterative cycles in which local models are trained on each client, followed by aggregation of these locally updated models on a central server (McMahan et al., 2017), as presented in Figure 1. During local training, each client performs multiple rounds of model updates using its private data. Once local training is complete, the updated model is uploaded to the server. The server then aggregates the uploaded models via parameter-wise averaging, with each model weighted based on factors such as the number of training samples on each client (Acar et al., 2021; Li et al., 2020). The aggregated model is then sent back to each client for next round of local training.

In the above process, model aggregation is a key step that facilitates knowledge sharing among clients in FL. However, it is well known that the model aggregation often leads to a significant performance drop compared to the model before aggregation (Jin et al., 2022; Lee et al., 2022; Yao et al., 2024). This phenomenon is particularly pronounced when data distributions across clients are heterogeneous, a common scenario in practical applications due to factors such as variations in data acquisition conditions across different clients (Zhu et al., 2021; Li et al., 2021b). To further investigate this phenomenon, we conduct preliminary experiments in a typical data-heterogeneous FL setting. Figure 1 presents a performance comparison of the model before and after aggregation, evaluated on the local

data from each client. As shown, the performance of the aggregated model significantly deteriorates compared to the model before aggregation.

Although this temporary performance drop can be mitigated after several rounds of local updates, its impact persists and continues to pose a challenge. The suboptimal initialization in each local update, caused by model aggregation, undermines the progress made in the previous round, potentially slowing the convergence rate of FL training. Most FL research treats this performance drop as an inherent cost of knowledge sharing among clients, giving it limited atten-

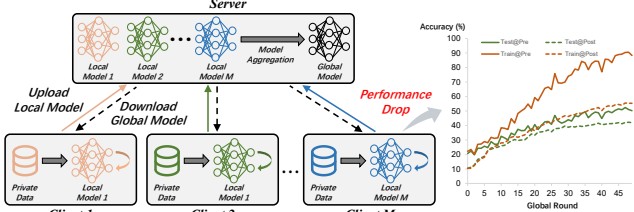

Figure 1: **Left:** Local model training and global model aggregation during FL training. **Right:** Performance comparison when evaluating models on local data, *@Pre refers to the evaluation results of the model before aggregation, while *@Post indicates the results after aggregation.

tion (McMahan et al., 2017; Wu et al., 2023; Li et al., 2021a). Recently, several studies attributes this phenomenon to 'client drift' (Li et al., 2020; Karimireddy et al., 2020) and 'knowledge forgetting' (Jin et al., 2022; Lee et al., 2022). To address this issue, methods such as partial parameter personalization (Arivazhagan et al., 2019; Li et al., 2021b; Sun et al., 2021), pre-trained initialization (Nguyen et al., 2023), and classifier fine-tuning (Oh et al., 2022; Li et al., 2023) are proposed.

While the above methods have achieved notable success in mitigating the performance degradation caused by model aggregation—mainly as measured by metrics such as accuracy or loss—the fundamental causes of this issue remain insufficiently understood. This gap can be largely attributed to the dominant use of deep neural networks (DNNs) in FL Li et al. (2021b); Chen et al. (2023); Li et al. (2023); Oh et al. (2022), which are often treated as black-box models Montavon et al. (2018); Samek et al. (2021) composed of stacked layers performing hierarchical feature extractionYosinski et al. (2014); Olah et al. (2017); Masarczyk et al. (2024); Wang et al. (2023b); Zeiler & Fergus (2014); Rangamani et al. (2023). Such architectures hinder the interpretation of internal feature representation dynamics that potentially underlie aggregation-induced degradation. Although some recent studies have attempted to analyze FL from a layer-peeled perspective (Luo et al., 2021; Chan et al., 2024; Adilova et al., 2024; Ma et al., 2022), they either focus on parameter-space analysis using loss-based metrics (Chan et al., 2024), or rely on feature similarity metrics that require two models for comparison Luo et al. (2021); Adilova et al. (2024). To the best of our knowledge, no existing work analyzes the dynamics of layer-peeled feature extraction using metrics that can be directly computed using a single model to quantify either feature quality or feature-parameter alignment.

To bridge this gap, we construct a framework to investigate model aggregation from a layer-peeled feature extraction perspective. We hypothesize that the model performance fundamentally depends on two key factors: (1) the intrinsic quality of the extracted features, and (2) the degree of alignment between features and the parameters of subsequent layers. The first factor determines whether the features are semantically meaningful and discriminative, while the second determines whether they can be effectively exploited by the model's decision-making or downstream feature extraction modules. Accordingly, our framework integrates quantitative metrics to evaluate both aspects, providing a more interpretable understanding of aggregation effects in FL.

Based on this framework, we perform a layer-peeled feature analysis of aggregation across multiple datasets and model architectures. In doing so, we aim to answer the following key questions:

- **How model aggregation affects feature representations?** We observe that model aggregation generally compromises the quality of extracted features and weakens the alignment between features and subsequent parameters. More importantly, this degradation is not confined to specific layers but accumulates progressively with network depth—a phenomenon we term Cumulative Feature Degradation (CFD). We identify CFD as a fundamental factor contributing to the performance drop when aggregating DNNs.

- **Why are common solutions effective?** We revisit several widely used solutions for mitigating the effects of model aggregation, including personalizing specific parameters, initializing models with pretrained parameters, and fine-tuning the classifier. Our analysis shows that these methods are effective because they can address the feature degradation issues describe above. This further validates the utility of our layer-peeled analysis framework.

We also conduct a theoretical analysis to support our empirical observations (see Appendix G). This study provide a comprehensive study of model aggregation in FL from a layer-peeled feature extraction perspective. The key findings can inspire the design of more effective FL algorithms.

## 2 PROBLEM FORMULATION OF FL FROM A LAYER-PEELED PERSPECTIVE

In this paper, we consider a standard FL system consisting of a central server and $M$ distributed clients. We assume that each client $m$ contains $N_m$ training samples, which are drawn from the data distribution $\mathcal{D}_m$. In practice, the underlying data distribution $\mathcal{D}_m$ for each client $m$ is typically different from one another due to the variations in data collection conditions. Formally, the training samples on client $m$ can be represented as $(\boldsymbol{x}_m^i, \boldsymbol{y}_m^i)_{i=1}^{N_m}$, where $\boldsymbol{x}_m^i \in \mathcal{X}_m \subseteq \mathbb{R}^n$ denotes the raw input data for the DNNs, and $\boldsymbol{y}_m^i \in \mathcal{Y}_m \subseteq \{0,1\}^C$ represents the corresponding ground truth labels used to optimize the DNNs, with $C$ denoting the number of classes.

We denote the DNN for client $m$ as $\boldsymbol{\psi}_m(\cdot)$, with parameters represented by $\boldsymbol{\Theta}_m$. The optimization objective for an FL system can then be formulated as:

$$\arg\min_{\boldsymbol{\Theta}_1,\ldots,\boldsymbol{\Theta}_M} \mathcal{L}(\boldsymbol{\Theta}_1,\ldots,\boldsymbol{\Theta}_M) \triangleq \arg\min_{\boldsymbol{\Theta}_1,\ldots,\boldsymbol{\Theta}_M} \frac{1}{M} \sum_{m=1}^{M} \mathcal{L}_m(\boldsymbol{\Theta}_m), \tag{1}$$

where $\mathcal{L}_m(\boldsymbol{\Theta}_m)$ represents the empirical risk for client $m$.

To optimize Equation (1) in a privacy-preserving manner, FL is typically carried out in two iterative stages: local model training and global model aggregation. During the local model training phase, each client optimizes its model for $E$ epochs using its private data. Once local training is complete, each client uploads its updated model to the server. The server then performs model aggregation to generate the global model. A common aggregation strategy involves applying a weighted average based on the number of training samples per client, which is expressed as $\tilde{\boldsymbol{\Theta}} = \sum_{m=1}^{M} \frac{N_m}{\sum_{m=1}^{M} N_m} \boldsymbol{\Theta}_m$.

Here, $\tilde{\boldsymbol{\Theta}}$ represents the aggregated global model. By repeating the above procedures for several rounds, the model eventually converges, resulting in the final FL model. For simplicity, we will sometimes refer to the locally updated model $\boldsymbol{\Theta}_m$ as the ***pre-aggregated model***, and $\tilde{\boldsymbol{\Theta}}$ as the ***post-aggregated model***. Additionally, we may omit the client and sample indices for simplicity.

It is well known that post-aggregated models often suffer a significant performance drop when evaluated on local data. We provide detailed evidence of this phenomenon in Appendix H. To better understand its underlying causes, we propose a layer-peeled feature analysis framework to investigate how feature extraction evolves across model depth during aggregation. Within this framework, we reformulate the parameters of the FL model as $\boldsymbol{\Theta} = \left\{\boldsymbol{W}^\ell\right\}_{\ell=1}^{L}$, where $L$ denotes the total number of layers, and the depth increases with $\ell$. This stacked layers progressively transforms the input data into prediction outputs, from the shallow layers to the deeper layers. The features of the $\ell$-th layer for input $\boldsymbol{x}$ can be formulated as:

$$\boldsymbol{z}^\ell = \boldsymbol{W}^\ell \ldots \boldsymbol{W}^1 \boldsymbol{x} = \boldsymbol{W}^{\ell:1} \boldsymbol{x}, \forall \ell = 1,\ldots,L-1 \tag{2}$$

where $\boldsymbol{z}^\ell$ denotes the intermidiated features of $\ell$-th layer and we define $\boldsymbol{z}^0 = \boldsymbol{x}$.

## 3 EVALUATION SETUP

### 3.1 IMPLEMENTATION DETAILS

**Datasets.** We conduct experiments on cross-domain datasets, including Digit-Five, PACS (Li et al., 2017), and DomainNet (Peng et al., 2019). For each dataset, samples from a single domain are assigned to one client. Details on dataset preprocessing and partitioning are provided in Appendix C.

**Model Architectures.** Our evaluation are conducted on various architectures, including convolutional networks (ConvNet), three variants of ResNet (ResNet18, ResNet34, and ResNet50) (He et al., 2016), VGG13_BN (Simonyan & Zisserman, 2014) , and ViT_B-16 (Dosovitskiy et al., 2021). Detailed descriptions of architectures are provided in Appendix D.

**Training Settings.** We adopt the standard FedAvg algorithm (McMahan et al., 2017) for federated training. Model optimization is performed using stochastic gradient descent (SGD) with a learning rate of 0.01, momentum of 0.5, and a batch size of 64. Each client trains locally for 10 epochs

per global round, and the training runs for a total of 50 global rounds. On the server side, model aggregation is conducted via weighted averaging of parameters, with weights proportional to the number of training samples on each client. To account for randomness, all experiments are repeated three times with different random seeds. Further implementation details are provided in Appendix E.

**Feature Evaluation Metrics.** Our feature evaluation framework incorporates a set of metrics designed to evaluate both the quality of extracted features (feature vairance (Wang et al., 2023b; Rangamani et al., 2023) and linear probing accuracy (Chen et al., 2020; He et al., 2022; Wang et al., 2023a)) and their alignment (Rangamani et al., 2023; Jordan, 1875; Björck & Golub, 1973) with subsequent model parameters at a given layer $\ell$. To quantify changes in feature extraction before and after aggregation, we additionally introduce the pairwise distance between features or models and the relative change of evaluated metrics. The detailed definitions can be found in Appendix F.

# 4    HOW MODEL AGGREGATION AFFECTS FEATURE REPRESENTATIONS?

## 4.1    MODEL AGGREGATION DISRUPTS FEATURE VARIANCE SUPPRESSION

**Motivation.** In this section, we investigate how feature discrimination evolves during model aggregation, as it plays a crucial role in determining model performance on local client data. To this end, we measure both within-class and between-class variances to quantify the degree of intra-class feature compactness and inter-class featre separability.

**Experimental Results.** Figures 3 and 4 presents the evaluation resutls on DomainNet using ResNet50 as the backbone. These figures reveal several key observations:

**(1) Features become increasingly compressed within the same class as training progresses and layer depth increases.** As shown in Figure 3 (a), from a temporal perspective, the within-class feature variance at a given layer progressively decreases as federated training proceeds. Moreover, as depicted in Figure 4 (a), from a spatial perspective, at a given training epoch, the within-class feature variance decreases with increasing network depth. These observations indicate that the within-class feature variance decreases consistently as training progresses and the model layer goes deeper.

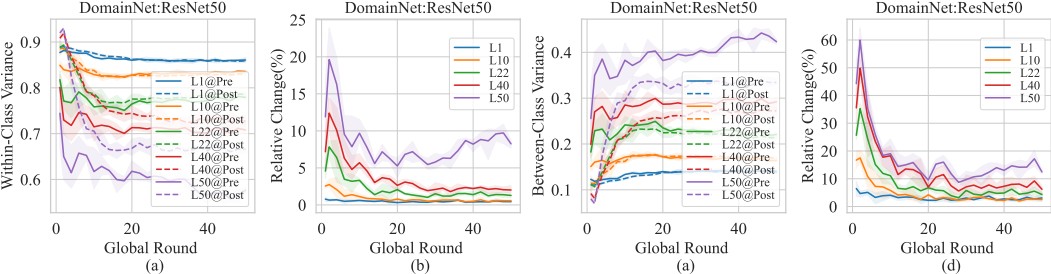

Figure 3: Normalized feature variances at different layers during FL training. The model is trained on DomainNet using ResNet50: (a) normalized within-class variance, (b) relative change of within-class variance, (c) normalized between-class variance, (d) relative change of between-class variance.

**(2) Features become increasingly discriminative across different class as training progresses and layer depth increases.** As shown in Figure 3 (a), from a temporal perspective, the normalized between-class feature variance at a given layer increases progressively over global rounds. Similarly, as shown in Figure 4 (a), from a spatial perspective, at a given global round, deeper layers exhibit higher between-class variance. These observations collectively indicate that the model progressively enhances feature separability across classes over both training time and network depth. Furthermore, prior work in centralized learning (CL) show that features become increasingly compact within classes and more separable across classes as training progresses and depth increases (Wang et al., 2023b; Rangamani et al., 2023). Together with Observation (1), our findings reveal that FL exhibits similar training dynamics from a layer-peeled feature extraction perspective, despite its decentralized nature.

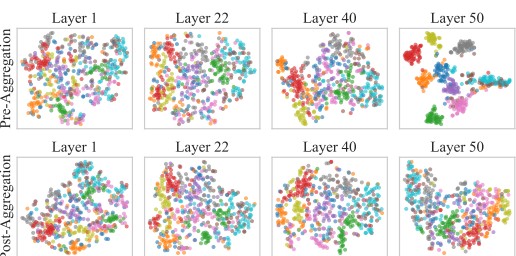

Figure 2: T-SNE Van der Maaten & Hinton (2008) visualization of features at different layers on the 'Quickdraw' domain of DomainNet. The features are extracted from ResNet50 in last round.

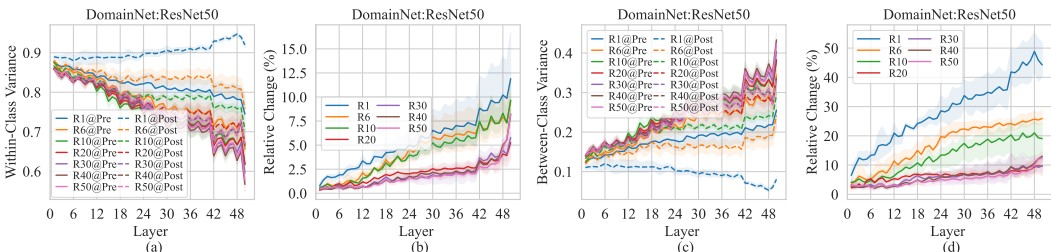

Figure 4: Normalized feature variances across layers at specific rounds. The model is trained on DomainNet using ResNet50: (a) normalized within-class variance, (b) relative change of within-class variance, (c) normalized between-class variance, (d) relative change of between-class variance.

**(3) Model aggregation disrupts feature variance suppression during FL training.** As shown in Figures 3 (a) and 4 (a), both temporally and spatially, model aggregation leads to increased within-class variance and decreased between-class variance. This opposes the training tendency of DNNs—namely, to reduce within-class variance and increase between-class variance—which has been empirically ver-

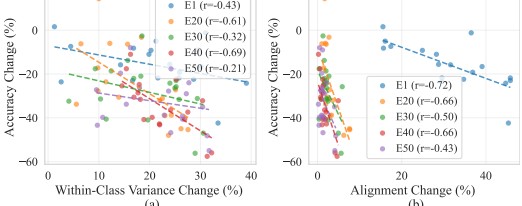

Figure 5: Correlations between feature degradation and performance drop during model aggregation.

ified in Observations 1 and 2. Further supporting evidence is provided in Figure 2, where the visualizations reveal degraded feature discrimination after aggregation across layers. Additionally, we compute the correlations between changes in variance and accuracy, as shown in Figure 5 (a), which demonstrates a strong relationship between variance degradation and performance drop. These findings suggest that, from the perspective of feature variance on local data, model aggregation inherently contradicts the training objective and can hinder the convergence rate of FL.

> **Takeaway**
>
> While FL shares similar training dynamics with CL—promoting within-class feature compactness and between-class feature separability—model aggregation can counteract this progression and ultimately impede FL convergence.

## 4.2 FEATURE VAIRANCE DISRUPTION ACCUMULATE AS MODEL DEPTH INCREASES

**Motivation.** In this section, we investigate the relative change in feature variance induced by model aggregation during training. The goal is to assess the sensitivity of different layers to aggregation and to reveal how the stacked architecture of DNNs affects feature variance in this process.

**Experimental Results.** The relative changes in within-class and between-class variance are shown in the Figures 3 and 4. From these figures, we make the following observations:

**(1) Feature variance degradation accumulates with increasing model depth.** The relative changes in both normalized within-class

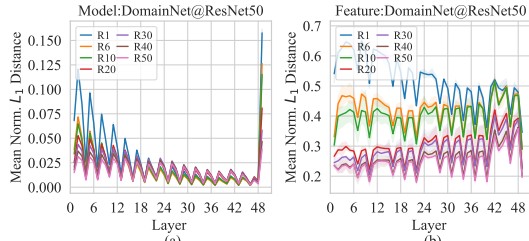

Figure 6: Mean normalized $L_1$ distance of the features and parameters between pre-aggregated and post-aggregated models across model layers for specific global rounds. The model is trained on DomainNet using ResNet50: (a) distance of model parameters (b) distance of features.

and between-class variance increase with network depth, indicating that feature variance degradation becomes more severe in deeper layers. We refer to this phenomenon as Cumulative Feature Degradation (CFD). Since this degradation is specifically defined in terms of feature variance, we denote it as CFD-V to distinguish it from CFD observed on other metrics. CFD-V can be attributed to the stacked architecture of DNNs, where degraded features in early layers propagate to subsequent layers, compounding the disruption. To further validate this, we compute layer-wise parameter and feature distances between pre-aggregated and post-aggregated models. As shown in Figure 6, the

magnitude of parameter distance is consistently smaller than that of feature distance. Moreover, parameter distance decreases with depth—except for the final classifier—while feature distance increases steadily across layers. Moreover, to demonstrate CFD, we conduct experiments by partially aggregating the local models in FedAvg. The results in Figure 4.2 show that the more shallow layers we keep unaggregated, the less feature degradation is observed in the final layer. The above results suggest that performance degradation may not stem solely from parameter divergence that has been explored previously Li et al. (2020), but is more closely associated with CFD-V.

**(2) Deeper features begin to compress only after aggregation stabilizes earlier layers.** As shown in Figures 3 - 4, during FL training, feature variance in shallower layers first converges to a stable level, after which deeper layers begin to exhibit compression. If this condition is not satisfied, model aggregation can instead destabilize features in deeper layers. This behavior is closely related to the aforementioned CFD-V and further complicates the convergence of FL models under aggregation.

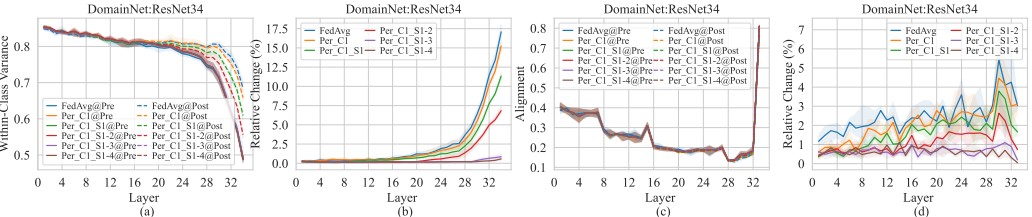

Figure 7: Changes in the normalized within-class feature variance and the alignment between features and parameters under various aggregation strategies. Per_* denotes keeping these parts unaggregated, with C1 denotes the first convolutional layer and S* denotes the block.

> **Takeaway**
>
> Feature variance degradation can accumulate due to the stacked architecture of DNNs, leading to severe disruption in penultimate-layer features despite minimal parameter divergence.

## 4.3 FEATURE-PARAMETER MISMATCH AFTER MODEL AGGREGATION

**Motivation.** In the previous sections, we analyze how the features themselves are influenced by model aggregation. However, due to the stacked architecture of DNNs, both the progressive feature extraction process and the final decision stage depend not only on the quality of the features, but also on their coupling with the parameters of subsequent layers. In this section, we examine the alignment between features and their subsequent parameters to understand how model aggregation influences the consistency between features and the parameters that transform them across layers.

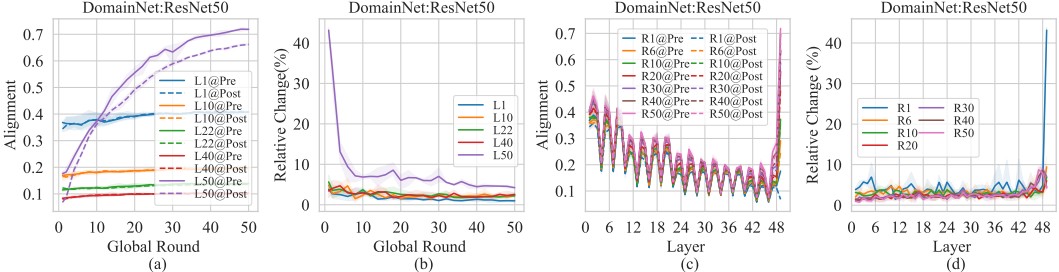

Figure 8: Evolution in feature-parameter alignment. The model is trained on DomainNet using ResNet50. Subfigures (a) and (b) show the alignment and their relative changes across global rounds. Subfigures (c) and (d) show the same metrics across model depth.

**Experimental Results** Figure 8 presents the results on DomainNet using Reset50. Additional results are provided in Appendix J. Based on Figure 8, we can make the following observations:

**(1) Feature-parameter alignment increases as training progresses.** From Figure 8 (a) and (c), it can be observed as training proceeds, the alignment between features at a give layer and their subsequent parameters gradually increases. This suggests that during the training process, the transformation between layers becomes progressively more coherent, allowing downstream parameters to better accommodate upstream feature representations. At the early stages of training, the feature-parameter alignment decreases with increasing model depth, reflecting weaker coupling in deeper layers.

However, the alignment between the penultimate layer (L50) and the classifier improves at a much faster rate than other layers, eventually surpassing all others after several global rounds.

**(2) Feature-parameter alignment is disrupted by model aggregation and exhibits a cumulative degradation trend.** As shown in Figure 8 (a) and (c), model aggregation clearly disrupts the alignment between features and their subsequent parameters. Furthermore, as shown in Figures 8 (b) and (d), the relative change in feature-parameter alignment also exhibits a CFD trend, which we denote as CFD-A. This results in a pronounced mismatch between the penultimate-layer features and the classifier. Notably, unlike feature variance, the relative change in alignment shows a sharp spike specifically at the interface between the penultimate layer and the classifier—substantially greater than the changes observed in earlier layers. This pronounced mismatch, together with the previously observed decline in feature discrimination, jointly accounts for the significant performance degradation observed in the post-aggregated model.

> **Takeaway**
>
> Model aggregation induces progressive misalignment between features and subsequent parameters, with the most severe disruption occurring between the penultimate layer and the classifier.

### 4.4 DOES CFD PERSIST IN MORE SCENARIOS?

**Motivation.** In this section, we aim to demonstrate whether CFD persists across more modalities and FL algorithms.

**Experimental Results.** Figure 9 shows the results for text classification on AmazonReviews dataset, and Figure 10 presents the experimental results for various advanced FL algorithms. It can be observed that the CFD phenomenon persists across more modalities and advanced FL algorithms. This suggests that CFD is not

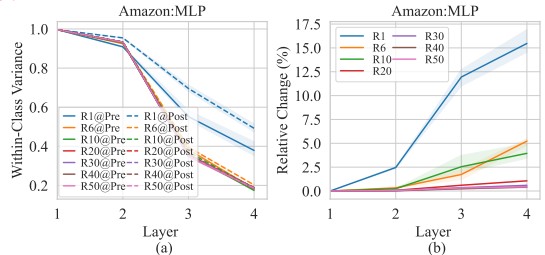

Figure 9: Normalized feature variances across layers at specific rounds on text classification task.

a peculiarity of FedAvg or vision tasks, but rather a common consequence of aggregating locally adapted models in data-heterogeneous FL.

Figure 10: Changes in the normalized within-class feature variance and the alignment between features and parameters with various advanced FL methods.

> **Takeaway**
>
> The CFD phenomenon persists across various data modalities and advanced FL algorithms.

### 4.5 MODEL AGGREGATION IMPROVES MODEL GENERALIZATION

**Motivation.** Previous experiments show that model aggregation disrupts locally formed feature structures, reducing feature discrimination and degrading performance on local client data. This raises a key question: if aggregation consistently harms local representations, what is the value of client collaboration in FL? To address this, we shift focus from feature discrimination to generalization, aiming to uncover the potential benefits of aggregation across diverse data distributions.

**Experimental Results.** We use linear probing accuracy to evaluate feature generalization. Figure 11 reports results on 'Real' and 'Sketch' domains using ResNet34. Detailed settings and additional results are provided in Appendix M. For clarity, we refer to the linear probing accuracy of the pre-aggregated models on its own local data as in-distribution (ID) accuracy, and its performance on other clients' data as out-of-distribution (OOD) accuracy. Based on Figure 11, we can draw the following observations:

**(1) As network depth increases, ID accuracy improves and then plateaus, while OOD accuracy rises initially but eventually declines.** We observe that the ID accuracy of the pre-aggregated model increases with depth and stabilizes at the final few layers, indicating that the model progressively extracts more discriminative features from its local data. In contrast, the OOD accuracy exhibits a different trend—it first increases, then decreases—indicating reduced generalization to other clients' data at deeper layers. This observation aligns with prior findings that DNNs tend to learn generalizable features in shallow layers and task-specific features in deeper layers (Yosinski et al., 2014; Masarczyk et al., 2024). These results suggest that in data-heterogeneous FL, clients may benefit more from collaboratively training on intermediate-layer features rather than relying solely on the final features.

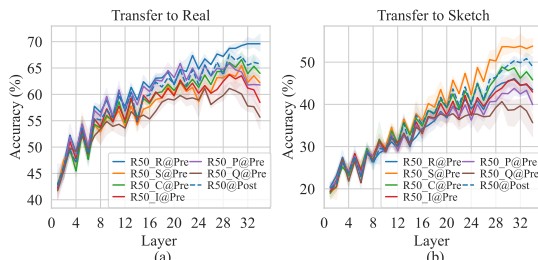

Figure 11: Linear probing accuracy when transferring to 'Real' and 'Sketch' domains in Domain-Net. Experiments are conducted on ResNet34. R50@Post denotes the results of post-aggregated model, while R50_@Pre* indicates the results of pre-aggregated models from domain *.

**(2) Post-aggregated model produces more generalizable features than the pre-aggregated model.** As shown in Figure 11, although the pre-aggregated model performs well on ID data, it exhibits a significant drop in OOD accuracy. This indicates that the pre-aggregated model tends to overfit its local data and fails to learn features that generalize to other clients. In contrast, the post-aggregated model maintains relatively high performance across diverse data distributions, indicating that aggregation effectively fuses knowledge from clients and yields more generalizable features.

> **Takeaway**
>
> Model aggregation mitigates the overfitting of local models by fusing locally learned knowledge and facilitates the extraction of more generalizable features across diverse data distributions.

## 5 HOW KEY FACTORS AFFECT MODEL AGGREGATION?

**Motivation.** In this section, we analyze how key settings in FL, such as model aggregation frequency and the number of clients, affect aggregation from the layer-peeled feature extraction perspective.

**Experimental Results.** Figure 12 presents the results with different model aggregation frequencies, while keeping the total number of local updates fixed at 100 and 500, respectively. It can be observed that increasing the number of local epochs more effectively compresses the within-class features but, in contrast, increases the CFD phenomenon. Figure 13 presents the results with different numbers of clients, while keeping the total training samples within the same domain constant. It can be seen that increasing the number of participating clients during FL training also exacerbates the CFD phenomenon. This demonstrates that addressing CFD is even more critical in practical applications.

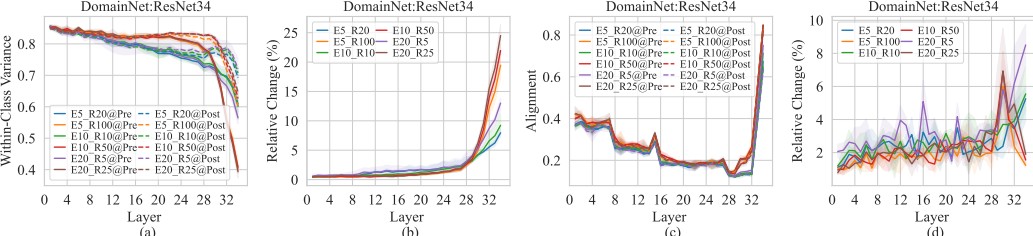

Figure 12: Changes in the normalized within-class feature variance and the alignment between features and parameters when training the FL model with different model aggregation frequency.

> **Takeaway**
>
> CFD becomes more pronounced as aggregation frequency decreases and client number increases.

## 6 WHY ARE COMMON SOLUTIONS EFFECTIVE?

In this section, we revisit several common yet effective solutions to address the accuracy drop caused by model aggregation using our layer-peeled feature analysis framework. These solutions

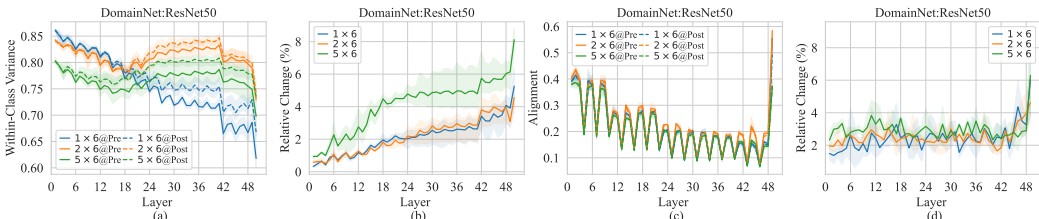

Figure 13: Changes in the normalized within-class feature variance and the alignment between features and parameters when training the FL model with different numbers of clients.

include parameter personalization, pre-trained parameters initialization, and classifier fine-tuning. Our analysis offers a deeper understanding of why these methods are effective, by linking their impact to the feature-level disruptions and alignment issues identified in the previous sections.

## 6.1 PERSONALIZING SPECIFIC PARAMETERS

**Motivation.** Parameter personalization has been shown to be effective in mitigating the performance degradation caused by model aggregation (Arivazhagan et al., 2019; Li et al., 2021b; Liang et al., 2020; Sun et al., 2021). In this section, we investigate how personalization affects the feature extraction process from a layer-peeled perspective. We implement various personalization strategies including FedPer(Arivazhagan et al., 2019), FedBN (Li et al., 2021b), and PartialFed (Sun et al., 2021). More details can be found in Appendix O.

**Experimental Results.** Figure 14 presents the results on DomainNet using ResNet34. Additional results are provided in Appendix O. It can be observed that personalizing more parameters within the feature extractor generally leads to more compact within-class features and smaller changes in feature variance after model aggregation. This benefit arises because personalizing shallow layers helps prevent the CFD effect—preserving the locally adapted feature extraction capability that can be disrupted by aggregation.

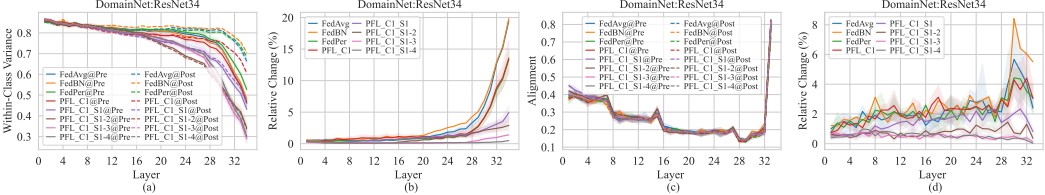

Figure 14: Changes in the normalized within-class feature variance and the alignment between features and parameters with various personalization strategies. PFL_* denotes the PFL methods that personalize different parts, with C1 denotes the first convolutional layer and S* denotes the block.

> **Takeaway**
>
> Personalizing shallow layers preserves locally adapted feature representations and mitigates the CFD effects introduced by model aggregation.

## 6.2 INITIALIZING WITH PRE-TRAINED PARAMETERS ON LARGE-SCALED DATASET

**Motivation.** Pre-trained parameters have been widely adopted to initialize FL models and have been shown to accelerate convergence (Nguyen et al., 2023; Chen et al., 2023). However, these studies primarily rely on loss or accuracy to assess the effects of pre-trained parameters. In this section, we investigate how initializing with pre-trained parameters affects the layer-peeled feature extraction.

**Experimental Results.** Figure 15 presnt the results on DomainNet using ResNet50 pretrained on ImageNet. Additional experimental results are provided in Appendix R. It can be observed that both the within-class variance and feature-parameter alignment exhibit lower sensitivity to model aggregation when models are initialized with pre-trained parameters compared to random initialization. This is because pre-training enables the shallow layers to extract meaningful features

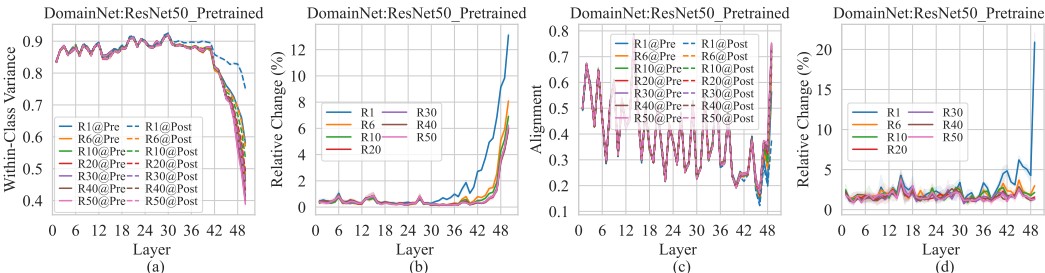

Figure 15: Changes in within-class feature variance and feature-parameter alignment across layers at specific global rounds. The model is trained on DomainNet using ResNet50 pre-trained on ImageNet.

early on, allowing the model to concentrate training efforts on deeper layers. Such initialization helps mitigate the CFD phenomenon discussed in Section 4.2, where deeper layers become increasingly sensitive to model aggregation and only begin to converge after sufficient compression in preceding layers. By accelerating the stabilization of shallow-layer features, pre-training effectively reduces the effective path length of CFD, thereby significantly lowering the relative changes in feature variance and alignment introduced by model aggregation.

> **Takeaway**
>
> Pre-trained initialization shortens the effective path length of CFD, thereby alleviating the negative impact of model aggregation on feature stability.

### 6.3 FINE-TUNING CLASSIFIER USING LOCAL DATA

**Motivation.** Fine-tuning the classifier using local data has been shown to effectively improve a model's adaptation to local distributions (Oh et al., 2022; Li et al., 2023). However, the underlying mechanism behind this improvement has not been thoroughly explored. In this section, we analyze the effect of classifier fine-tuning using the feature-parameter alignment metric in our framework.

**Experimental Results.** Figure 16 shows both accuracy and feature-parameter alignment on DomainNet when fine-tuning the classifier at various global rounds, using ResNet18 as the backbone. Additional experiments are provided in Appendix Q. We observe that the alignment between penultimate layer features and the classifier consistently improves with fine-tuning as FL training progresses. This suggests that fine-tuning strengthens the coupling between penultimate-layer features and the classifier, ultimately enhancing the model performance.

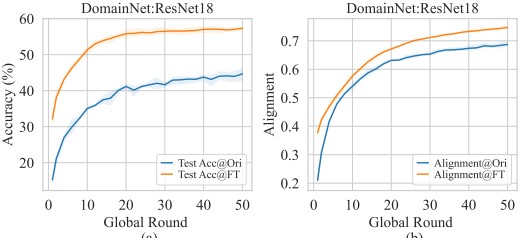

Figure 16: Accuracy and feature-parameter alignment when fine-tuning classifier. The experiments are conducted on DomainNet using ResNet18.

> **Takeaway**
>
> Fine-tuning classifier align global classifier with local features, thereby improving model adaptation and performance on local data.

## 7 CONCLUSION

In this paper, we introduce a layer-peeled feature analysis framework to study the impact of model aggregation on feature extraction in FL. Our analysis reveals that while FL's training dynamics generally align with those of CL, aggregation disrupts feature quality and alignment, with degradation accumulating across network layers. We show that strategies like parameter personalization, pre-trained initialization, and classifier fine-tuning effectively mitigate this degradation. Our work provides new insights into model aggregation in FL, potentially guiding the development of more robust and interpretable FL algorithms.

ETHICS STATEMENT

We affirm that our research adheres to the ICLR Code of Ethics. Our study does not involve human subjects, nor does it raise any ethical concerns related to data privacy, discrimination, or harmful insights. The datasets used in our experiments are publicly available and properly cited, and all methods and analyses are conducted in accordance with research integrity standards. We have no conflicts of interest or external sponsorship that would bias the results presented in this paper. Our work complies with legal and ethical standards, and we are committed to transparency and fairness in both the methodology and the application of our findings.

REPRODUCIBILITY STATEMENT

We strive to ensure the reproducibility of our work. The source code for the experiments and models in this paper is available at: `https://anonymous.4open.science/r/ICLR_14921_Code-3565`. In the main text and supplementary materials, we provide detailed descriptions of the datasets, model architectures, and training procedures used to enable reproducibility of our findings.

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

# A    APPENDIX

# APPENDIX

# CONTENTS

## B   RELATED WORK

**Federated Learning.** FedAvg (McMahan et al., 2017) is widely recognized as the pioneering work in FL. It facilitates collaborative model training by iteratively performing independent training on each client and aggregating the models on the server. While FedAvg demonstrates satisfactory performance in IID settings (Stich, 2018; Woodworth et al., 2020), a performance drop is commonly observed when data distributions are non-IID. This phenomenon has led to multiple explanations for the degradation, such as 'client drift' (Zhao et al., 2018; Karimireddy et al., 2020) and 'knowledge forgetting' (Jin et al., 2022). To address the challenges posed by non-IID data, various solutions have been proposed, including local model regularization (Li et al., 2020), correction techniques (Karimireddy et al., 2020; Wu et al., 2022), knowledge distillation (Jin et al., 2022; Lee et al., 2022), and partial parameter personalization (Li et al., 2021b; Collins et al., 2021; Arivazhagan et al., 2019; Liang et al., 2020; Sun et al., 2021; Zheng et al., 2022; Wu et al., 2023; 2024). However, these methods generally adhere to the standard framework of local training and global aggregation introduced by FedAvg. There remains a lack of in-depth exploration into how model aggregation intuitively impacts model training from a feature extraction perspective.

**Federated Learning within Feature Space.** Numerous studies have recently observed that heterogeneous data can lead to suboptimal feature extraction in FL models and are working on improving FL models by directly calibrating the resulting suboptimal feature spaces. A common line of research attributes performance degradation to inconsistent feature spaces across clients. To tackle this problem, CCVR (Luo et al., 2021) introduces a post-calibration strategy that fine-tunes the classifier after FL training using virtual features generated by an approximate Gaussian Mixture Model (GMM). Several methods (Tan et al., 2022; Zhu et al., 2022; 2023; Zhou et al., 2023; Huang et al., 2023) propose using prototypes to align feature distributions across different clients, ensuring a consistent feature space. Studies such as FedBABU (Oh et al., 2022), SphereFed (Dong et al., 2022), and FedETF (Li et al., 2023) utilize various fixed classifiers (e.g., random or orthogonal initialization) as targets to align features across clients. Fed2 Yu et al. (2021) proposes a structural alignment method based on the permutation invariance of DNNs. In addition to inconsistent feature spaces, FedDecorr (Shi et al., 2023a;b) observed that heterogeneous data leads to severe dimensional collapse in the global model. To combat this, it applies a regularization term based on the Frobenius norm of the correlation matrix during local training, encouraging the different dimensions of features to remain uncorrelated. FedPLVM (Wang et al., 2024) discovers differences in domain-specific feature variance in cross-domain FL. Consequently, they propose dual-level prototype clustering that adeptly captures variance information and addresses the aforementioned problem. However, these studies primarily focus on calibrating features in the penultimate layer while overlooking intermediate features during feature extraction. In this paper, we present a layer-peeled analysis of how model aggregation affects the feature extraction process and identify several issues related to the hierarchical topology of DNNs.

**Feature Learning in DNNs.** Advanced DNNs are typically structured with hierarchical layers, enabling them to efficiently and automatically extract informative features from raw data (Krizhevsky et al., 2012; Allen-Zhu & Li, 2023; Wang et al., 2023b). Numerous efforts have been made to understand how DNNs transform raw data from shallow to deep layers. A commonly accepted view is that DNNs initially extract transferable universal features and then progressively filter out irrelevant information to form task-specific features (Yosinski et al., 2014; Zeiler & Fergus, 2014; Evci et al., 2022; Kumar et al., 2022). Another line of studies demonstrates that DNNs progressively learn features that are compressed within classes and discriminative between classes. Specifically, (Alain & Bengio, 2017) observes that the linear separability of features increases as the layers become deeper, using a linear probe. (Masarczyk et al., 2024) proposes the tunnel effect hypothesis, which states that the initial layers create linearly separable features, while the subsequent layers (referred to as the tunnel) compress these features. Meanwhile, certain research has extended the feature analysis of neural collapse (NC)—originally observed in the penultimate layers (Papyan et al., 2020)—to intermediate layers (Ansuini et al., 2019; Rangamani et al., 2023; Li et al., 2024), showing that the features of each layer exhibit gradual collapse as layer depth increases. (Wang et al., 2023b) provides a theoretical analysis of feature evolution across depth based on deep linear networks (DLNs). These studies focus on centralized training where there is no model aggregation during training, they provide solid support for us to effectively analyze feature evolution in FL, especially the influence of model aggregation on feature extraction. It should be noted that several recent studies have begun to explore FL from a layer-wise or layer-peeled perspective (Luo et al., 2021; Chan et al., 2024; Adilova et al., 2024; Ma et al., 2022; Zhou & Konukoglu, 2023). However, these works

differ significantly from ours in both methodology and analytical focus. For example, Adilova et al. (Adilova et al., 2024) investigate aggregation behavior primarily in the parameter space and use loss-based metrics to characterize layer-wise changes, without directly analyzing intermediate feature representations. Ma et al. (Ma et al., 2022) propose assigning personalized aggregation weights at a layer-wise granularity, but they do not consider the feature extraction process of DNNs in FL. Studies such as (Luo et al., 2021; Chan et al., 2024) rely on feature similarity metrics that require pairwise comparisons between multiple models, which limits their applicability in single-model diagnostics or online evaluation. FedFA (Zhou & Konukoglu, 2023) proposes leveraging feature statistics from different layers to augment the features. However, it treats each layer's features independently, ignoring the dependencies between subsequent layers in the feature extraction process of DNNs. In contrast, our work proposes a unified feature-level evaluation framework that can be applied directly to a single model and its corresponding data. To the best of our knowledge, this is the first approach to characterize the dynamics of layer-wise feature extraction in FL using metrics that independently quantify feature quality and feature-parameter alignment—without the need for auxiliary models or downstream tasks.

## C  DATASET DESCRIPTION AND PARTITION

In the experiments, we primarily focus on the data-heterogeneous FL setting, where the distribution of raw input data differs across clients. This data heterogeneity, often referred to as cross-domain FL, is commonly observed in practical FL applications due to variations in data collection conditions across clients. Building on previous studies (Li et al., 2021b; Zhu et al., 2022; 2023; Wang et al., 2024), we use three widely used public cross-domain datasets: Digit-Five, PACS (Li et al., 2017), and DomainNet (Peng et al., 2019). These datasets contain multiple domains, with each domain consisting of images with different backgrounds and styles, which effectively simulate the data heterogeneity caused by variations in raw input.

The Digit-Five dataset includes images across 10 classes and 5 domains, namely: MNIST-M (Ganin & Lempitsky, 2015), MNIST (LeCun et al., 1998), USPS (Hull, 1994), SynthDigits (Ganin & Lempitsky, 2015), and SVHN (Netzer et al., 2011). The PACS dataset includes 4 distinct domains with a total of 7 classes: Photo (P), Art (A), Cartoon (C), and Sketch (S). The DomainNet dataset comprises six domains: Clipart (C), Infograph (I), Painting (P), Quickdraw (Q), Real (R), and Sketch (S). Initially, the DomainNet dataset includes 345 classes per domain. Based on prior research (Li et al., 2021b; Zhu et al., 2022), we reduce the number of classes to 10 commonly used ones for our layer-peeled feature analysis. Figure 17 shows some example images from these three datasets. The representative images demonstrate significant variations across different domains, as observed in Figure 17.

To create the data-heterogeneous setting across different clients, we assign the images from a single domain to each client. As a result, there are 5 clients for the Digit-Five dataset, 4 clients for PACS, and 6 clients for DomainNet in our analysis. For the Digit-Five dataset, the number of training and testing samples are set to 500 and 1000, respectively. For both the PACS and DomainNet datasets, the number of training and testing samples is set to 500. The images in the Digit-Five dataset are scaled to $32 \times 32$ for both the training and testing datasets. For PACS and DomainNet, the images are scaled to $224 \times 224$ and we apply data augmentations such as random flipping and rotation for the training samples. No data augmentations are applied to the Digit-Five or test datasets across all experiments.

In addition to the vision datasets above, we further include the AmazonReviews dataset (Blitzer et al., 2007) to examine whether CFD also emerges in the textual modality. Following the standard setup introduced in (Blitzer et al., 2007), the dataset consists of reviews from four distinct domains: Books, DVD, Electronics, and Kitchen. Each domain is treated as a separate client in our federated setting, thereby naturally inducing cross-domain heterogeneity in text distributions.

To construct a balanced cross-domain FL scenario comparable to the vision datasets, we randomly sample 500 training samples and 500 testing samples from each domain. All text inputs are first tokenized and mapped onto a fixed vocabulary, and then normalized before being fed into a 4-layer MLP encoder. As the input data is non-visual, no data augmentations are applied to this dataset.

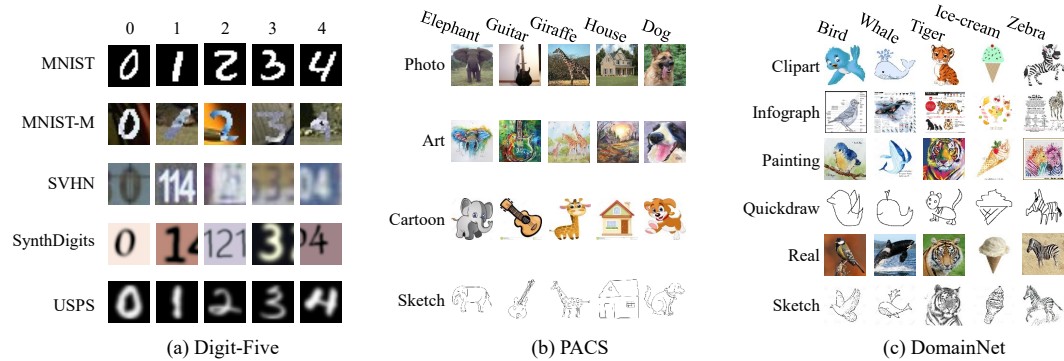

Figure 17: Visualization of example samples within the datasets used for layer-wise feature evaluation: (a) Digit-Five, (b) PACS, (c) DomainNet. For each domain within these adopted datasets, we show the representative samples from 5 classes.

## D   MODEL ARCHITECTURES

We utilize multiple models for the selected datasets to perform layer-wise feature extraction analysis, including both Convolutional Neural Networks (CNN) and Vision Transformers (ViT). Specifically, the CNN models we employ are ConvNet, VGG13_BN (Simonyan & Zisserman, 2014), and three variants of ResNet (He et al., 2016) (ResNet18, ResNet34, and ResNet50). For the ViT architecture, we apply ViT_B/16 (Dosovitskiy et al., 2021).

The ConvNet model consists of several convolutional layers followed by fully connected (FC) layers. The detailed architecture of ConvNet is presented in Table 1, with the output size calculated using the input scale of the Digit-Five dataset as an example. For the other models, we modify only the classifiers by adjusting the number of output classes to match the requirements of our dataset, while leaving the backbone architecture unchanged.

For the Digit-Five dataset, we use ConvNet, ResNet18, and ResNet34 for FL training. For both PACS and DomainNet datasets, we utilize ConvNet, VGG1_BN, ResNet18, ResNet34, ResNet50, and ViT_B/16 to accomplish the FL training.

Table 1: Detailed Architecture of ConvNet

| Layer | Output Size | Description |
|---|---|---|
| Input | 32x32x3 | Input image |
| Conv1_1 | 32x32x64 | 5x5 Convolution, 64 filters, stride 1, padding 2 |
| BN1_1 | 32x32x64 | Batch Normalization |
| ReLU1_1 | 32x32x64 | ReLU activation |
| MaxPool1_1 | 16x16x64 | 2x2 Max Pooling, stride 2 |
| Conv1_2 | 16x16x64 | 5x5 Convolution, 64 filters, stride 1, padding 2 |
| BN1_2 | 16x16x64 | Batch Normalization |
| ReLU1_2 | 16x16x64 | ReLU activation |
| MaxPool1_2 | 8x8x64 | 2x2 Max Pooling, stride 2 |
| Conv1_3 | 8x8x128 | 5x5 Convolution, 128 filters, stride 1, padding 2 |
| BN1_3 | 8x8x128 | Batch Normalization |
| ReLU1_3 | 8x8x128 | ReLU activation |
| Flatten | 8192 | Flatten layer for fully connected input |
| Linear2_1 | 2048 | Fully connected layer, 2048 units |
| BN2_1 | 2048 | Batch Normalization |
| ReLU2_1 | 2048 | ReLU activation |
| Linear2_2 | 512 | Fully connected layer, 512 units |
| BN2_2 | 512 | Batch Normalization |

| Layer | Output Size | Description |
|---|---|---|
| ReLU2_2 | 512 | ReLU activation |
| Output (C) | $C$ | Output layer (number of classes) |

# E IMPLEMENTATION DETAILS OF EXPERIMENTS

In our experiments, we train the model using the standard FL training process introduced by FedAvg (McMahan et al., 2017). This process involves iterative local model updating and global model aggregation. During the local updating phase, the model is optimized using each client's private data for $E$ epochs, after which the local models are uploaded to the server for aggregation. In the model aggregation stage, we apply parameter-wise averaging, using the number of samples as weights for each client's local model. The above procedures are repeated for $R$ global rounds until the global model converges.

For local model updating, we use stochastic gradient descent (SGD) with momentum for model optimization, where the learning rate is set to 0.01 and the momentum is set to 0.5. The batch size for local updates is set to 64. Unless otherwise specified, the number of local update epochs $E$ is set to 10, and the number of global rounds $R$ is set to 50.

All of our experiments are conducted using the PyTorch framework (Paszke et al., 2019) and implemented on a four-card Nvidia V100 (32G) cluster. During the layer-peeled feature analysis, we use the *forward hook* in PyTorch to extract input features from each evaluated layer as we move from shallow to deeper layers.

For ConvNet, the features preceding each convolutional and linear layer are used for evaluation. For ResNet, the features before each convolutional layer (excluding the first convolutional layer), the global average pooling layer, and the classifier are used for evaluation. For VGG13_BN, the features before each convolutional layer, the linear layers, and the classifier are used for evaluation. For ViT_B/16, we evaluate the features passed to the multilayer perceptron (MLP), self-attention layers, layer normalization layers, and the final classifier. For AmazonMLP, the features before each linear layer in the encoder and the classifier are used for evaluation.

To reduce computational cost during feature evaluation, we apply average pooling to the intermediate features, thereby lowering their dimensionality. For CNNs, let the intermediate features at the $\ell$-th layer be represented as $B \times H^\ell \times W^\ell \times C^\ell$, where $B$ denotes the batch size, and $H^\ell$, $W^\ell$, and $C^\ell$ are the height, width, and channel dimensions, respectively. Following previous studies (Sarfi et al., 2023; Harun et al., 2024), we apply $2 \times 2$ *adaptive average pooling* to the height and width dimensions ($H^\ell \times W^\ell$). After this adaptive average pooling operation, the intermediate features within the convolutional layers are reduced to $B \times 2 \times 2 \times C^\ell$. We then flatten this tensor into a one-dimensional vector and compute the feature metrics, where the features of one sample within the batch has a dimension of $4C^\ell$, and the total dimension of the samples within the batch is $B \times 4C^\ell$. For ViTs, following previous studies (Raghu et al., 2021; Harun et al., 2024), we apply *global average pooling* to aggregate the image tokens, excluding the class token. For features input to the linear layer with a two-dimensional shape of $B \times D^\ell$, we directly use these features to perform evaluation.

During feature evaluation, we process the features in a batch-wise manner, concatenating them across all batches in the evaluated dataset. The corresponding feature metrics are then computed based on these concatenated features. For all experiments, we evaluate the features every 20 local updating epochs, which corresponds to 2 global rounds when the local update epoch is set to 10. To minimize the impact of randomness, each experiment is repeated three times with different random seeds.

In the experiments involving additional FL algorithms, we follow the same local and global training configurations as described above and modify only the algorithm-specific hyperparameters. For FedProx (Li et al., 2020), the proximal coefficient is set to $\mu = 0.01$. For SCAFFOLD (Karimireddy et al., 2020), the control variate update uses a learning rate of $1.0$. For FedDyn (Acar et al., 2021), the dynamic regularization parameter is set to $\alpha = 0.01$. All other hyperparameters remain identical to those used in the FedAvg experiments.

## F   FEATURE EVALUATION METRICS

We apply the following metrics to evaluate the features generated by the pre-aggregated model and post-aggregated model, including the *feature variance, alignment between features and parameters, accuracy of linear probing, pairwise distance of features and models, relative change of evaluated metrics*. For simplicity, we omit the client and sample indices and focus solely on the computation process of these metrics. As previously stated, the features in our experiments are first stacked into a two-dimensional tensor, and then used to compute the feature metrics. We assume that the stacked features at $\ell$-th layer is denoted as $\boldsymbol{Z}^\ell \in R^{N \times D^\ell}$, where $N$ is the total number of samples used for feature evaluation, and $D^\ell$ denote the feature dimension for one sample at $\ell$-th layer. The metrics used to evaluate features in this paper are computed as follows.

### F.1   FEATURE VARIANCE

Building on previous studies (Rangamani et al., 2023; Wang et al., 2023b), we use within-class feature variance and between-class feature variance to evaluate the feature structure. Specifically, the within-class variance quantifies the degree of feature compression within the same class, while the between-class variance measures the degree of feature discrimination between different classes.

Before calculating the within-class and between-class variances, we first compute the within-class, between-class, and total covariance matrices of the features at $\ell$-th layer as follows:

$$\Sigma_W^\ell = \frac{1}{N} \sum_{c=1}^{C} \sum_{i=1}^{N_c} \left(\boldsymbol{z}_{c,i}^\ell - \boldsymbol{\mu}_c^\ell\right) \left(\boldsymbol{z}_{c,i}^\ell - \boldsymbol{\mu}_c^\ell\right)^\top$$

$$\Sigma_B^\ell = \frac{1}{C} \sum_{c=1}^{C} \left(\boldsymbol{\mu}_c^\ell - \boldsymbol{\mu}_G^\ell\right) \left(\boldsymbol{\mu}_c^\ell - \boldsymbol{\mu}_G^\ell\right)^\top \tag{3}$$

$$\Sigma_T^\ell = \frac{1}{N} \sum_{c=1}^{C} \sum_{i=1}^{N_c} \left(\boldsymbol{z}_{c,i}^\ell - \boldsymbol{\mu}_G^\ell\right) \left(\boldsymbol{z}_{c,i}^\ell - \boldsymbol{\mu}_G^\ell\right)^\top,$$

where

$$\boldsymbol{\mu}_c^\ell := \frac{1}{N_c} \sum_{i=1}^{N_c} \boldsymbol{z}_{c,i}^\ell \quad \boldsymbol{\mu}_G^\ell := \frac{1}{N} \sum_{c=1}^{C} \sum_{i=1}^{N_c} \boldsymbol{z}_{c,i}^\ell. \tag{4}$$

In the above equation, $\boldsymbol{\mu}_c^\ell$ is the mean class feature computed from the samples within the $c$-th class, and $\boldsymbol{\mu}_G^\ell$ as the global mean feature computed from all samples. $N_c$ represents the number of samples within the $c$-th class. It should be noted that the total covariance matrix can be decomposed as the sum of the within-class and between-class covariances, i.e.,

$$\Sigma_T^\ell = \Sigma_W^\ell + \Sigma_B^\ell. \tag{5}$$

Based on the computed covariance matrices, we then use the total variance $\text{Tr}(\Sigma_T^\ell)$ as the normalization factor and compute the normalized within-class variance and between-class variance as follows:

$$\bar{\sigma}_W^\ell = \frac{\text{Tr}(\Sigma_W^\ell)}{\text{Tr}(\Sigma_T^\ell)}, \tag{6}$$

$$\bar{\sigma}_B^\ell = \frac{\text{Tr}(\Sigma_B^\ell)}{\text{Tr}(\Sigma_T^\ell)}, \tag{7}$$

where $\text{Tr}(\cdot)$ denotes the trace of the covariance matrices. These two normalized variances are used to measure the within-class feature compression and between-class feature discrimination, respectively, in this paper.

### F.2 ALIGNMENT BETWEEN FEATURES AND SUBSEQUENT PARAMETERS

Following previous studies, we use the principal angles between subspaces (PABS) (Rangamani et al., 2023; Jordan, 1875; Björck & Golub, 1973)—denoted as $\theta_1, \ldots, \theta_C$—to measure the alignment between the range space of class-wise feature means $\bar{\boldsymbol{Z}}^\ell$ and the top $C$-rank subspace of the subsequent input layer parameters $\boldsymbol{W}^{\ell+1}$.

Specifically, for a linear layer, we first apply singular value decomposition (SVD) to $\boldsymbol{W}^{\ell+1}$ and $\bar{\boldsymbol{Z}}^\ell$, i.e., $\boldsymbol{W}^{\ell+1} = \boldsymbol{U}_{\boldsymbol{W}}^{\ell+1} \boldsymbol{S}_{\boldsymbol{W}}^{\ell+1} (\boldsymbol{V}_{\boldsymbol{W}}^{\ell+1})^T$ and $\bar{\boldsymbol{Z}}^\ell = \boldsymbol{U}_{\bar{\boldsymbol{Z}}}^\ell \boldsymbol{S}_{\bar{\boldsymbol{Z}}}^\ell (\boldsymbol{V}_{\bar{\boldsymbol{Z}}}^\ell)^T$. We then compute the PABS between $\boldsymbol{V}_{\boldsymbol{W}}^{\ell+1}$ and $\boldsymbol{U}_{\bar{\boldsymbol{Z}}}^\ell$, which represent the basis for the input subspace $\boldsymbol{W}^{\ell+1}$ and the range space of $\bar{\boldsymbol{Z}}^\ell$, respectively. The alignment is finally computed by the mean of the singular values of $(\boldsymbol{V}_W^{\ell+1})^T \boldsymbol{U}_{\bar{\boldsymbol{Z}}}^\ell$.

For the convolutional layer, assume the filter kernel has shape $\boldsymbol{W}^{\ell+1} \in R^{C_{\text{out}}^{\ell+1} \times C_{\text{in}}^\ell \times k_H^{\ell+1} \times k_W^{\ell+1}}$, and the class-wise feature means have shape $\bar{\boldsymbol{Z}}^\ell \in \mathbb{R}^{C \times C_{\text{in}}^\ell \times H^\ell \times W^\ell}$. As previously stated, $\bar{\boldsymbol{Z}}^l$ can be reshaped into $C \times (C_{\text{in}}^\ell \times 4)$. We begin by flattening the features and parameters along the $C_{\text{in}}^\ell$ dimension, resulting in shapes $C_{\text{in}}^\ell \times (C \times 4)$ for the features and $C_{\text{in}}^\ell \times (C_{\text{out}}^{\ell+1} \times k_H^{\ell+1} \times k_W^{\ell+1})$ for the parameters. We then compute the alignment along the $C_{\text{in}}^\ell$ dimension as described above.

For the self-attention layer in a ViT, we compute the alignments of the features and the $QKV$ matrices separately, and then take the average of these alignments as the final metric.

### F.3 ACCURACY OF LINEAR PROBING

Linear probing is a technique used in transfer learning to evaluate the quality of learned features by training a simple linear classifier on top of the features extracted from a pre-trained model (Chen et al., 2020; He et al., 2022; Wang et al., 2023a). In this paper, we employ the linear probing technique to assess feature generalization across diverse data distributions. Specifically, after extracting features from different layers, we apply a randomly initialized linear classifier on top of these features. This classifier is then trained using the training subset of the evaluated datasets, and we compute accuracy by testing on the corresponding samples from the test datasets. The testing accuracy serves as the metric for evaluating feature generalization on each dataset.

### F.4 PAIRWISE DISTANCE OF FEATURES AND MODELS

We use four metrics to evaluate the distance between two models or the features extracted by them: *mean normalized $L_1$ distance, mean squared distance, mean $L_1$ distance, and cosine similarity*. In this section, we focus on the computation of the distance between features, since the model parameters are reshaped into a single vector, which can be treated as a feature with a sample size of 1. Thus, the distance computation is applied to these features can be directly transferred to models. Let the features of the pre-aggregated and post-aggregated models be denoted as $\boldsymbol{Z}_{pre}^\ell$ and $\boldsymbol{Z}_{post}^\ell$, respectively. The corresponding distance can then be computed as follows.

- **Mean Normalized $L_1$ Distance.** This measure computes the mean normalized $L_1$ distance between the pre-aggregated and post-aggregated feature matrices, and then averages the distances across all elements, as shown below.

$$D_{\hat{L}_1}^\ell = \frac{1}{ND} \sum_{i=1}^N \sum_{j=1}^D \frac{|\boldsymbol{Z}_{pre}^\ell(i,j) - \boldsymbol{Z}_{post}^\ell(i,j)|}{|\boldsymbol{Z}_{pre}^\ell(i,j)| + |\boldsymbol{Z}_{post}^\ell(i,j)|} \tag{8}$$

- **Mean Squared Error.** This distance measure computes the average squared differences between corresponding elements of the feature vectors, as shown below.

$$D_s^\ell = \frac{1}{ND} \sum_{i=1}^N \sum_{j=1}^D (\boldsymbol{Z}_{pre}^\ell(i,j) - \boldsymbol{Z}_{post}^\ell(i,j))^2 \tag{9}$$

- **Mean $L_1$ Distance.** This distance computes the average $L_1$ distances between corresponding elements of the pre-aggregated and post-aggregated feature matrices, as shown below.

$$D_{L_1}^\ell = \frac{1}{ND} \sum_{i=1}^N \sum_{j=1}^D |\boldsymbol{Z}_{pre}^\ell(i,j) - \boldsymbol{Z}_{post}^\ell(i,j)| \tag{10}$$

- **Mean Cosine Similarity.** This measure computes the cosine of the angle between the pre-aggregated and post-aggregated features of the same samples, and then averages these values across all clients. It quantifies the cosine of the angle between the pre-aggregated and post-aggregated features, where a value of 1 indicates identical directions and a value of -1 indicates opposite directions. The formulation of the mean cosine similarity is shown below.

$$D_{cos}^\ell = \frac{1}{N} \sum_{i=1}^N \frac{Z_{pre}^\ell(i,:) \cdot Z_{post}^\ell(i,:)}{||Z_{pre}^\ell(i,:)||||Z_{post}^\ell(i,:)||} \tag{11}$$

where $Z_{pre}^\ell(i,:) \cdot Z_{post}^\ell(i,:)$ denotes the dot product of the pre-aggregated and post-aggregated features of sample $i$ at $\ell$-th layer, respectively, and $||Z_{pre}^\ell(i,:)||$ and $||Z_{post}^\ell(i,:)||$ denote the Euclidean norms of the pre-aggregated and post-aggregated features at $\ell$-th layer, respectively.

## F.5 RELATIVE CHANGE OF EVALUATED METRICS

Since the original metrics of features at different layers can vary in magnitude, we use the relative change in the evaluated metrics to measure the ratio of change before and after aggregation. Let $V_{pre}^\ell$ and $V_{post}^\ell$ represent the metrics of features generated by the models before and after aggregation at $\ell$-th layer, respectively. The relative change in the evaluated metrics is then defined as:

$$\Delta^\ell(V) = \frac{|V_{post}^\ell - V_{pre}^\ell|}{|V_{pre}^\ell| + |V_{post}^\ell|} * 100\%. \tag{12}$$

## G THEORETICAL ANALYSIS

In this section, we provide a theoretical framework to analyze the mechanism of performance degradation in FL under data heterogeneity. Adopting the Deep Linear Networks (DLN) formulation Wang et al. (2023b), we rigorously prove three key phenomena: (1) Model aggregation disrupts feature statistics by inflating within-class variance and shrinking between-class variance; (2) It reduces the alignment between features and subsequent parameters; and (3) These degradations accumulate along the network depth, resulting in the most severe deterioration at the penultimate layer.

### G.1 PRELIMINARIES

Consider an $L$-layer neural network parameterized by $\Theta = \{\mathbf{W}^\ell\}_{\ell=1}^L$. The feature representation at layer $\ell$ is defined recursively as $\mathbf{z}^\ell = \mathbf{W}^\ell \mathbf{z}^{\ell-1}$, with $\mathbf{z}^0 = \mathbf{x}$. In a Federated Learning system with $M$ clients, let $\mathbf{W}_m^\ell$ denote the optimal local parameters for client $m$, and $\tilde{\mathbf{W}}^\ell = \sum_{m=1}^M \alpha_m \mathbf{W}_m^\ell$ denote the aggregated global parameters (where $\alpha_m$ is the aggregation weight, e.g., $N_m/N$).

**Definition 1** (Heterogeneity Drift). *Consider a Federated Learning system with $M$ clients. Let $\mathbf{W}_m^\ell$ denote the optimal local parameters for client $m$ at layer $\ell$, and $\tilde{\mathbf{W}}^\ell = \sum_{m=1}^M \frac{N_m}{N} \mathbf{W}_m^\ell$ denote the aggregated global parameters. We define the **Heterogeneity Drift** matrix $\boldsymbol{\Delta}_m^\ell$ as the deviation of the local parameters from the global parameters:*

$$\boldsymbol{\Delta}_m^\ell \triangleq \mathbf{W}_m^\ell - \tilde{\mathbf{W}}^\ell. \tag{13}$$

*We assume the magnitude of this drift is bounded by a constant $\delta_\ell$, such that $\|\boldsymbol{\Delta}_m^\ell\|_F \le \delta_\ell$, where $\delta_\ell$ quantifies the intensity of data heterogeneity at layer $\ell$. By definition of the weighted average, the drifts satisfy the zero-mean property: $\sum_{m=1}^M \frac{N_m}{N} \boldsymbol{\Delta}_m^\ell = \mathbf{0}$.*

### G.2 PART I: DISRUPTION OF FEATURE STATISTICS

We first analyze the impact of aggregation on feature clustering properties at a single layer level.

**Proposition 1** (Variance Shift). *Due to heterogeneity-induced noise, model aggregation increases the within-class variance ($\Sigma_W$) and decreases the between-class variance ($\Sigma_B$) of features on local data, compared to the optimal local model:*

$$\mathrm{Tr}(\tilde{\Sigma}_W^\ell) > \mathrm{Tr}(\Sigma_{W,m}^\ell) \quad and \quad \mathrm{Tr}(\tilde{\Sigma}_B^\ell) < \mathrm{Tr}(\Sigma_{B,m}^\ell). \tag{14}$$

*Proof.* **1. Proof of Within-class Variance Inflation.** The within-class covariance measures the dispersion of features around their class means. Let $\mathbf{h} = \mathbf{z}^{\ell-1}$ be the input feature. The aggregated feature output can be decomposed as:

$$\tilde{\mathbf{z}}^\ell = \tilde{\mathbf{W}}^\ell \mathbf{h} = (\mathbf{W}_m^\ell - \boldsymbol{\Delta}_m^\ell)\mathbf{h} = \underbrace{\mathbf{W}_m^\ell \mathbf{h}}_{\mathbf{z}_m^\ell \text{ (Local Signal)}} + \underbrace{(-\boldsymbol{\Delta}_m^\ell \mathbf{h})}_{\boldsymbol{\mathcal{E}} \text{ (Heterogeneity Noise)}}. \tag{15}$$

We assume the heterogeneity drift $\boldsymbol{\Delta}_m^\ell$ acts as unstructured noise that is uncorrelated with the specific semantic structure of local features. Applying the trace operator to the covariance matrix:

$$\begin{aligned} \mathrm{Tr}(\tilde{\Sigma}_W^\ell) &= \mathrm{Tr}(\mathrm{Cov}(\mathbf{z}_m^\ell + \boldsymbol{\mathcal{E}})) \\ &= \mathrm{Tr}(\mathrm{Cov}(\mathbf{z}_m^\ell)) + \mathrm{Tr}(\mathrm{Cov}(\boldsymbol{\mathcal{E}})) + 2\mathrm{Tr}(\mathrm{Cov}(\mathbf{z}_m^\ell, \boldsymbol{\mathcal{E}})). \end{aligned} \tag{16}$$

Given the uncorrelated assumption, the cross-term vanishes ($\approx 0$). Since data heterogeneity implies $\boldsymbol{\Delta}_m^\ell \neq 0$, the noise variance is strictly positive, i.e., $\mathrm{Tr}(\mathrm{Cov}(\boldsymbol{\mathcal{E}})) > 0$. Thus:

$$\mathrm{Tr}(\tilde{\Sigma}_W^\ell) > \mathrm{Tr}(\Sigma_{W,m}^\ell). \tag{17}$$

This indicates that aggregation introduces noise that expands the feature clusters, degrading compression.

**2. Proof of Between-class Variance Shrinkage.** Between-class variance relies on the magnitude of class centroids. Let $\mathbf{c}_{m,k}$ be the centroid of class $k$ produced by client $m$'s local model. The centroid produced by the aggregated model is the weighted average of local centroids:

$$\tilde{\mathbf{c}}_k = \tilde{\mathbf{W}}^\ell \boldsymbol{\mu}_k = \sum_{m=1}^M \alpha_m (\mathbf{W}_m^\ell \boldsymbol{\mu}_k) = \sum_{m=1}^M \alpha_m \mathbf{c}_{m,k}, \tag{18}$$

where $\boldsymbol{\mu}_k$ is the input mean for class $k$. In non-IID settings, local models optimize feature mappings in divergent directions to fit local distributions, causing the vectors $\{\mathbf{c}_{m,k}\}_m$ to be *misaligned* (i.e., pointing in different directions). By the strict triangle inequality for non-collinear vectors:

$$\|\tilde{\mathbf{c}}_k\|_2 = \left\| \sum_{m=1}^M \alpha_m \mathbf{c}_{m,k} \right\|_2 < \sum_{m=1}^M \alpha_m \|\mathbf{c}_{m,k}\|_2. \tag{19}$$

This implies that the magnitude (energy) of the class centroids in the aggregated model is strictly smaller than the average magnitude of the local centroids. A reduced centroid magnitude directly corresponds to collapsed distances between classes (shrinkage towards the origin). Therefore, $\mathrm{Tr}(\tilde{\Sigma}_B^\ell) < \mathrm{Tr}(\Sigma_{B,m}^\ell)$, proving the loss of discrimination. $\square$

### G.3 PART II: REDUCTION OF FEATURE-PARAMETER ALIGNMENT

We quantify the coupling between features and subsequent parameters using the Principal Angles Between Subspaces (PABS).

**Proposition 2** (Alignment Reduction). *The feature-parameter alignment score of the aggregated model is strictly lower than that of the optimal local model:*

$$\mathcal{A}(\tilde{\mathbf{Z}}_m^\ell, \tilde{\mathbf{W}}^{\ell+1}) < \mathcal{A}(\mathbf{Z}_m^\ell, \mathbf{W}_m^{\ell+1}). \tag{20}$$

*Proof.* Let $\mathbf{Z}_m^\ell$ be the matrix of class-wise feature means and $\mathbf{W}^{\ell+1}$ be the weight of the subsequent layer. Let their SVDs be $\mathbf{Z}_m^\ell = \mathbf{U}_{\mathbf{Z}} \mathbf{S}_{\mathbf{Z}} \mathbf{V}_{\mathbf{Z}}^\top$ and $\mathbf{W}^{\ell+1} = \mathbf{U}_{\mathbf{W}} \mathbf{S}_{\mathbf{W}} \mathbf{V}_{\mathbf{W}}^\top$. The alignment is measured by the cosine similarity of their principal subspaces: $\mathcal{A} = \mathrm{mean}(\sigma(\mathbf{V}_{\mathbf{W}}^\top \mathbf{U}_{\mathbf{Z}}))$.

**Step 1: Local Optimality.** Local training maximizes the signal propagation, aligning the input subspace of the weights ($\mathbf{V}_{\mathbf{W}}$) with the output subspace of the features ($\mathbf{U}_{\mathbf{Z}}$). Thus, for the local model, $\mathbf{V}_{\mathbf{W}_m}^\top \mathbf{U}_{\mathbf{Z}_m} \approx \mathbf{I}$, and $\mathcal{A}_{local} \approx 1$.

**Step 2: Subspace Perturbation.** For the aggregated model, perturbations arise from two independent sources:

- **Weight Drift:** $\tilde{\mathbf{W}} = \mathbf{W}_m - \boldsymbol{\Delta}_W$. By matrix perturbation theory, this rotates the right singular vectors: $\tilde{\mathbf{V}}_{\mathbf{W}} \approx \mathbf{V}_{\mathbf{W}_m} + \mathbf{E}_W$.

- **Feature Drift:** $\tilde{\mathbf{Z}} = \mathbf{Z}_m + \boldsymbol{\Delta}_Z$. This rotates the left singular vectors: $\tilde{\mathbf{U}}_{\mathbf{Z}} \approx \mathbf{U}_{\mathbf{Z}_m} + \mathbf{E}_Z$.

**Step 3: Alignment Degradation.** The alignment of the aggregated model depends on the interaction term:

$$\tilde{\mathbf{V}}_{\mathbf{W}}^\top \tilde{\mathbf{U}}_{\mathbf{Z}} \approx (\mathbf{V}_{\mathbf{W}_m} + \mathbf{E}_W)^\top (\mathbf{U}_{\mathbf{Z}_m} + \mathbf{E}_Z) \approx \mathbf{I} + \underbrace{\mathbf{V}_{\mathbf{W}_m}^\top \mathbf{E}_Z + \mathbf{E}_W^\top \mathbf{U}_{\mathbf{Z}_m}}_{\text{Misalignment Terms}}. \tag{21}$$

In high-dimensional spaces, the heterogeneity-induced rotation $\mathbf{E}_W$ (from parameter averaging) and the cumulative feature noise $\mathbf{E}_Z$ (from input degradation) are driven by different statistical sources and are generally incoherent. These random rotations deviate the subspaces from their optimal intersection ($\theta \approx 0 \to \theta > 0$). Since $\cos(\theta)$ is a strictly decreasing function for $\theta \in [0, \pi/2]$, any perturbation away from perfect alignment reduces the singular values. Thus, $\mathcal{A}_{agg} < \mathcal{A}_{local}$. $\quad\square$

### G.4 PART III: CUMULATIVE FEATURE DEGRADATION (CFD)

Finally, we prove that the degradations described in Part I and II are not static but accumulate along the network depth.

**Theorem 1** (Cumulative Feature Degradation). *Let $E^\ell = \|\tilde{\mathbf{z}}_m^\ell - \mathbf{z}_m^\ell\|$ be the feature error at layer $\ell$ for client $m$. Assume the spectral norm of aggregated weights is bounded by $\lambda$ ($\|\tilde{\mathbf{W}}^\ell\|_2 \leq \lambda$) and the layer-wise heterogeneity noise is bounded by $\gamma$ ($\|\boldsymbol{\Delta}_m^\ell \mathbf{z}_m^{\ell-1}\| \leq \gamma$). Then, the feature error accumulates exponentially with depth $\ell$:*

$$E^\ell \leq \gamma \frac{\lambda^\ell - 1}{\lambda - 1} \quad \text{(for } \lambda > 1\text{)}. \tag{22}$$

*Proof.* We derive the bound by establishing a recurrence relation for error propagation.

**Step 1: Decomposition of Feature Error.** We compare the aggregated feature $\tilde{\mathbf{z}}^\ell$ with the ideal local feature $\mathbf{z}^\ell$ at layer $\ell$. Using the recursive definitions:

$$\begin{aligned}
\tilde{\mathbf{z}}^\ell - \mathbf{z}^\ell &= \tilde{\mathbf{W}}^\ell \tilde{\mathbf{z}}^{\ell-1} - \mathbf{W}_m^\ell \mathbf{z}^{\ell-1} \\
&= \tilde{\mathbf{W}}^\ell \tilde{\mathbf{z}}^{\ell-1} - (\tilde{\mathbf{W}}^\ell + \boldsymbol{\Delta}_m^\ell) \mathbf{z}^{\ell-1} \\
&= \underbrace{\tilde{\mathbf{W}}^\ell (\tilde{\mathbf{z}}^{\ell-1} - \mathbf{z}^{\ell-1})}_{\text{Propagated Error}} - \underbrace{\boldsymbol{\Delta}_m^\ell \mathbf{z}^{\ell-1}}_{\text{Intrinsic Heterogeneity Noise}}.
\end{aligned} \tag{23}$$

**Step 2: Triangle Inequality.** Taking the norm of both sides and applying the triangle inequality ($\|A + B\| \leq \|A\| + \|B\|$) and sub-multiplicative property ($\|AB\| \leq \|A\|\|B\|$):

$$\begin{aligned}
E^\ell = \|\tilde{\mathbf{z}}^\ell - \mathbf{z}^\ell\| &\leq \|\tilde{\mathbf{W}}^\ell (\tilde{\mathbf{z}}^{\ell-1} - \mathbf{z}^{\ell-1})\| + \| - \boldsymbol{\Delta}_m^\ell \mathbf{z}^{\ell-1}\| \\
&\leq \|\tilde{\mathbf{W}}^\ell\|_2 \cdot \|\tilde{\mathbf{z}}^{\ell-1} - \mathbf{z}^{\ell-1}\| + \|\boldsymbol{\Delta}_m^\ell \mathbf{z}^{\ell-1}\|.
\end{aligned} \tag{24}$$

Substituting the bounds $\|\tilde{\mathbf{W}}^\ell\|_2 \leq \lambda$ and $\|\boldsymbol{\Delta}_m^\ell \mathbf{z}^{\ell-1}\| \leq \gamma$, we obtain the linear recurrence inequality:

$$E^\ell \leq \lambda E^{\ell-1} + \gamma. \tag{25}$$

**Step 3: Unrolling the Recurrence.** We expand the recurrence from layer $\ell$ down to layer 0. Note that at the input layer, the data is identical for both models, so $E^0 = \|\mathbf{x} - \mathbf{x}\| = 0$.

$$\begin{aligned}
E^\ell &\leq \lambda E^{\ell-1} + \gamma \\
&\leq \lambda(\lambda E^{\ell-2} + \gamma) + \gamma = \lambda^2 E^{\ell-2} + \lambda\gamma + \gamma \\
&\cdots \\
&\leq \lambda^\ell E^0 + \gamma \sum_{j=0}^{\ell-1} \lambda^j = \gamma \sum_{j=0}^{\ell-1} \lambda^j.
\end{aligned} \tag{26}$$

For deep neural networks, weights typically maintain signal magnitude ($\lambda \geq 1$) to prevent vanishing gradients. For $\lambda > 1$, the geometric series sums to:

$$E^\ell \leq \gamma \frac{\lambda^\ell - 1}{\lambda - 1}. \tag{27}$$

$\square$

**Synthesis: The Mechanism of Cumulative Feature Degradation.** Combining Theorem 1 with Propositions 1 and 2 provides a holistic view of the failure mode. The feature error $E^\ell$ does not remain constant; it is amplified layer-by-layer due to the recursive nature of the network ($\lambda > 1$). At the **penultimate layer** ($\ell \approx L$), the accumulated error $E^L$ reaches its maximum. This maximized error $\mathcal{E}$ (from Eq. (2)) causes the most significant inflation of within-class variance (Prop. 1) and induces the largest subspace rotation $\mathbf{E}_Z$ (from Eq. (12)), leading to severe misalignment with the final classifier (Prop. 2). This theoretically validates the empirical observation that degradation is progressive and peaks at the deepest layers.

## H   DETAILED RESULTS OF PERFORMANCE DROP IN MODEL AGGREGATION

In this section, we provide more detailed results that demonstrate the performance drop during model aggregation. In these experiments, we perform inference on both the training and testing datasets using the pre-aggregated and post-aggregated models. The experimental results are shown in Figure 18. These experiments are conducted on different datasets, including Digit-Five, PACS, and DomainNet, and on various model architectures. From Figure 18, we can observe that the performance drop introduced by model aggregation is consistent across all adopted datasets and model architectures, on both training and testing dataset. The performance drop consistently occurs throughout the entire training procedure of FL. These results indicate that the performance drop during model aggregation is a common phenomenon in FL.

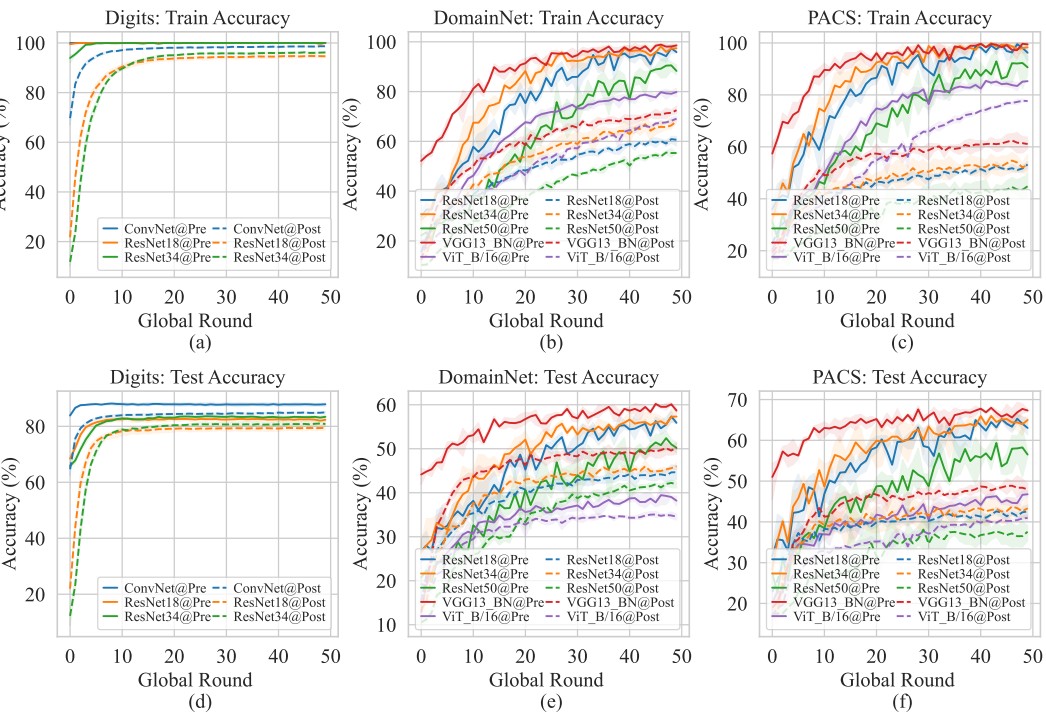

Figure 18: Training and testing accuracy curves of the model before and after aggregation, evaluated on the local dataset during FL training. The experiments are conducted on Digit-Five, DomainNet, and PACS, using multiple model architectures as the backbone.

# I  DETAILED RESULTS OF FEATURE VARIANCE

In this section, we provide a detailed analysis of the layer-wise feature variance during FL training. Our experiments are conducted on three datasets: Digit-Five, PACS, and DomainNet, using various model architectures, as previously described. For each dataset, we presents four types of feature variances: normalized within-class feature variance, normalized between-class feature variance, original unnormalized within-class feature variance, and original unnormalized between-class feature variance.

To better visualize feature evolution over time (across different epochs of FL training) and space (across different layers), we employ two types of visualizations. The first visualizes feature changes across different layers while keeping the training global round fixed. The second focuses on visualizing feature evolution across training rounds while fixing specific layers.

The experimental results are presented in the following sections. From these results, we observe that both the original within-class and between-class feature variances increase as the layer depth increases. In contrast, the normalized within-class feature variance decreases with both layer depth and training rounds, which is in contrast to the normalized between-class feature variance. This suggests that features within the same class become more compressed, while features across different classes become more discriminative.

However, after model aggregation, the normalized within-class variance increases while the normalized between-class variance decreases. This indicates that model aggregation disrupts the feature compression objective during DNN training. More importantly, this disruption progressively accumulates across model layers, causing the features in the penultimate layer (which are used for final decision-making) to degrade more significantly.

## I.1  CHANGES OF FEATURE VARIANCE ACROSS LAYERS

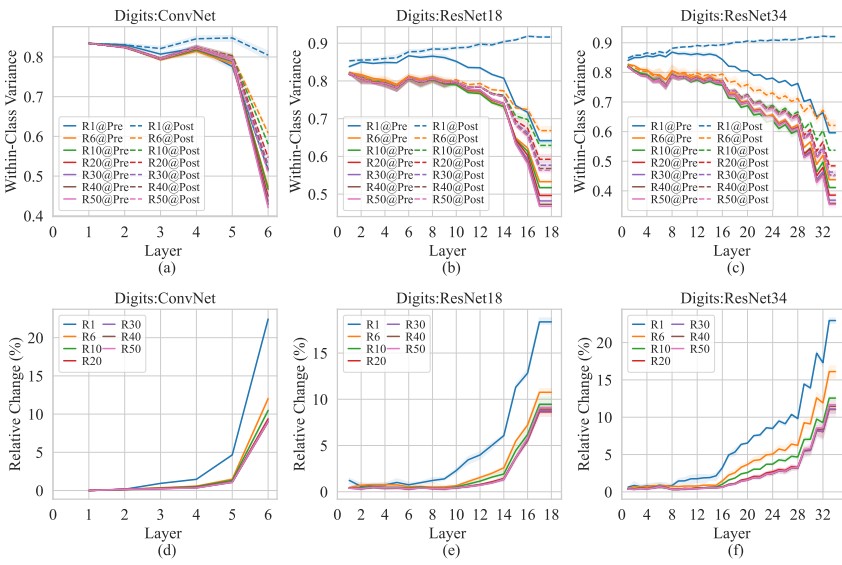

Figure 19: Changes in the normalized within-class variance of features across model layers for specific global rounds, with larger X-axis values indicating deeper layers. The model is trained on Digit-Five with multiple models that are randomly initialized. The top half of the figure shows the normalized within-class variance, while the bottom half displays the relative change in variance before and after model aggregation.

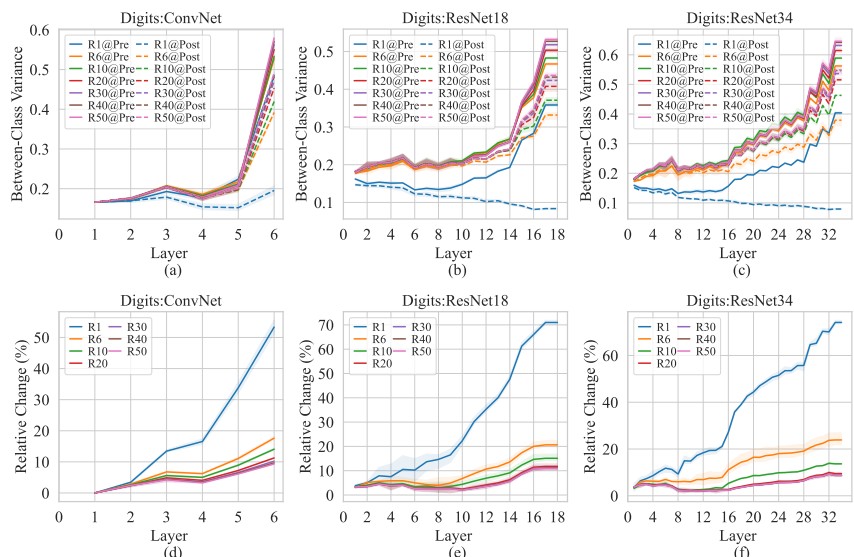

Figure 20: Changes in the normalized between-class variance of features across model layers for specific global rounds, with larger X-axis values indicating deeper layers. The model is trained on Digit-Five with multiple models that are randomly initialized. The top half of the figure shows the normalized between-class variance, while the bottom half displays the relative change in variance before and after model aggregation.

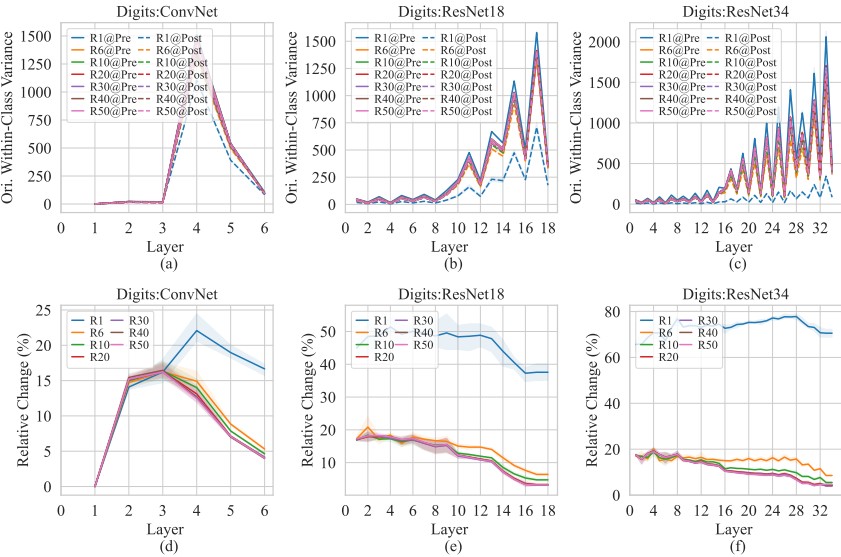

Figure 21: Changes in the original unnormalized within-class variance of features across model layers for specific global rounds, with larger X-axis values indicating deeper layers. The model is trained on Digit-Five with multiple models that are randomly initialized. The top half of the figure shows the original unnormalized within-class variance, while the bottom half displays the relative change in variance before and after model aggregation.

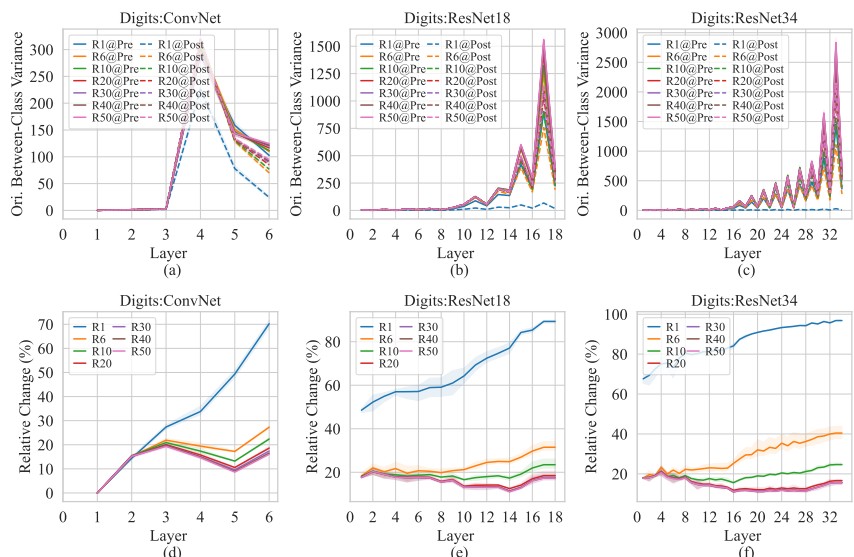

Figure 22: Changes in the original unnormalized between-class variance of features across model layers for specific global rounds, with larger X-axis values indicating deeper layers. The model is trained on Digit-Five with multiple models that are randomly initialized. The top half of the figure shows the original unnormalized between-class variance, while the bottom half displays the relative change in variance before and after model aggregation.

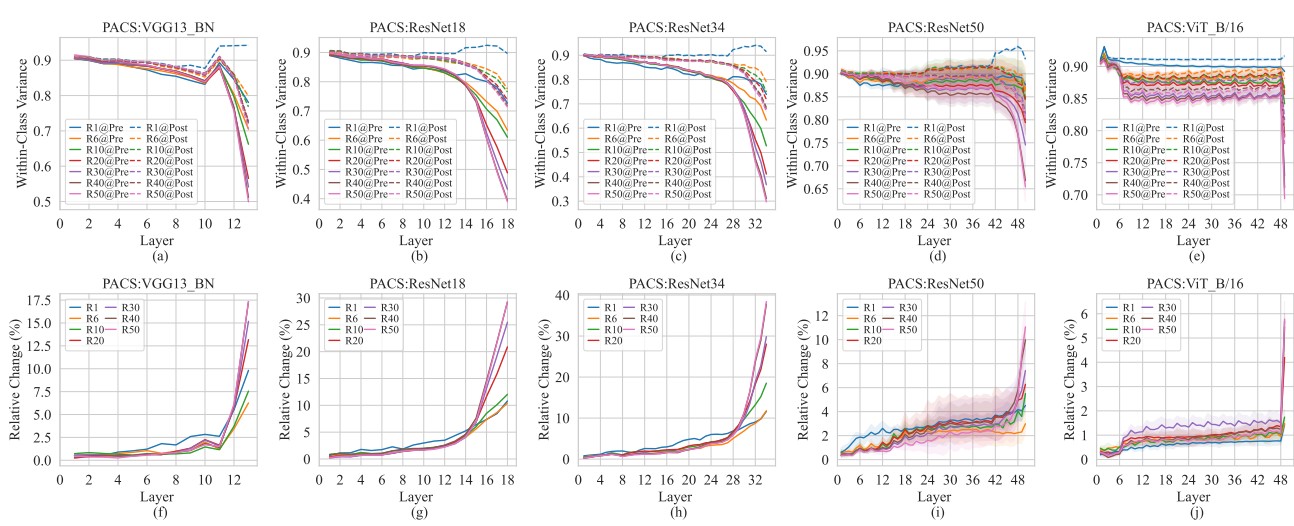

Figure 23: Changes in the normalized within-class variance of features across model layers for specific global rounds, with larger X-axis values indicating deeper layers. The model is trained on PACS with multiple models that are randomly initialized. The top half of the figure shows the normalized within-class variance, while the bottom half displays the relative change in variance before and after model aggregation.

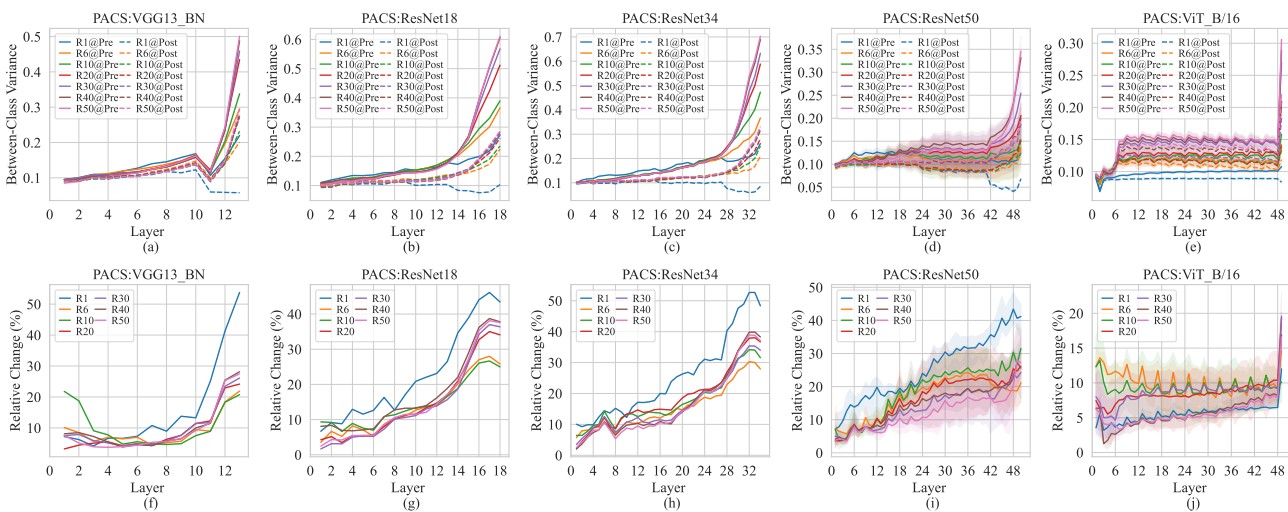

Figure 24: Changes in the normalized between-class variance of features across model layers for specific global rounds, with larger X-axis values indicating deeper layers. The model is trained on PACS with multiple models that are randomly initialized. The top half of the figure shows the normalized between-class variance, while the bottom half displays the relative change in variance before and after model aggregation.

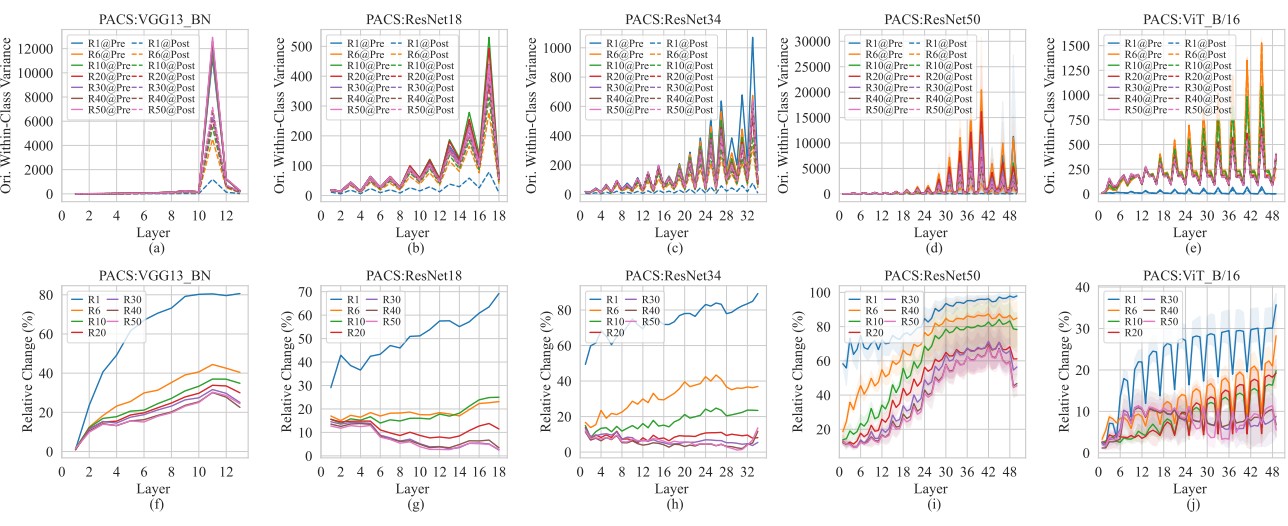

Figure 25: Changes in the original unnormalized within-class variance of features across model layers for specific global rounds, with larger X-axis values indicating deeper layers. The model is trained on PACS with multiple models that are randomly initialized. The top half of the figure shows the original unnormalized within-class variance, while the bottom half displays the relative change in variance before and after model aggregation.

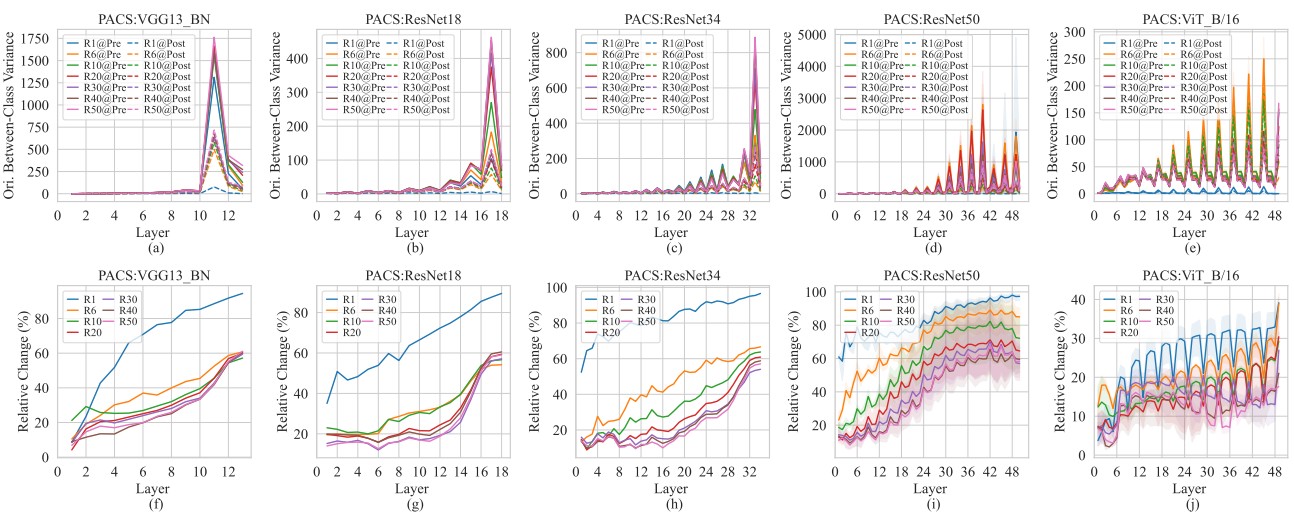

Figure 26: Changes in the original unnormalized between-class variance of features across model layers for specific global rounds, with larger X-axis values indicating deeper layers. The model is trained on PACS with multiple models that are randomly initialized. The top half of the figure shows the original unnormalized between-class variance, while the bottom half displays the relative change in variance before and after model aggregation.

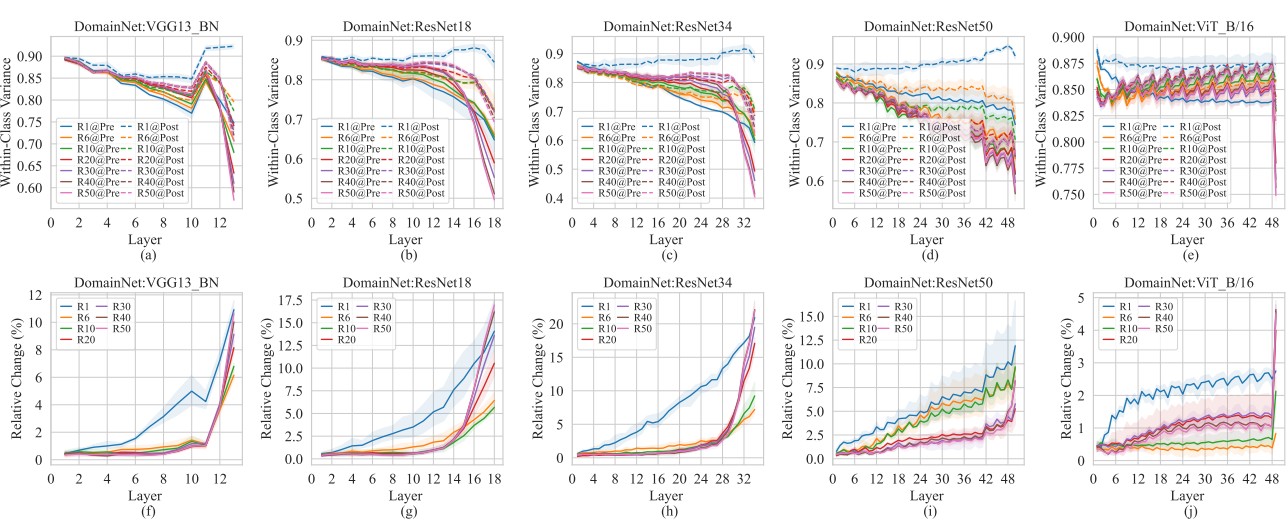

Figure 27: Changes in the normalized within-class variance of features across model layers for specific global rounds, with larger X-axis values indicating deeper layers. The model is trained on DomainNet with multiple models that are randomly initialized. The top half of the figure shows the normalized within-class variance, while the bottom half displays the relative change in variance before and after model aggregation.

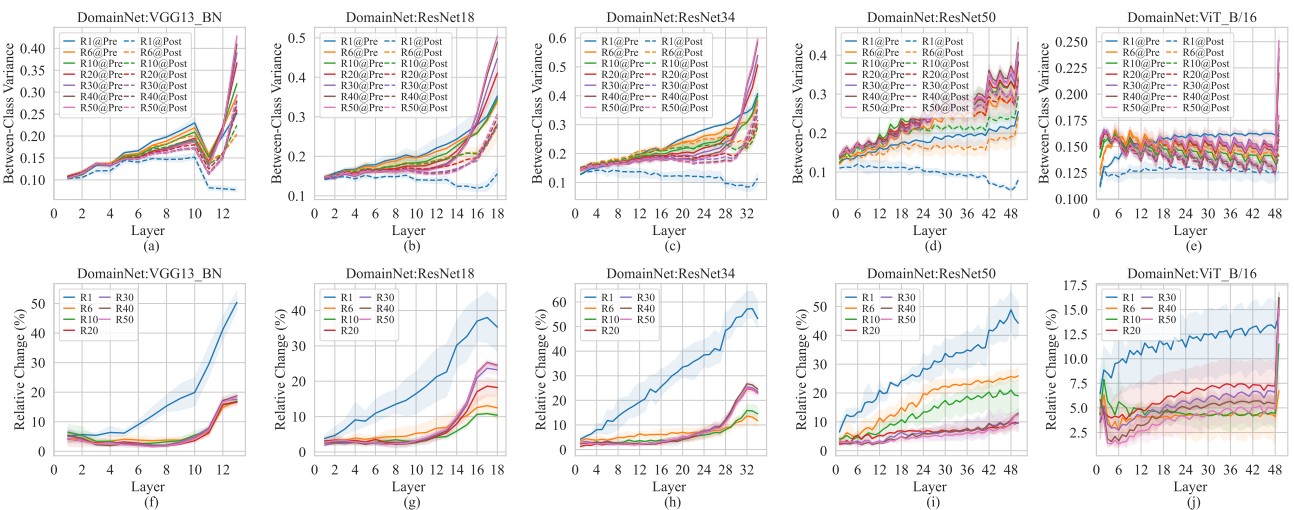

Figure 28: Changes in the normalized between-class variance of features across model layers for specific global rounds, with larger X-axis values indicating deeper layers. The model is trained on DomainNet with multiple models that are randomly initialized. The top half of the figure shows the normalized between-class variance, while the bottom half displays the relative change in variance before and after model aggregation.

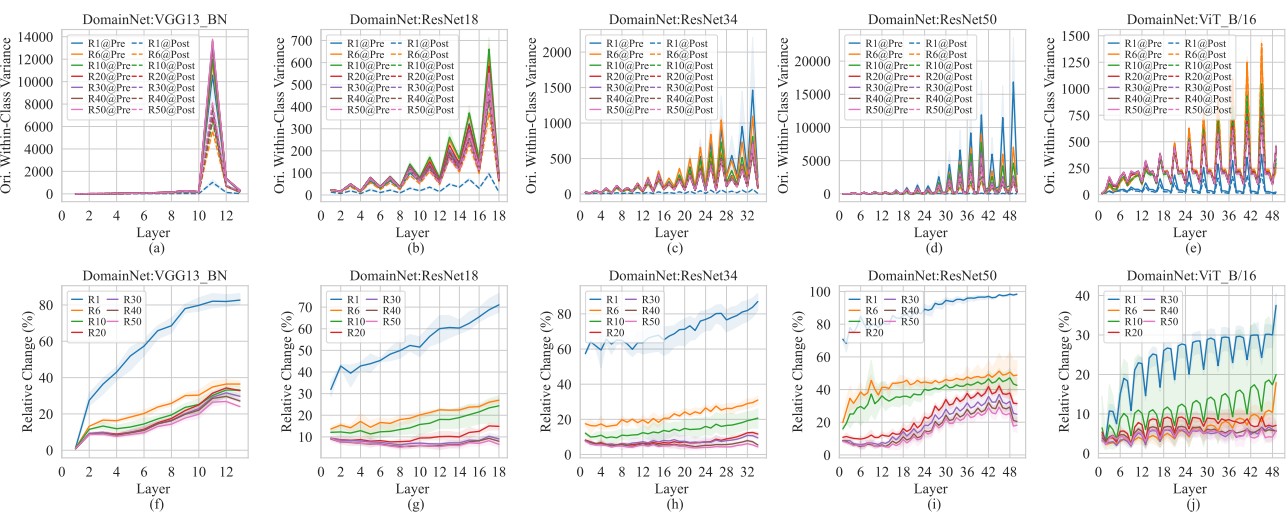

Figure 29: Changes in the unnormalized within-class variance of features across model layers for specific global rounds, with larger X-axis values indicating deeper layers. The model is trained on DomainNet with multiple models that are randomly initialized. The top half of the figure shows the original unnormalized within-class variance, while the bottom half displays the relative change in variance before and after model aggregation.

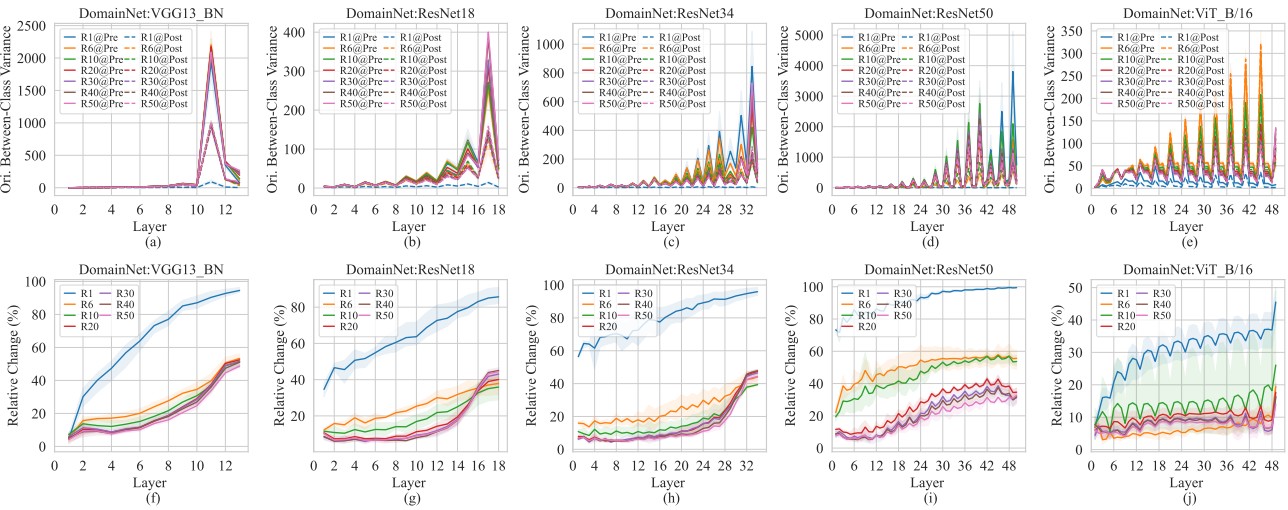

Figure 30: Changes in the unnormalized between-class variance of features across model layers for specific global rounds, with larger X-axis values indicating deeper layers. The model is trained on DomainNet with multiple models that are randomly initialized. The top half of the figure shows the original unnormalized between-class variance, while the bottom half displays the relative change in variance before and after model aggregation.

## I.2 CHANGES OF FEATURE VARIANCE ACROSS TRAINING ROUNDS

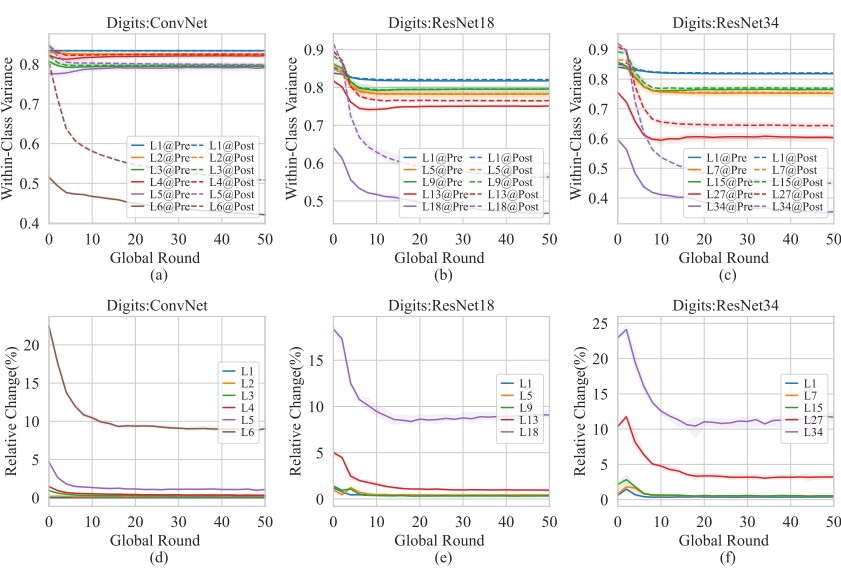

Figure 31: Changes in the normalized within-class variance of features across FL training at specific model layers. The model is trained on Digit-Five with multiple models that are randomly initialized. The top half of the figure shows the normalized within-class variance, while the bottom half displays the relative change in variance before and after model aggregation.

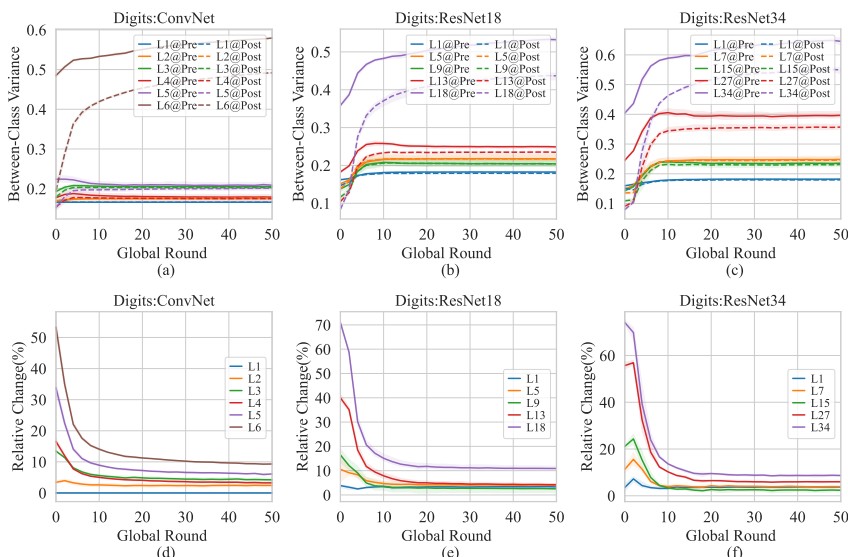

Figure 32: Changes in the normalized between-class variance of features across FL training at specific model layers. The model is trained on Digit-Five with multiple models that are randomly initialized. The top half of the figure shows the normalized between-class variance, while the bottom half displays the relative change in variance before and after model aggregation.

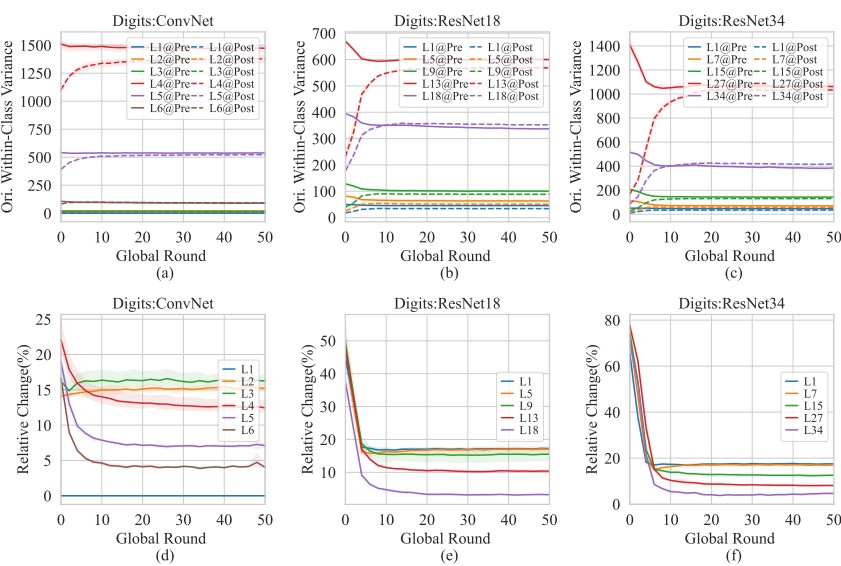

Figure 33: Changes in the original unnormalized within-class variance of features across FL training at specific model layers. The model is trained on Digit-Five with multiple models that are randomly initialized. The top half of the figure shows the original unnormalized within-class variance, while the bottom half displays the relative change in variance before and after model aggregation.

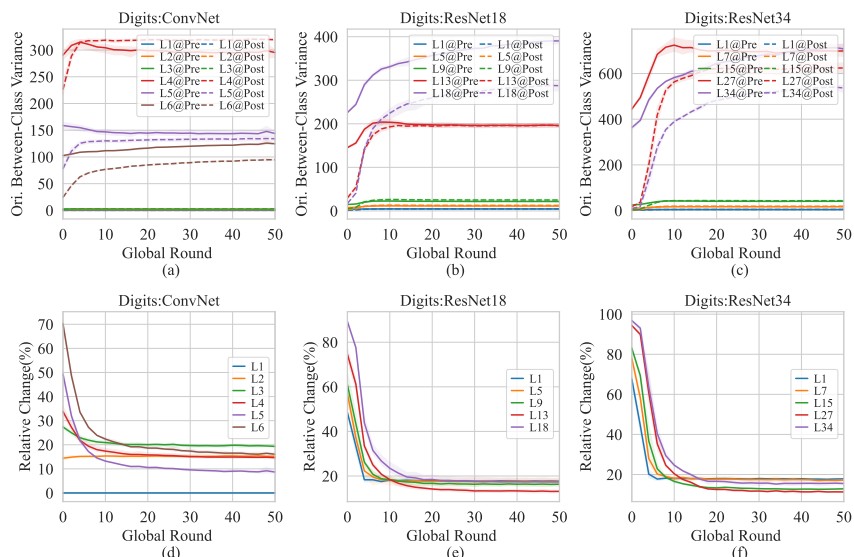

Figure 34: Changes in the original unnormalized between-class variance of features across FL training at specific model layers. The model is trained on Digit-Five with multiple models that are randomly initialized. The top half of the figure shows the original unnormalized between-class variance, while the bottom half displays the relative change in variance before and after model aggregation.

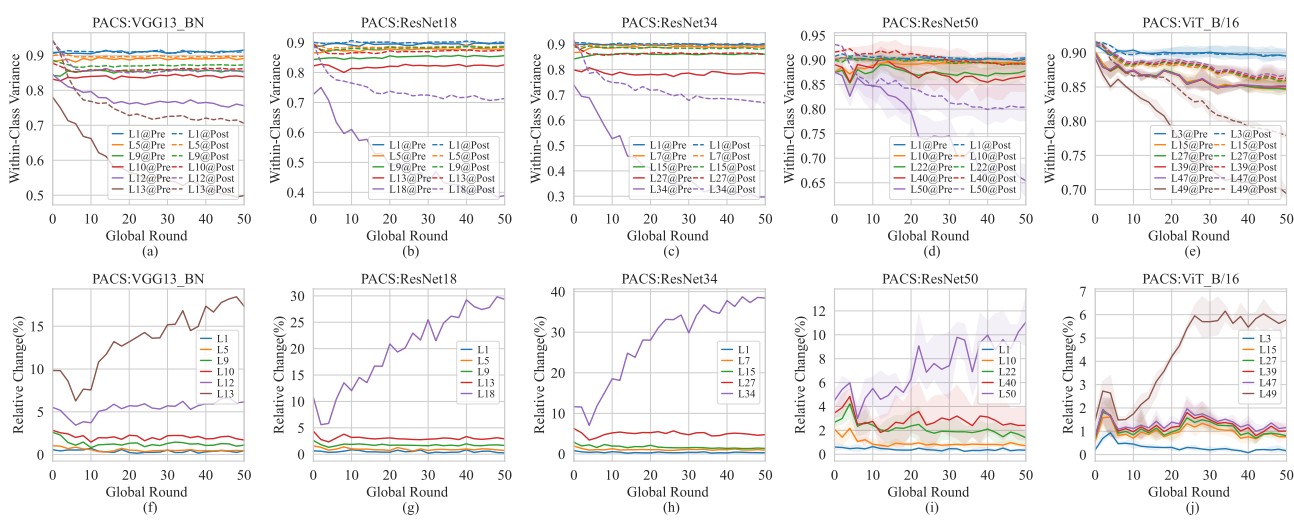

Figure 35: Changes in the normalized within-class variance of features across FL training at specific model layers. The model is trained on PACS with multiple models that are randomly initialized. The top half of the figure shows the normalized within-class variance, while the bottom half displays the relative change in variance before and after model aggregation.

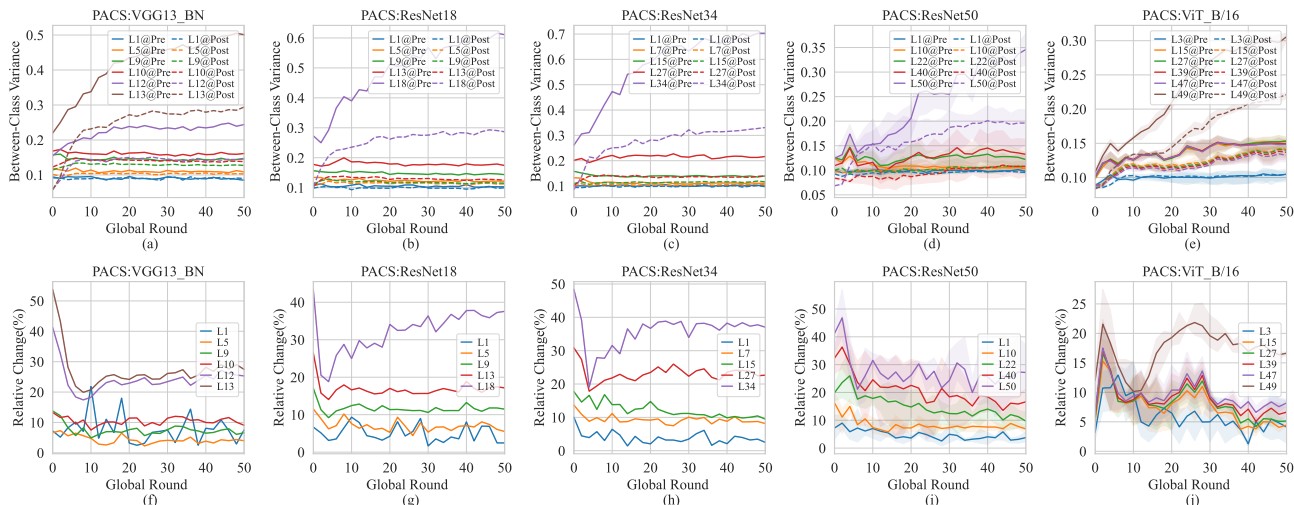

Figure 36: Changes in the normalized between-class variance of features across FL training at specific model layers. The model is trained on PACS with multiple models that are randomly initialized. The top half of the figure shows the normalized between-class variance, while the bottom half displays the relative change in variance before and after model aggregation.

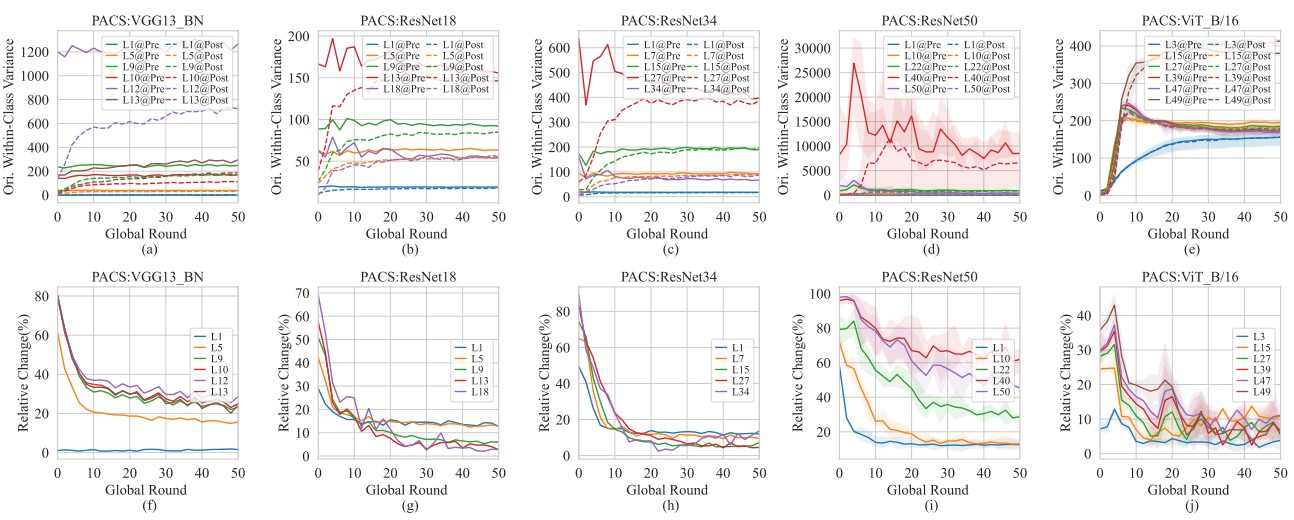

Figure 37: Changes in the original unnormalized within-class variance of features across FL training at specific model layers. The model is trained on PACS with multiple models that are randomly initialized. The top half of the figure shows the original unnormalized within-class variance, while the bottom half displays the relative change in variance before and after model aggregation.

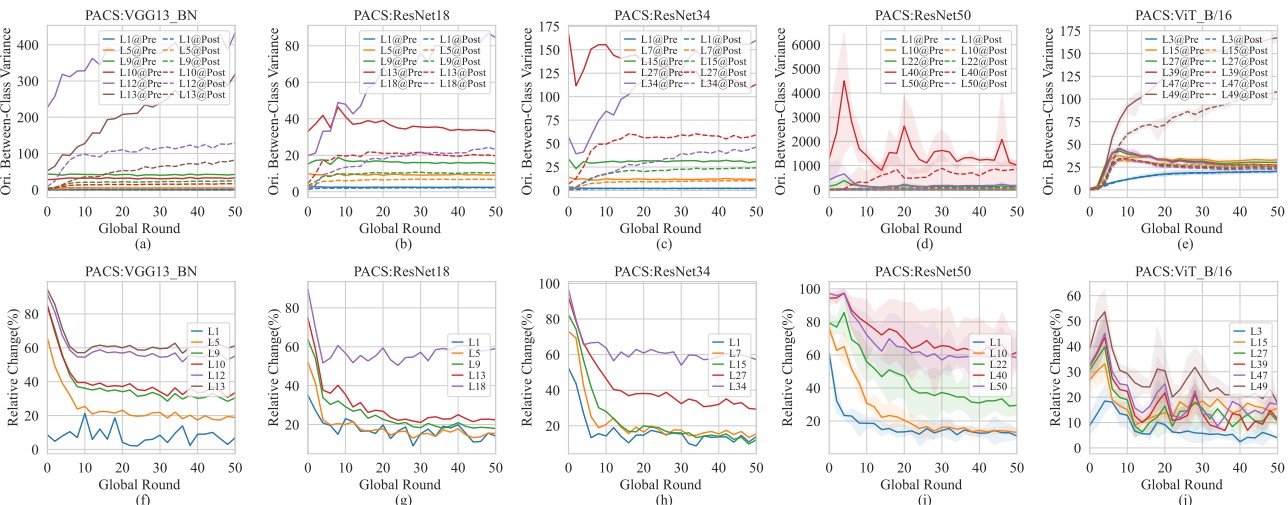

Figure 38: Changes in the original unnormalized between-class variance of features across FL training at specific model layers. The model is trained on PACS with multiple models that are randomly initialized. The top half of the figure shows the original unnormalized between-class variance, while the bottom half displays the relative change in variance before and after model aggregation.

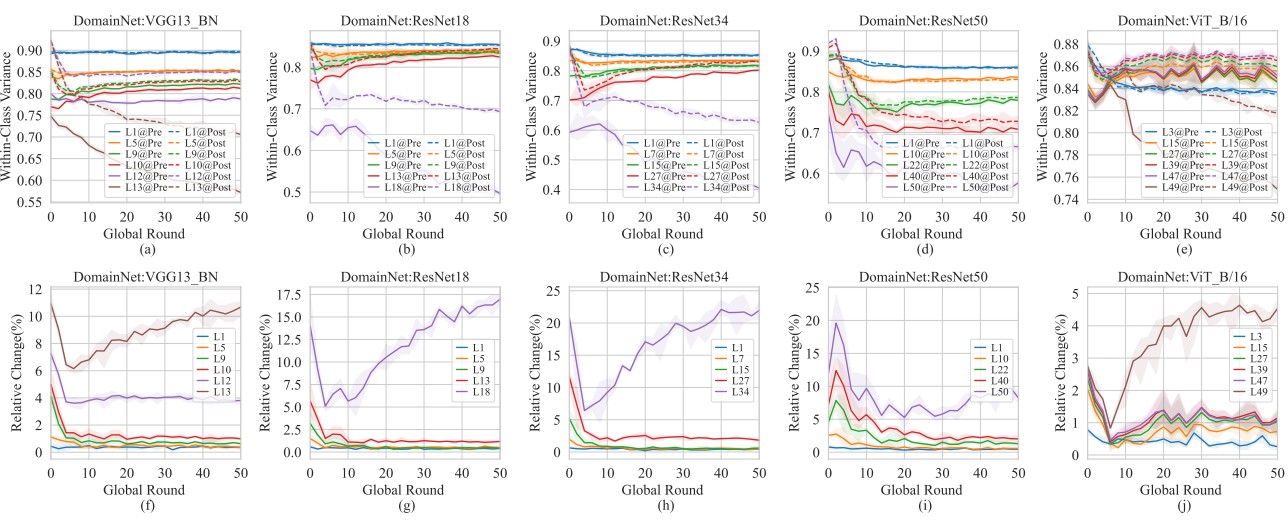

Figure 39: Changes in the normalized within-class variance of features across FL training at specific model layers. The model is trained on DomainNet with multiple models that are randomly initialized. The top half of the figure shows the normalized within-class variance, while the bottom half displays the relative change in variance before and after model aggregation.

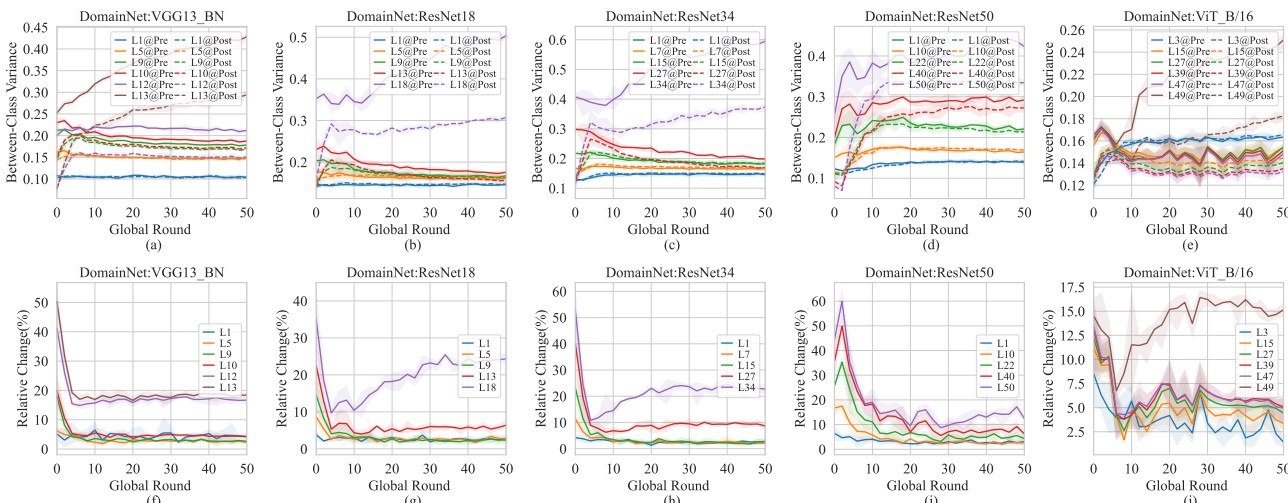

Figure 40: Changes in the normalized between-class variance of features across FL training at specific model layers. The model is trained on DomainNet with multiple models that are randomly initialized. The top half of the figure shows the normalized between-class variance, while the bottom half displays the relative change in variance before and after model aggregation.

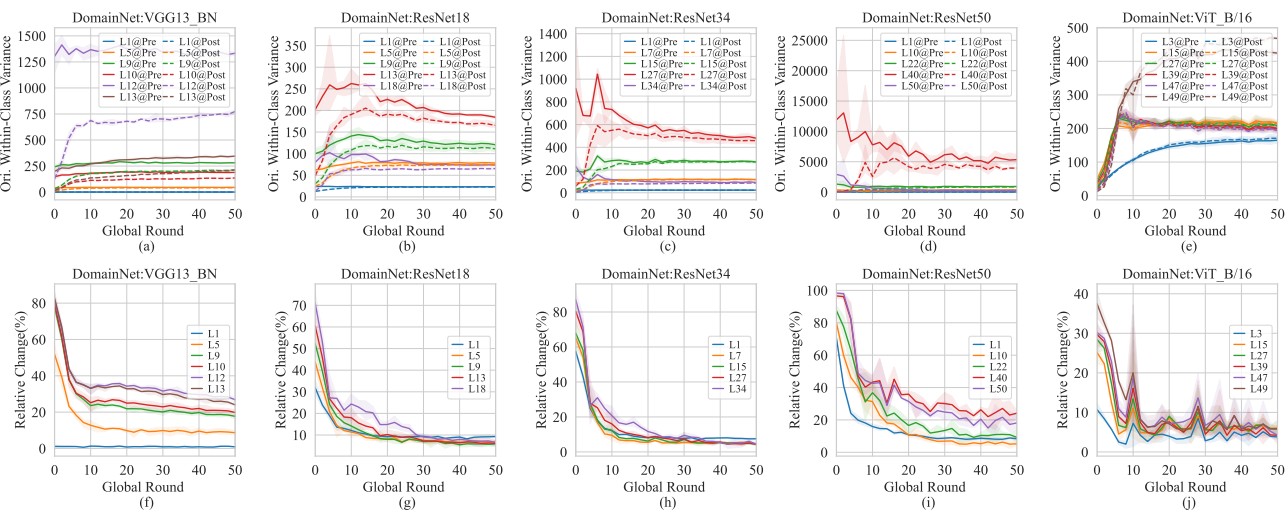

Figure 41: Changes in the original unnormalized within-class variance of features across FL training at specific model layers. The model is trained on DomainNet with multiple models that are randomly initialized. The top half of the figure shows the original unnormalized within-class variance, while the bottom half displays the relative change in variance before and after model aggregation.

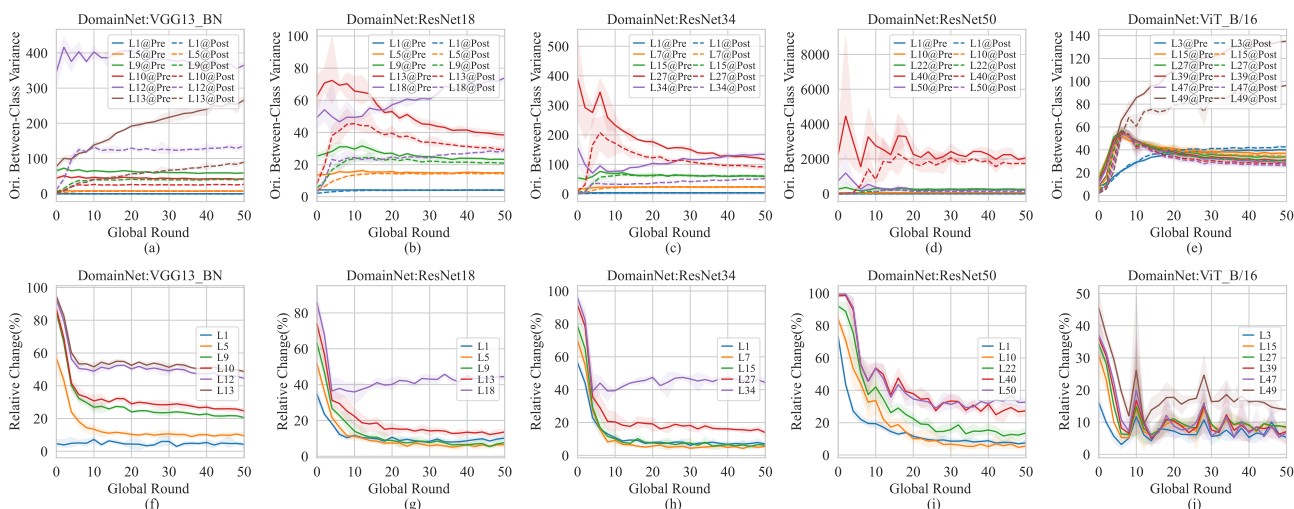

Figure 42: Changes in the original unnormalized between-class variance of features across FL training at specific model layers. The model is trained on DomainNet with multiple models that are randomly initialized. The top half of the figure shows the original unnormalized between-class variance, while the bottom half displays the relative change in variance before and after model aggregation.

## J    DETAILED RESULTS OF ALIGNMENT BETWEEN FEATURES AND PARAMETERS

In this section, we present the experimental results corresponding to the alignment of features and the parameters of the subsequent layers. From these results, we observe that the alignment between features and parameters improves as training progresses, with the alignment of penultimate layer features and the classifier increasing more rapidly.

After model aggregation, the alignment between features and parameters tends to decrease. However, the decrease is more pronounced in the classifier. This increased mismatch between penultimate layer features and the classifier, along with the degradation of the penultimate layer features, causes the aggregated model to perform significantly worse when sent back to each client.

### J.1    CHANGES OF ALIGNMENT ACROSS LAYERS

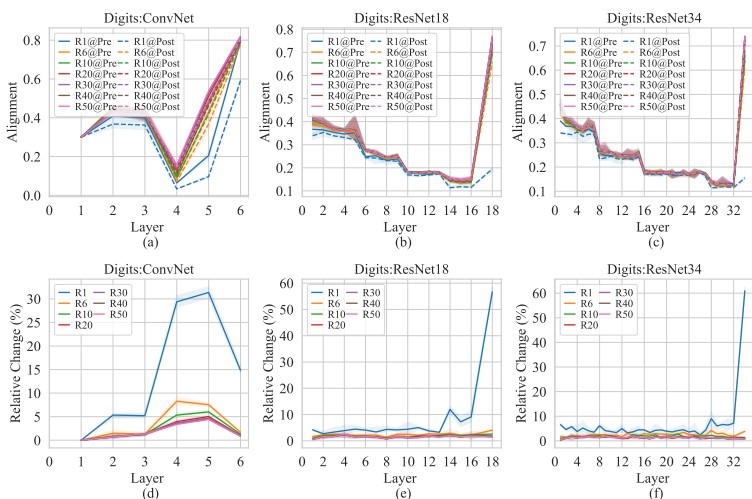

Figure 43: Changes in the alignment between features and parameters across model layers for specific global rounds, with larger X-axis values indicating deeper layers. The model is trained on Digit-Five with multiple models that are randomly initialized. The top half of the figure shows the original alignment values between features and parameters, while the bottom half displays the relative change in alignment before and after model aggregation.

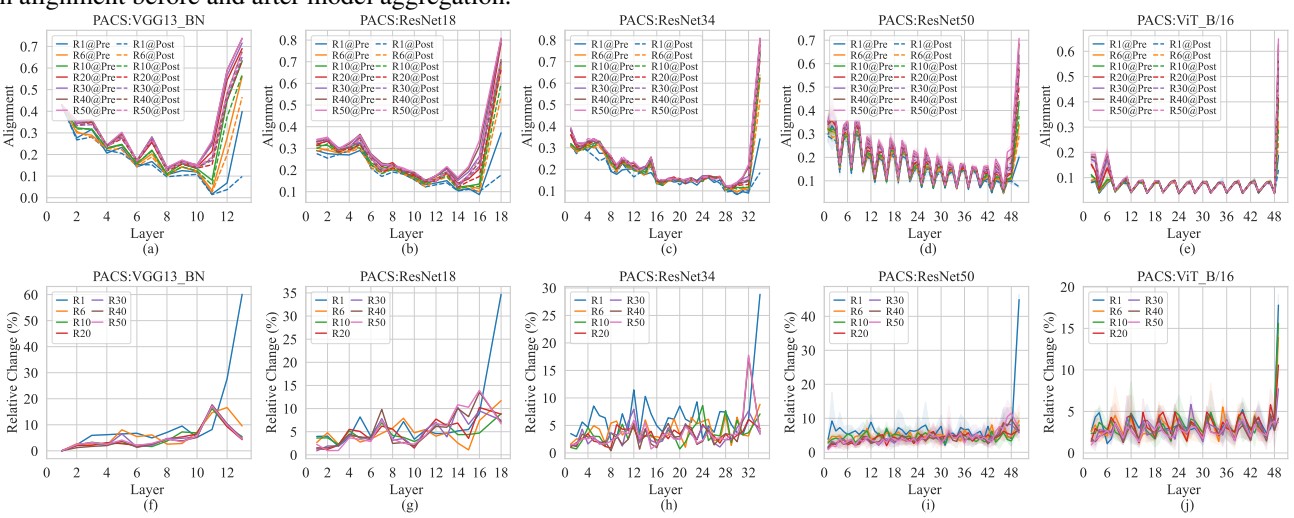

Figure 44: Changes in the alignment between features and parameters across model layers for specific global rounds, with larger X-axis values indicating deeper layers. The model is trained on PACS with multiple models that are randomly initialized. The top half of the figure shows the original alignment values between features and parameters, while the bottom half displays the relative change in alignment before and after model aggregation.

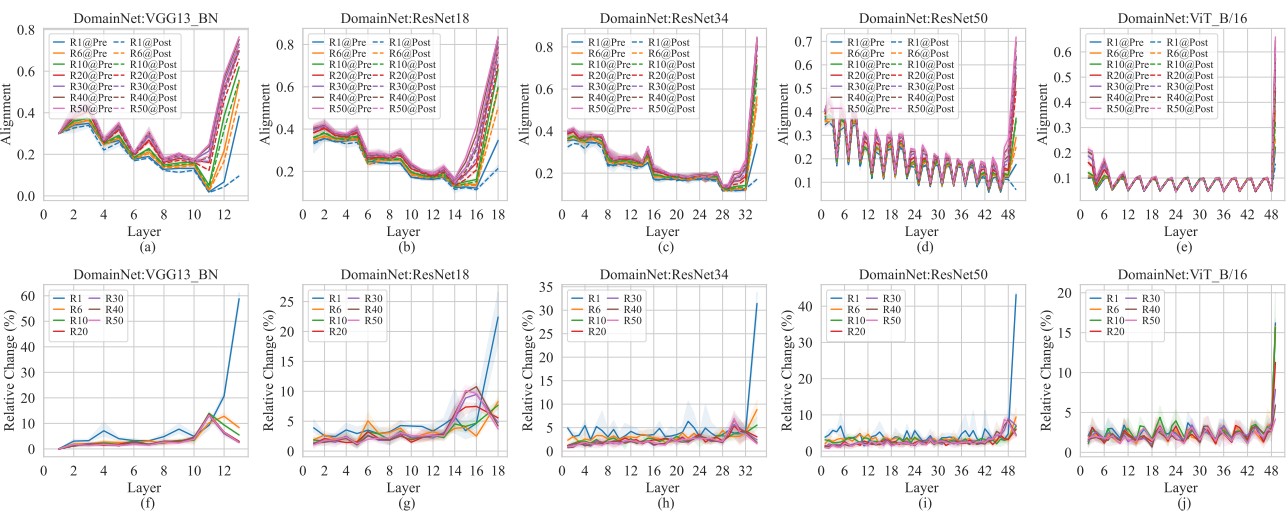

Figure 45: Changes in the alignment between features and parameters across model layers for specific global rounds, with larger X-axis values indicating deeper layers. The model is trained on DomainNet with multiple models that are randomly initialized. The top half of the figure shows the original alignment values between features and parameters, while the bottom half displays the relative change in alignment before and after model aggregation.

## J.2 CHANGES OF ALIGNMENT ACROSS TRAINING ROUNDS

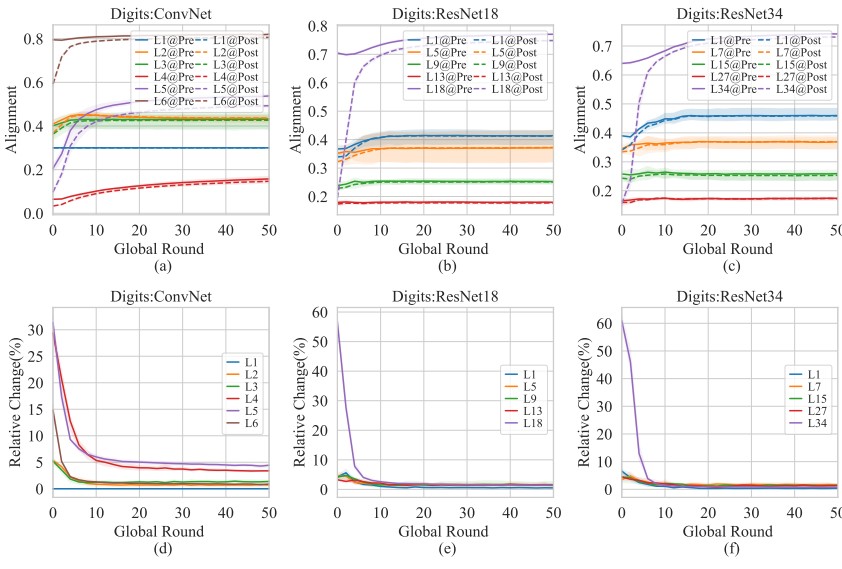

Figure 46: Changes in the alignment between features and parameters across FL training at specific model layers. The model is trained on Digit-Five with multiple models that are randomly initialized. The top half of the figure shows the original alignment values between features and parameters, while the bottom half displays the relative change in alignment before and after model aggregation.

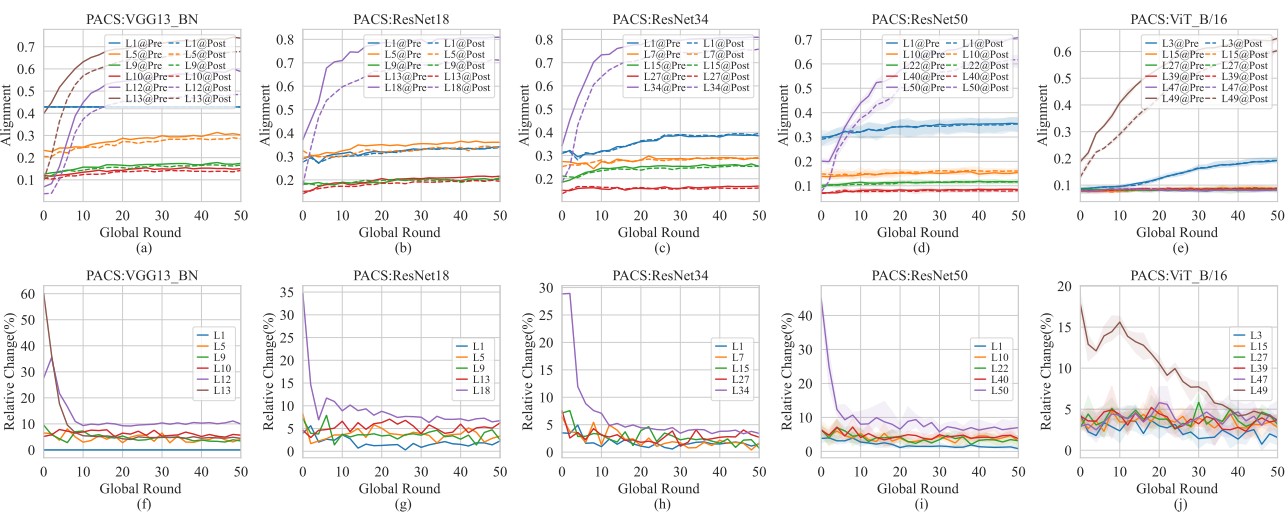

Figure 47: Changes in the alignment between features and parameters across FL training at specific model layers. The model is trained on PACS with multiple models that are randomly initialized. The top half of the figure shows the original alignment values between features and parameters, while the bottom half displays the relative change in alignment before and after model aggregation.

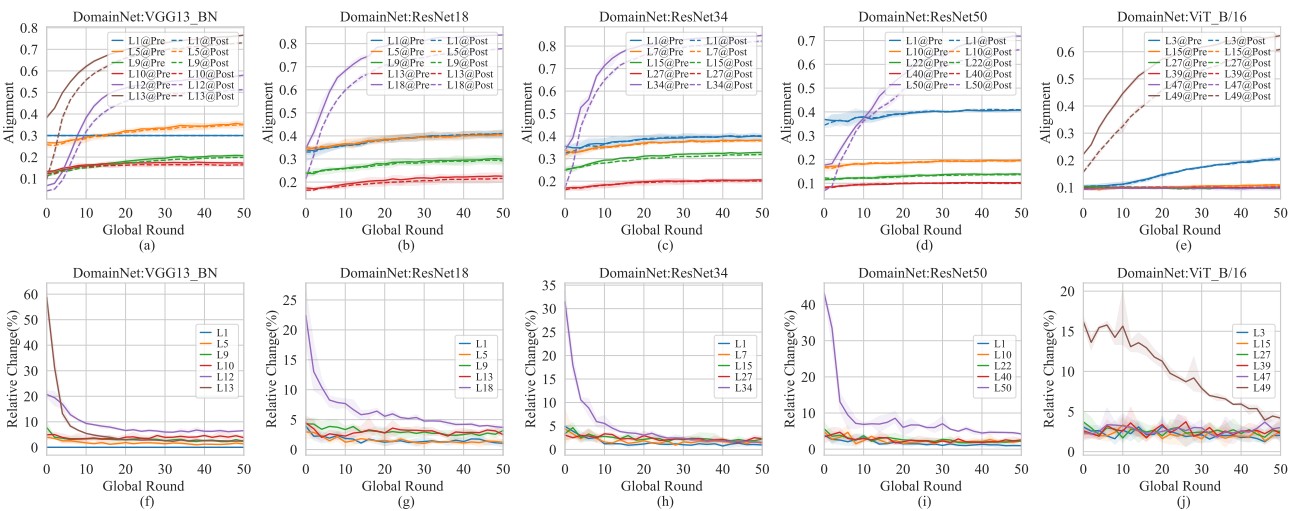

Figure 48: Changes in the alignment between features and parameters across FL training at specific model layers. The model is trained on DomainNet with multiple models that are randomly initialized. The top half of the figure shows the original alignment values between features and parameters, while the bottom half displays the relative change in alignment before and after model aggregation.

# K    VISUALIZATION OF PRE-AGGREGATED AND POST-AGGREGATED FEATURES

This section visualizes the comparison between pre-aggregated and post-aggregated features. It can be observed that, as layer depth increases, features become more compressed within the same class and more discriminative across different classes. However, after model aggregation, features become more scattered within the same class and less discriminative across classes. This phenomenon is

particularly pronounced in the penultimate layer features (the leftmost subfigure). These observations are consistent with the findings from the quantitative metrics described earlier.

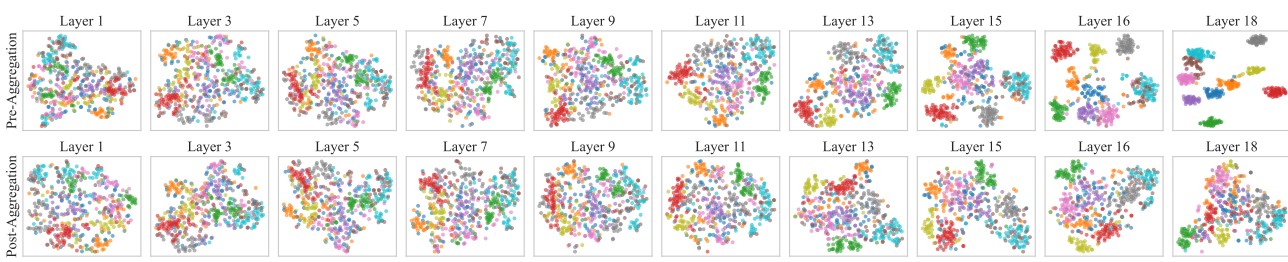

Figure 49: T-SNE visualization of features at different layers on the 'Quickdraw' domain of Domain-Net before and after aggregation. The features are extracted from ResNet18 in the final global round of FL training, whose parameters are randomly initialized at the beginning.

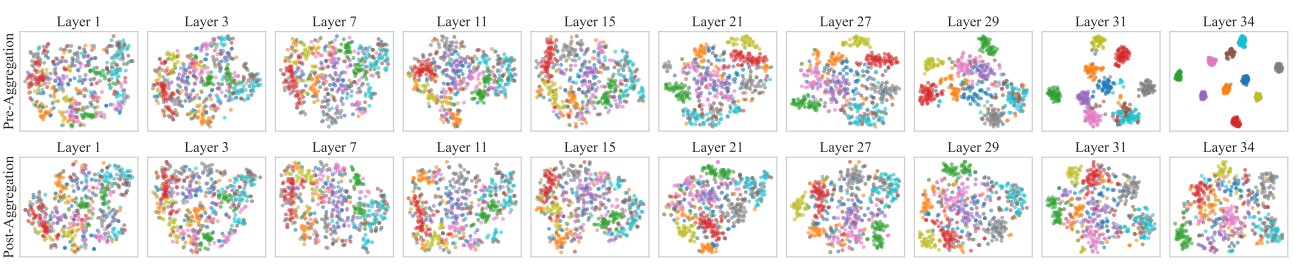

Figure 50: T-SNE visualization of features at different layers on the 'Quickdraw' domain of Domain-Net before and after aggregation. The features are extracted from ResNet34 in the final global round of FL training, whose parameters are randomly initialized at the beginning.

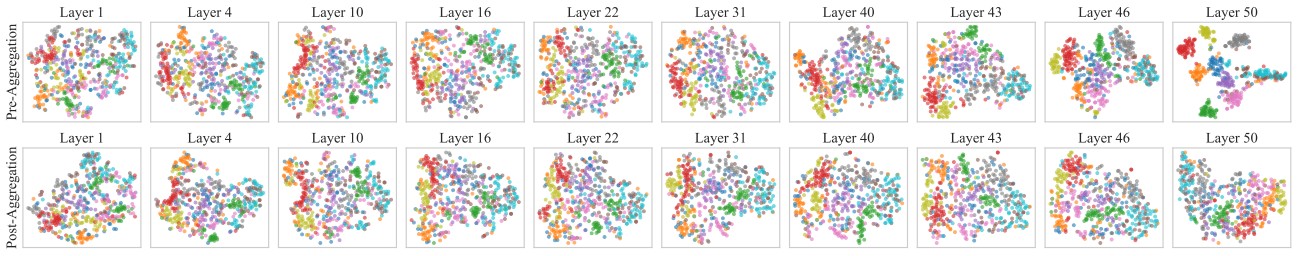

Figure 51: T-SNE visualization of features at different layers on the 'Quickdraw' domain of Domain-Net before and after aggregation. The features are extracted from ResNet50 in the final global round of FL training, whose parameters are randomly initialized at the beginning.

## L    FEATURE DISTANCE AND PARAMETER DISTANCE

This section compares the distance between features and parameters in the pre-aggregated and post-aggregated models at each layer. It can be observed that, with the exception of the final classifier, the parameter distance between the pre-aggregated and post-aggregated models is significantly smaller than the feature distance. Furthermore, the distances between parameters and features show different trends across layers. While the parameter distance decreases, the feature distance increases as the

layer depth increases. This demonstrates that even small variations in the parameter space can lead to significant feature variations, as the stacked structure of DNNs tends to magnify errors in feature extraction. This observation suggests that the 'client drift' proposed by previous studies may not be the root cause of performance drops during model aggregation.

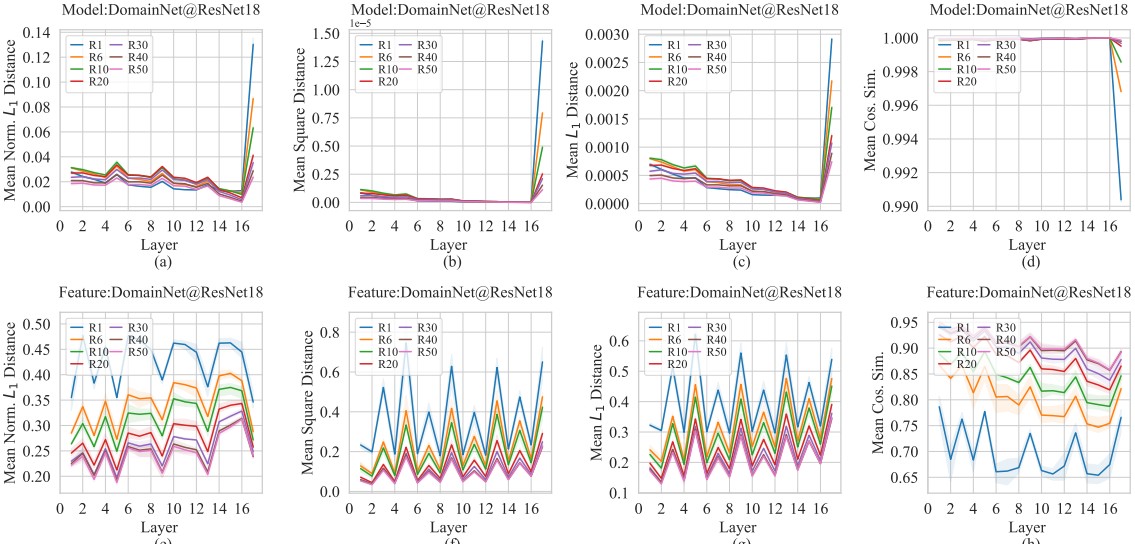

Figure 52: Changes in distance of the features and parameters obtained from models before and after aggregation across model layers for specific global rounds, with larger X-axis values indicating deeper layers. The model is trained on DomainNet using ResNet18. The distance is measured by 4 metric including mean normalized distance, mean squared distance, mean $L_1$ distance, and mean cosine similarity. The top half of the figure shows the distance between the parameters of pre-aggregated and post-aggregated models, while the bottom half displays the distance between the features of pre-aggregated and post-aggregated models.

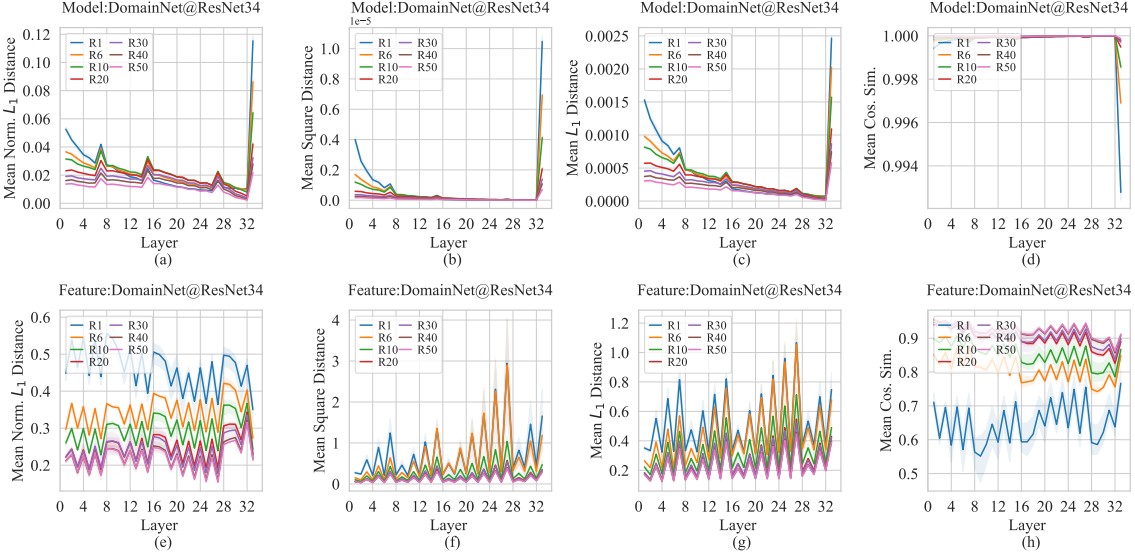

Figure 53: Changes in distance of the features and parameters obtained from models before and after aggregation across model layers for specific global rounds, with larger X-axis values indicating deeper layers. The model is trained on DomainNet using ResNet18. The distance is measured by 4 metric including mean normalized distance, mean squared distance, mean $L_1$ distance, and mean cosine similarity. The top half of the figure shows the distance between the parameters of pre-aggregated and post-aggregated models, while the bottom half displays the distance between the features of pre-aggregated and post-aggregated models.

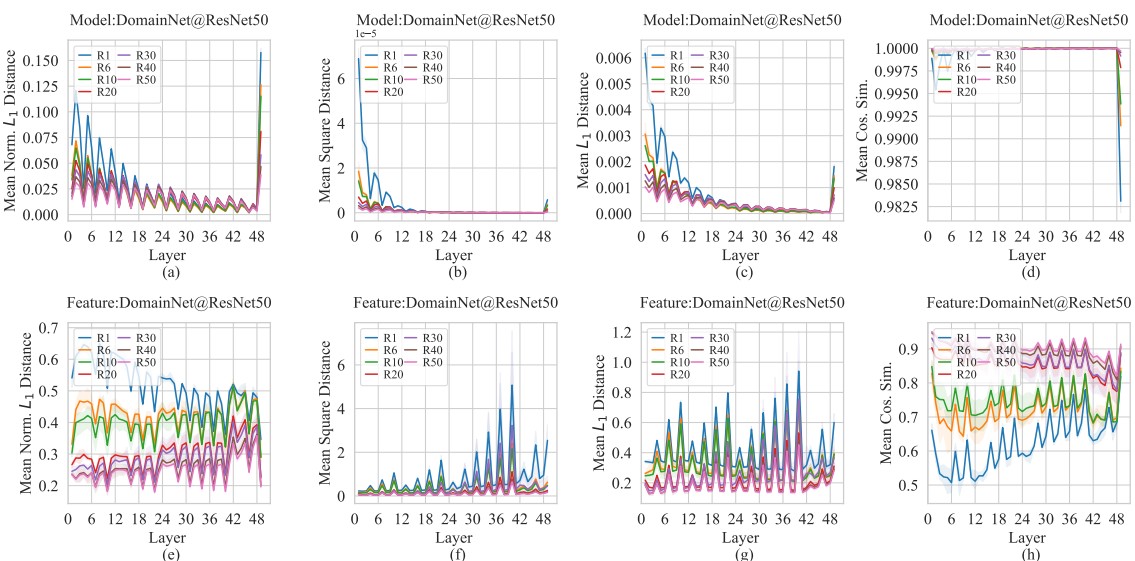

Figure 54: Changes in distance of the features and parameters obtained from models before and after aggregation across model layers for specific global rounds, with larger X-axis values indicating deeper layers. The model is trained on DomainNet using ResNet18. The distance is measured by 4 metric including mean normalized distance, mean squared distance, mean $L_1$ distance, and mean cosine similarity. The top half of the figure shows the distance between the parameters of pre-aggregated and post-aggregated models, while the bottom half displays the distance between the features of pre-aggregated and post-aggregated models.

## M    DETAILED RESULTS OF LINEAR PROBING

This section evaluates the accuracy of linear probing across diverse data distributions. In our experiments, we use SGD with a learning rate of 0.01 to optimize the randomly initialized linear layer. The batch size is set to 64, and the total number of epochs is set to 100. We report the best test accuracy as the accuracy of linear probing.

From the experimental results, we observe that the features generated by the post-aggregated model perform well across diverse data distributions. In contrast, the pre-aggregated model only performs well on its local data distribution. This suggests that, while the model after aggregation may not extract task-specific features for local distributions, it is more capable of extracting generalized features that can be applied to different distributions. This is also how model aggregation improves performance compared to purely local training.

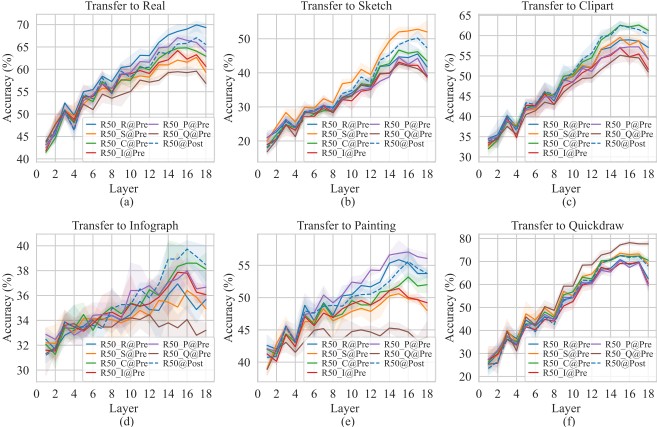

Figure 55: Accuracy of linear probing at different layers across different domains in DomainNet. The experiments are performed in global round 50 using ResNet18 as the backbone model. In the figures, R50@Post represents the results using the model after aggregation, while R50_*@Pre represents the results using the model before aggregation for domain *.

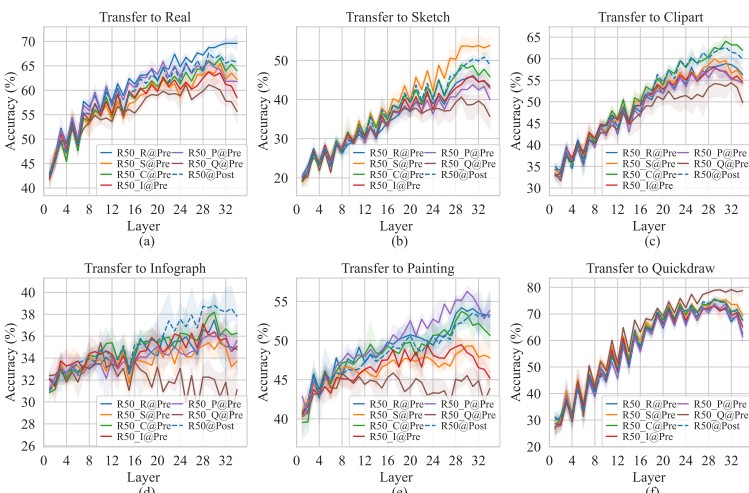

Figure 56: Accuracy of linear probing at different layers across different domains in DomainNet. The experiments are performed in global round 50 using ResNet34 as the backbone model. In the figures, R50@Post represents the results using the model after aggregation, while R50_*@Pre represents the results using the model before aggregation for domain *.

# N    EFFECT OF RESIDUAL CONNECTION

Motivated by the observed zigzag pattern linked to residual blocks, this section investigates the role of residual connections in model aggregation. We use ResNet18 and ResNet34 as backbones and remove the residual connections from these models. We then employ these modified architectures for FL training. This section presents a comparison of accuracy and feature metrics between the models with and without residual connections. We observe that the relative change in feature metrics is more pronounced when the residual connections are removed, particularly during the early stages of model training and in the normalized within-class and between-class feature variance metrics. This may be because residual connections help mitigate feature degradation by allowing less disrupted features in lower layers to be propagated to deeper layers.

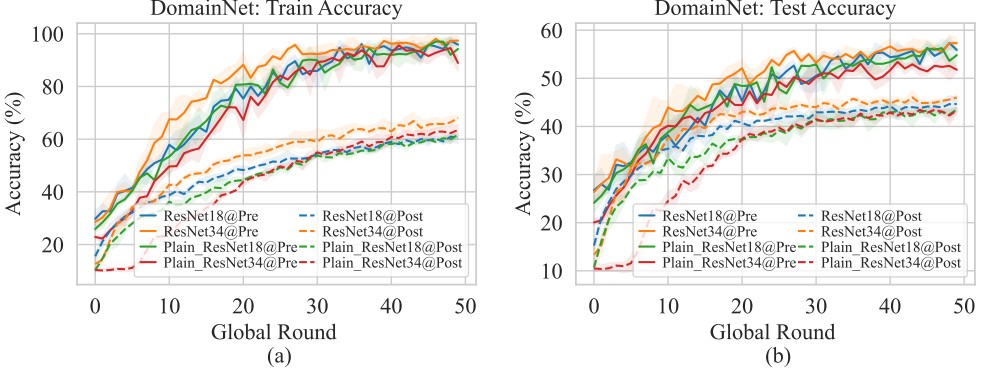

Figure 57: Averaged training and testing accuracy curves of the model before and after aggregation, evaluated on the local dataset during FL training. The experiments are conducted on the DomainNet dataset. The models trained include the original ResNet18 and ResNet, as well as their plain versions, with residual connections removed.

## O    DETAILED RESULTS WHEN PERSONALIZING SPECIFIC PARAMETERS

This section presents the detailed experimental results obtained by personalizing specific layers within the model during FL training, a method known as personalized federated learning (PFL). The experiments are conducted using the ResNet18 and ResNet34 models. As shown in Figure 58, both models can be divided into several blocks: the first convolutional layer, stages 1-4 (which consist of stacked convolutional layers), and the fully connected (FC) layer used for classification.

In the experiments, we explore PFL methods by personalizing specific layers within the model. Specifically, we first examine two commonly used PFL methods: FedPer (Arivazhagan et al., 2019), which personalizes the classifier, and FedBN (Li et al., 2021b), which personalizes the batch normalization (BN) layers. Additionally, motivated by (Sun et al., 2021), we consider two more strategies for parameter personalization: the successive parameter personalization strategy and the skip parameter personalization strategy. For the successive parameter personalization strategy, we personalize multiple consecutive layers starting from the first layer. In the skip parameter personalization strategy, we randomly select a layer or block for personalization.

The experimental results show that these parameter personalization methods generally lead to better featured distributions on local data. Moreover, the relative changes introduced by model aggregation decrease as more parameters within the feature extractor are personalized. This improvement is due to the reduction in feature disruption accumulation, which is alleviated by personalizing these parameters, thereby maintaining their ability to extract locally task-specific features without being affected by model aggregation.

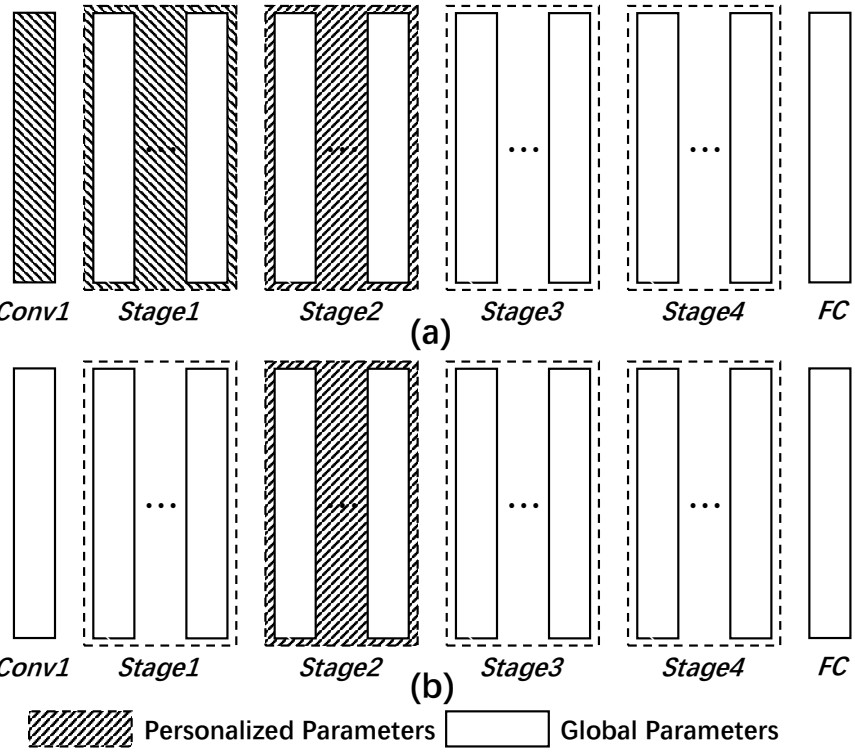

Figure 58: Illustration of the architectures of ResNet18 and ResNet34, along with the parameter personalization strategies: (a) Successive parameter personalization strategy, (b) Skip parameter personalization strategy.

## O.1 SUCCESSIVE PARAMETER PERSONALIZATION

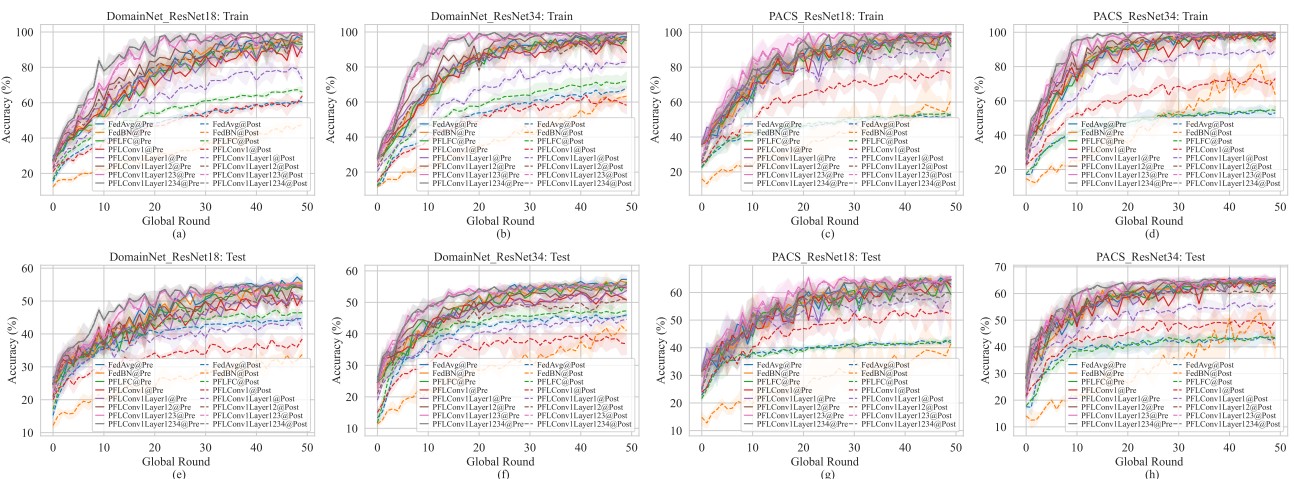

Figure 59: Accuracy curves when training FL models with different successive parameter personalization strategies, along with FedAvg, FedPer, and FedBN. In the legend, 'C1' is an abbreviation for the Conv1 layer, and 'S*' represents the abbreviation for Stage* block.

## O.2 SKIP PARAMETER PERSONALIZATION

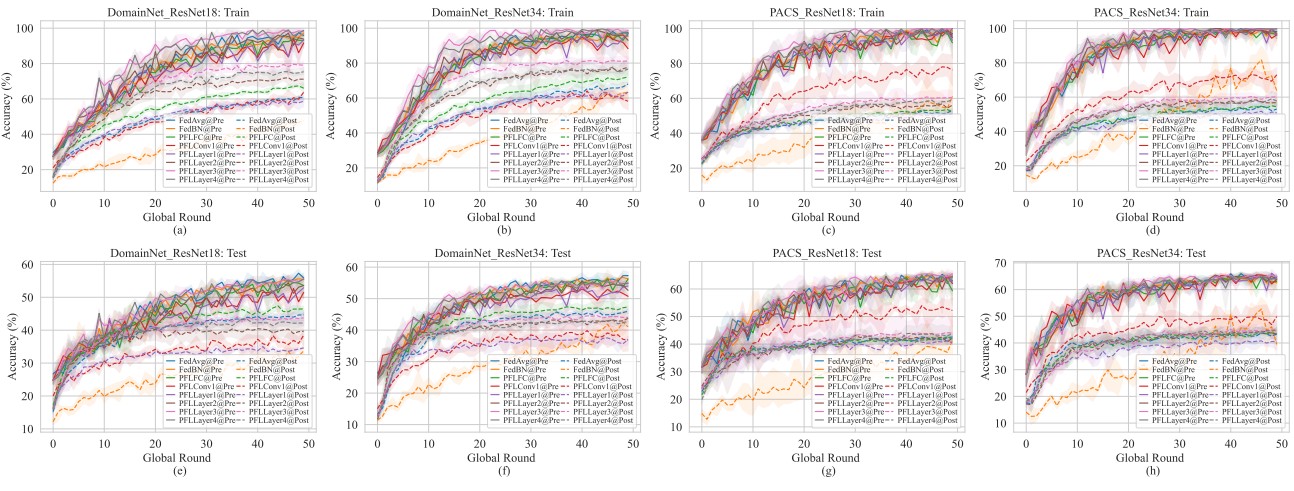

Figure 60: Accuracy curves when training FL models with different skip parameter personalization strategies, along with FedAvg, FedPer, and FedBN. In the legend, 'C1' is an abbreviation for the Conv1 layer, and 'S*' represents the abbreviation for Stage* block.

## P EFFECT OF LOCAL UPDATING EPOCHS

The local update epoch is a key configuration in federated learning that determines the aggregation frequency. In this section, we conduct experiments to investigate its impact on model aggregation. During experimets, we keep the total number of local updates fixed and vary the number of local epochs. The experimental results are presented in Figure 61. To ensure a fair comparison, we maintain a consistent total number of updating epochs across comparisons. The results show that increasing the number of local epochs will compress the within-class features more effectively. However, when

the models are aggregated, the relative change for larger local update epochs is noticeably greater than for smaller ones. This highlights the sensitivity of the model aggregation process to the number of local updating epochs.

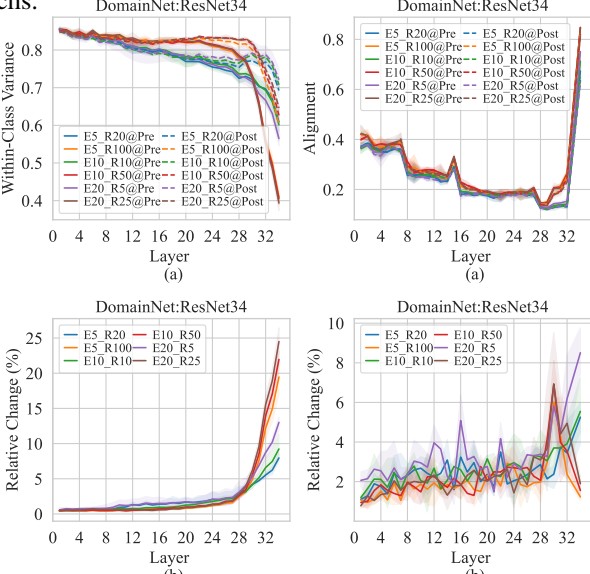

Figure 61: Changes in the normalized within-class feature variance and the alignment between features and parameters when training the FL model with different local update epochs. The model is trained on DomainNet using ResNet34 as backbone. We present two groups of experiments, keeping the total number of updating epochs ($E \times R$) fixed at 100 and 500.

## Q  DETAILED RESULTS WHEN FINE-TUNING CLASSIFIER

This section provides detailed experimental results on fine-tuning the classifier to reduce the mismatch between the features extracted by the global feature extractor and the classifier. The experiments are conducted on DomainNet using ResNet18 and ResNet34 as backbones. During fine-tuning, we use SGD with momentum as the optimizer, with a learning rate of 0.01 and momentum set to 0.1. We perform only 10 epochs of fine-tuning.

It can be observed that, after fine-tuning the classifier, the alignment between the features and the classifier consistently improves using models at different global rounds during the FL training. As a result, the testing accuracy also improves.

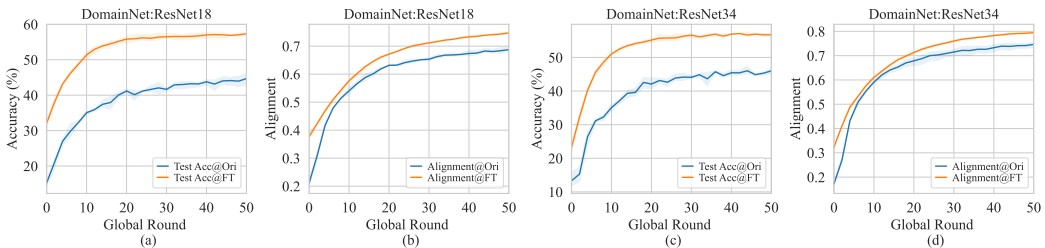

Figure 62: Accuracy and alignment between features and parameters during classifier fine-tuning across global rounds. The experiments are conducted on DomainNet dataset, using ResNet18 and ResNet34 as backbone model.

2700
2701
2702
2703
2704
2705
2706
2707
2708
2709
2710
2711
2712
2713
2714
2715
2716
2717
2718
2719
2720
2721
2722
2723
2724
2725
2726
2727
2728
2729
2730
2731
2732
2733
2734
2735
2736
2737
2738
2739
2740
2741
2742
2743
2744
2745
2746
2747
2748
2749
2750
2751
2752
2753

# R    DETAILED RESULTS WHEN INITIALIZING MODEL WITH PRE-TRAINED PARAMETERS

In this section, we present detailed experimental results from using pre-trained parameters as initialization for FL training. These experiments follow the same setup as the ones described above, with the only difference being that the model is initialized with parameters pre-trained on large-scale datasets. The experimental results include the changes in feature variance across layers and training rounds, the alignment between features and parameters across layers and rounds, and the visualization of pre-aggregated and post-aggregated features. From these experiments, we find that initializing with pre-trained parameters effectively mitigates the accumulation of feature degradation. This conclusion is based on observations from experiments with randomly initialized parameters, where deeper layers only begin to converge once shallow features have reached a specific state. This is primarily because the degradation of unconverged features in the shallow layers propagates to the deeper layers, preventing them from converging until the shallow layers are well-trained. This significantly hinders the convergence rate of FL training. However, when initialized with pre-trained parameters, we find that the features of most shallow layers converge early in training, as the model already possesses strong feature extraction capabilities from being trained on a large-scale dataset. This significantly reduces the feature degradation accumulation introduced by model aggregation, thereby stabilizing and accelerating the convergence of FL training.

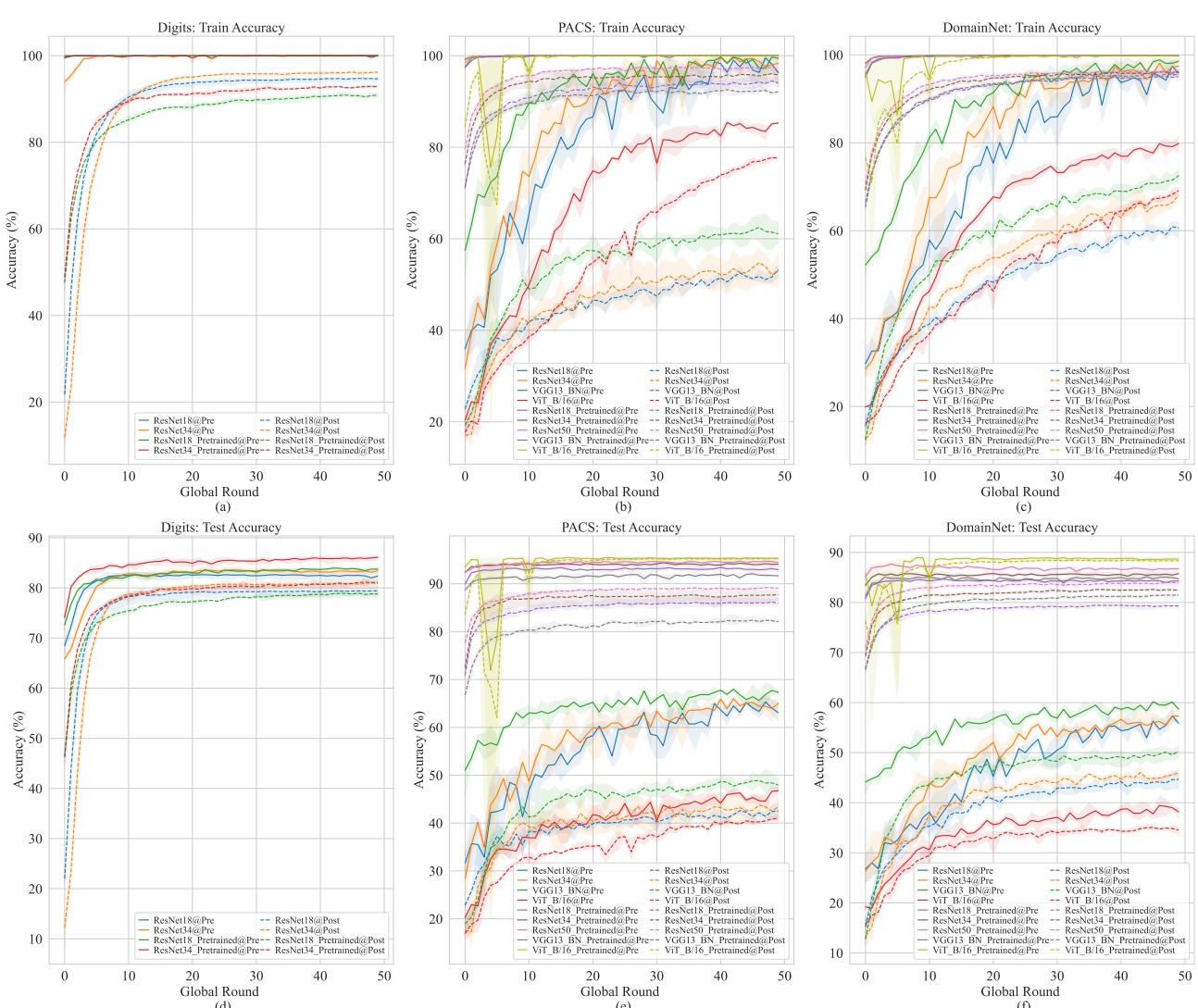

Figure 63: Comparison of accuracy with and without pre-trained parameters as initialization. The performance drop can be significantly mitigated by using pre-trained parameters.

R.2   CHANGES OF FEATURE VARIANCE ACROSS LAYERS

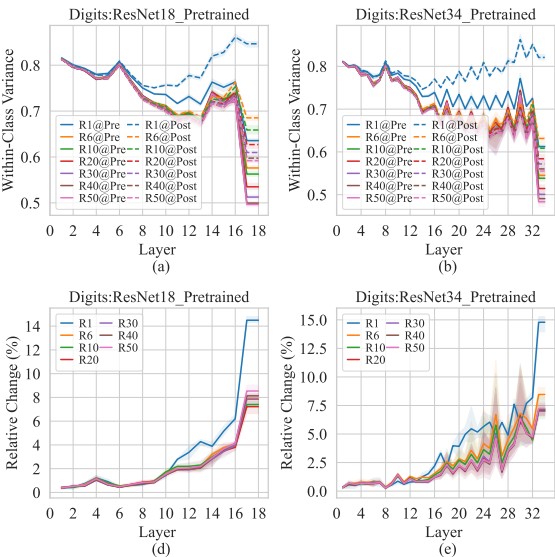

Figure 64: Changes in the normalized within-class variance of features across model layers for specific global rounds, with larger X-axis values indicating deeper layers. The model is trained on Digit-Five with multiple models that are initialized by parameters pre-trained on large-scaled datasets. The top half of the figure shows the normalized within-class variance, while the bottom half displays the relative change in variance before and after model aggregation.

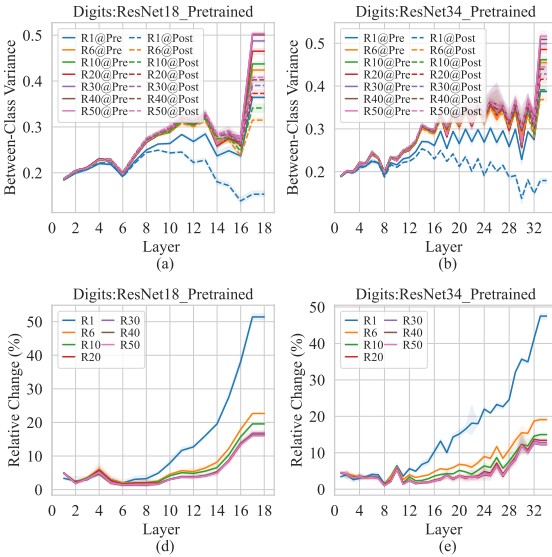

Figure 65: Changes in the normalized between-class variance of features across model layers for specific global rounds, with larger X-axis values indicating deeper layers. The model is trained on Digit-Five with multiple models that are initialized by parameters pre-trained on large-scaled datasets. The top half of the figure shows the normalized between-class variance, while the bottom half displays the relative change in variance before and after model aggregation.

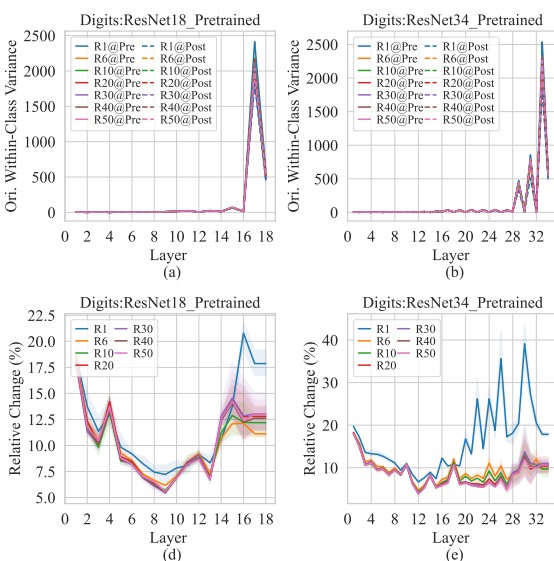

Figure 66: Changes in the original unnormalized within-class variance of features across model layers for specific global rounds, with larger X-axis values indicating deeper layers. The model is trained on Digit-Five with multiple models that are initialized by parameters pre-trained on large-scaled datasets. The top half of the figure shows the original unnormalized within-class variance, while the bottom half displays the relative change in variance before and after model aggregation.

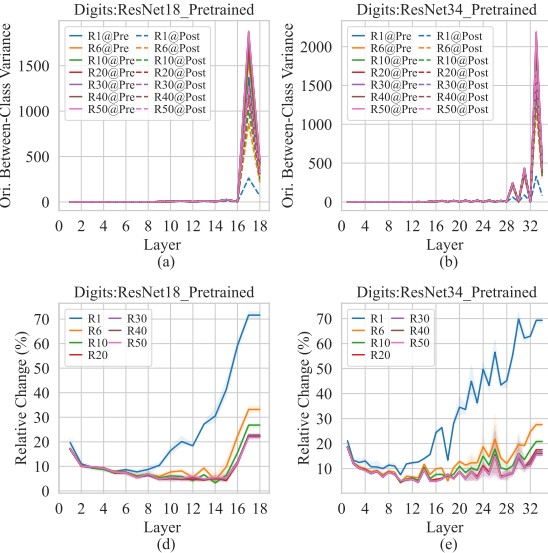

Figure 67: Changes in the original unnormalized between-class variance of features across model layers for specific global rounds, with larger X-axis values indicating deeper layers. The model is trained on Digit-Five with multiple models that are initialized by parameters pre-trained on large-scaled datasets. The top half of the figure shows the original unnormalized between-class variance, while the bottom half displays the relative change in variance before and after model aggregation.

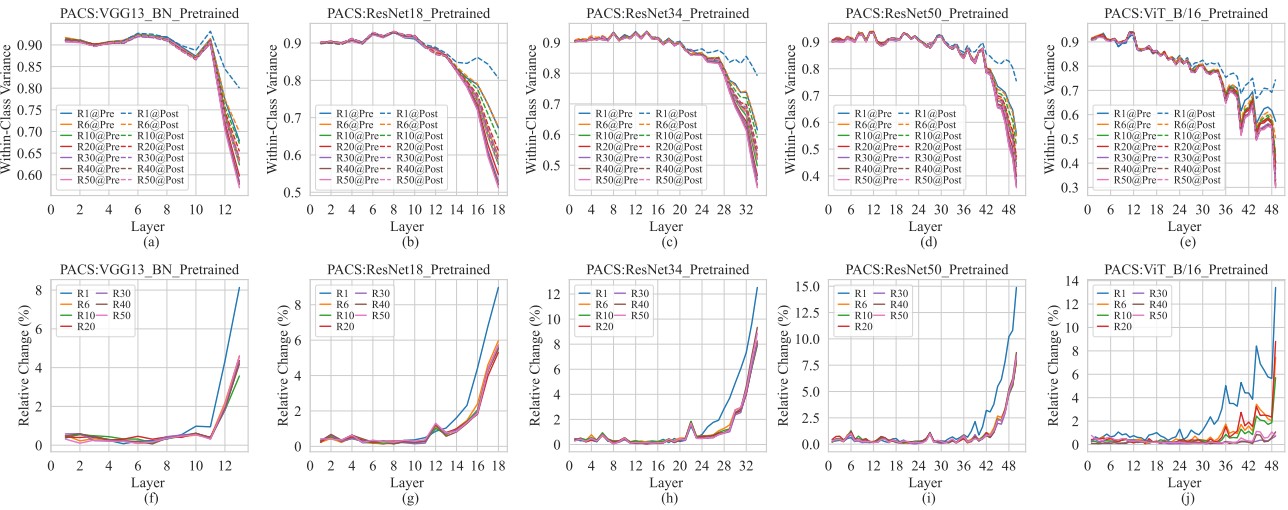

Figure 68: Changes in the normalized within-class variance of features across model layers for specific global rounds, with larger X-axis values indicating deeper layers. The model is trained on PACS with multiple models that are initialized by parameters pre-trained on large-scaled datasets. The top half of the figure shows the normalized within-class variance, while the bottom half displays the relative change in variance before and after model aggregation.

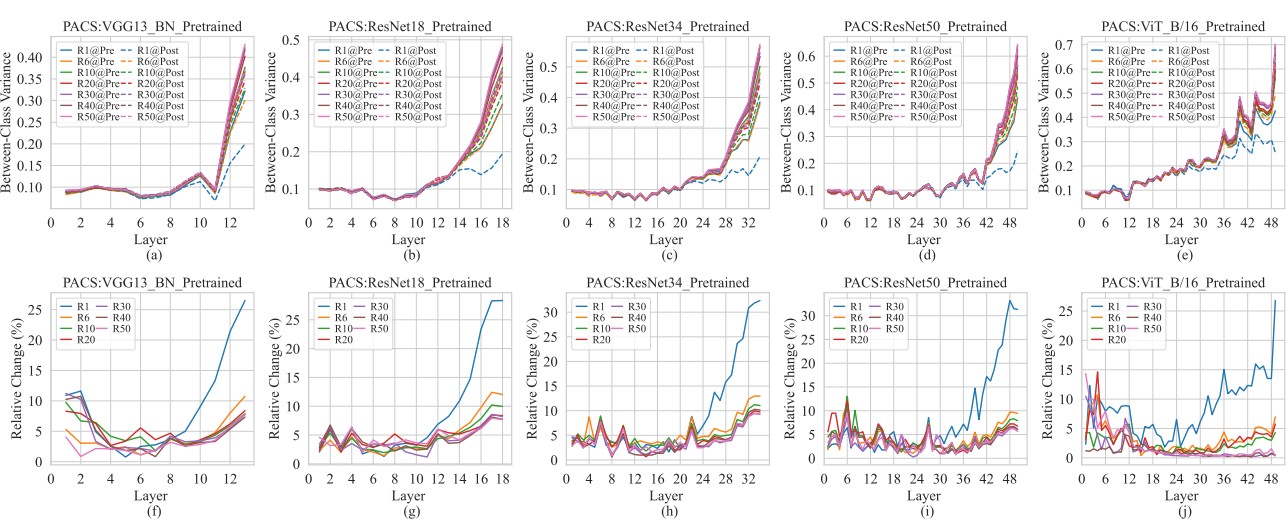

Figure 69: Changes in the normalized between-class variance of features across model layers for specific global rounds, with larger X-axis values indicating deeper layers. The model is trained on PACS with multiple models that are initialized by parameters pre-trained on large-scaled datasets. The top half of the figure shows the normalized between-class variance, while the bottom half displays the relative change in variance before and after model aggregation.

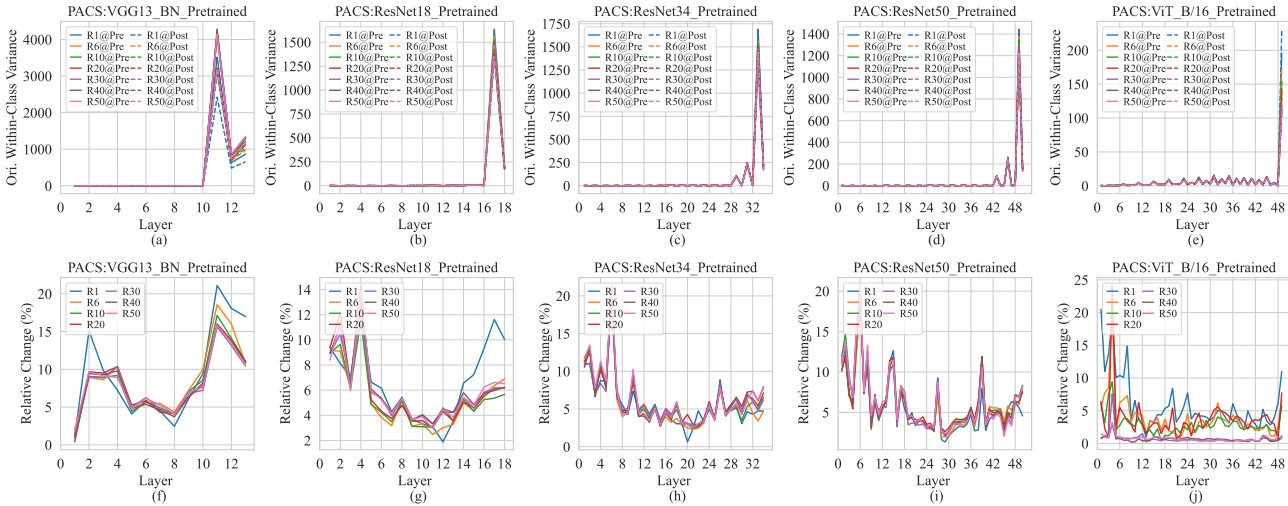

Figure 70: Changes in the original unnormalized within-class variance of features across model layers for specific global rounds, with larger X-axis values indicating deeper layers. The model is trained on PACS with multiple models that are initialized by parameters pre-trained on large-scaled datasets. The top half of the figure shows the original unnormalized within-class variance, while the bottom half displays the relative change in variance before and after model aggregation.

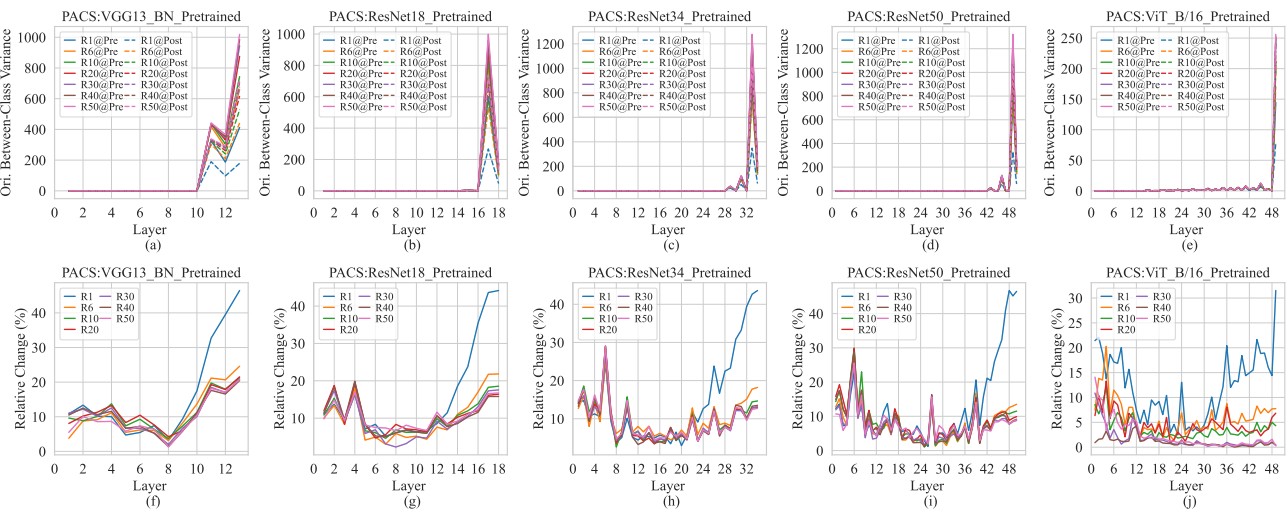

Figure 71: Changes in the original unnormalized between-class variance of features across model layers for specific global rounds, with larger X-axis values indicating deeper layers. The model is trained on PACS with multiple models that are initialized by parameters pre-trained on large-scaled datasets. The top half of the figure shows the original unnormalized between-class variance, while the bottom half displays the relative change in variance before and after model aggregation.

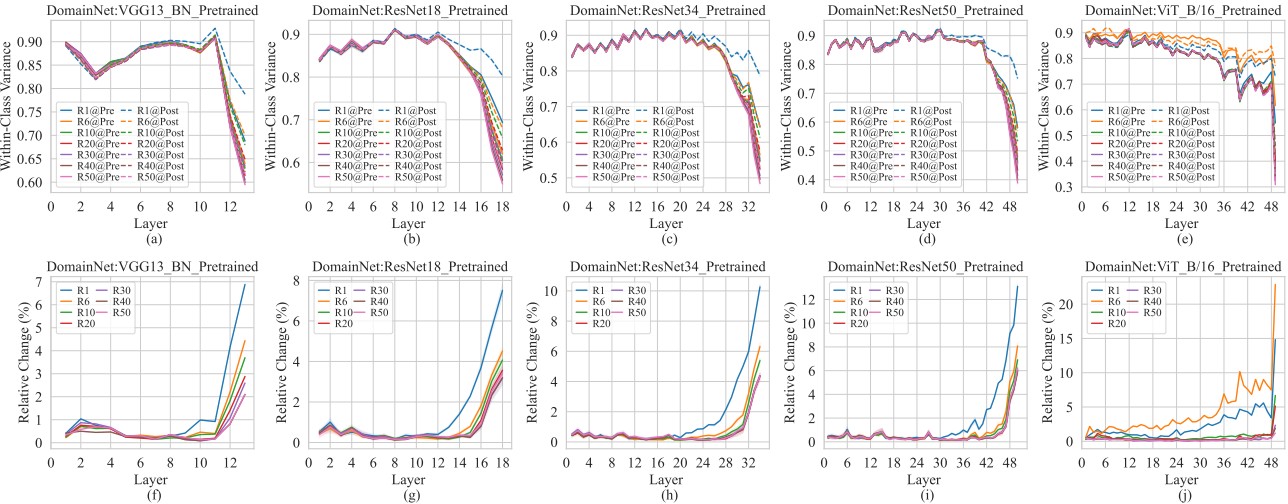

Figure 72: Changes in the normalized within-class variance of features across model layers for specific global rounds, with larger X-axis values indicating deeper layers. The model is trained on DomainNet with multiple models that are initialized by parameters pre-trained on large-scaled datasets. The top half of the figure shows the normalized within-class variance, while the bottom half displays the relative change in variance before and after model aggregation.

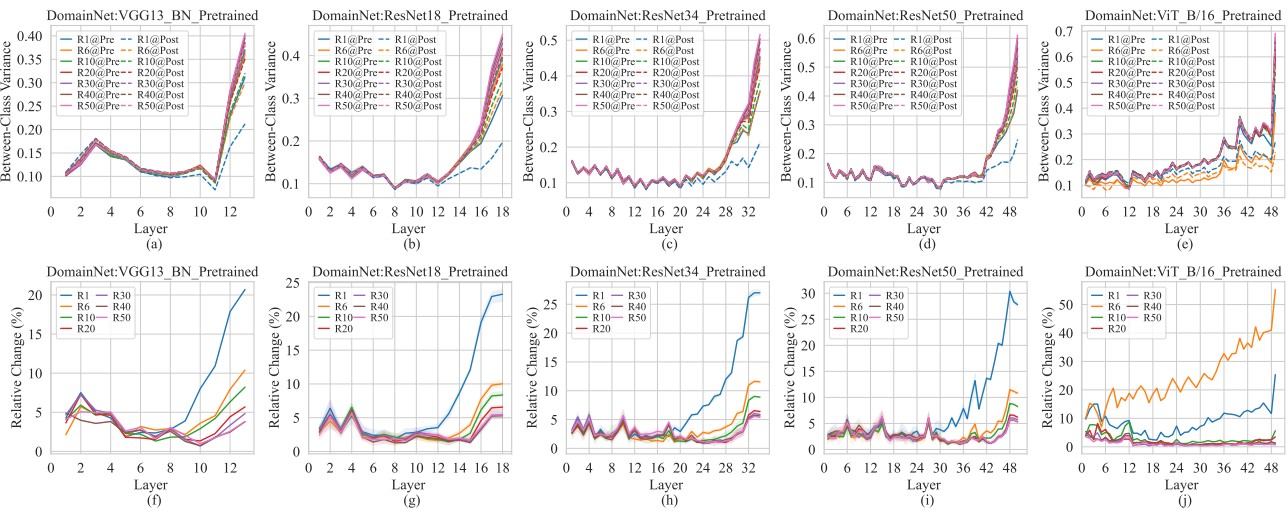

Figure 73: Changes in the normalized between-class variance of features across model layers for specific global rounds, with larger X-axis values indicating deeper layers. The model is trained on DomainNet with multiple models that are initialized by parameters pre-trained on large-scaled datasets. The top half of the figure shows the normalized between-class variance, while the bottom half displays the relative change in variance before and after model aggregation.

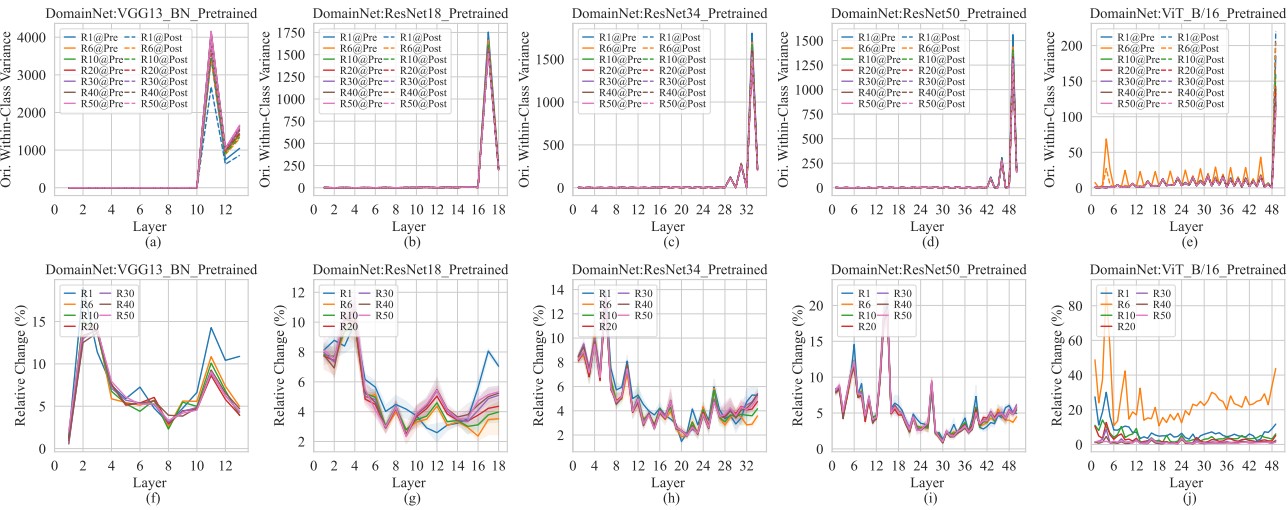

Figure 74: Changes in the unnormalized within-class variance of features across model layers for specific global rounds, with larger X-axis values indicating deeper layers. The model is trained on DomainNet with multiple models that are initialized by parameters pre-trained on large-scaled datasets. The top half of the figure shows the original unnormalized within-class variance, while the bottom half displays the relative change in variance before and after model aggregation.

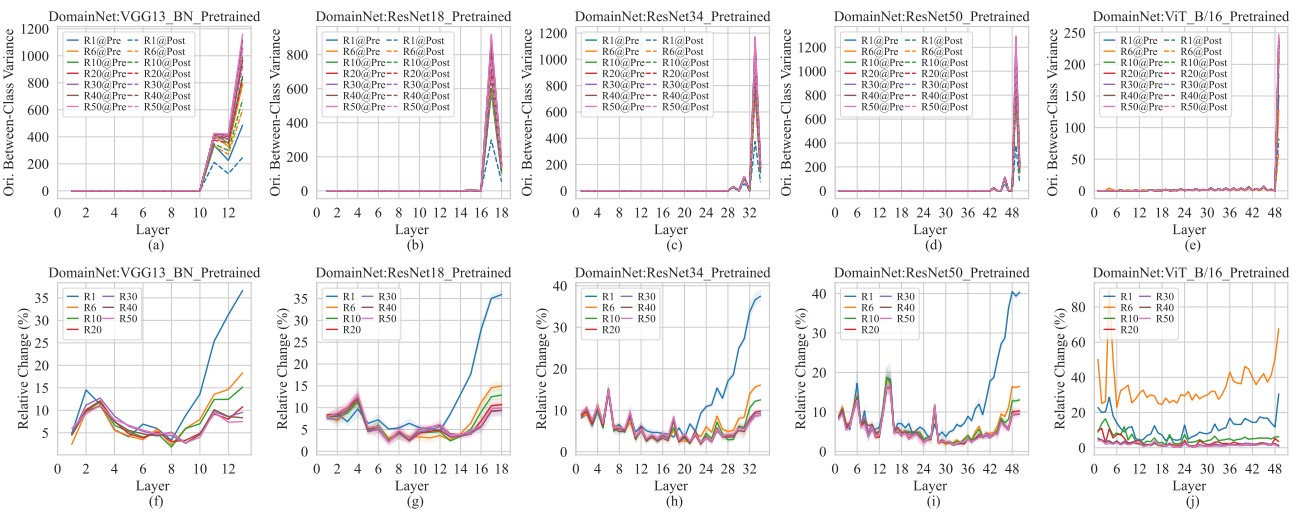

Figure 75: Changes in the unnormalized between-class variance of features across model layers for specific global rounds, with larger X-axis values indicating deeper layers. The model is trained on DomainNet with multiple models that are initialized by parameters pre-trained on large-scaled datasets. The top half of the figure shows the original unnormalized between-class variance, while the bottom half displays the relative change in variance before and after model aggregation.

R.3 CHANGES OF FEATURE VARIANCE ACROSS TRAINING ROUNDS

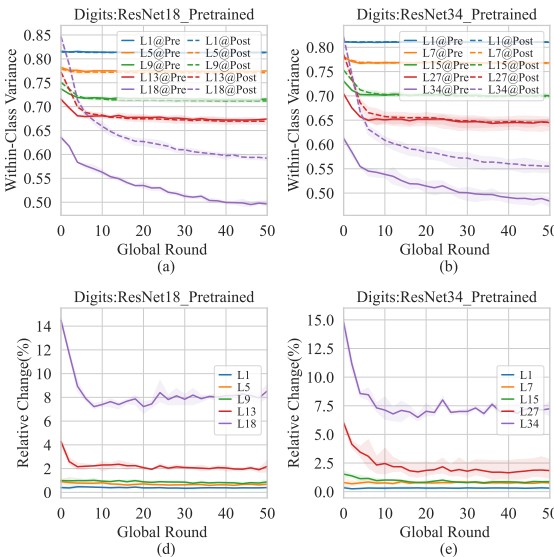

Figure 76: Changes in the normalized within-class variance of features across FL training at specific model layers. The model is trained on Digit-Five with multiple models that are initialized by parameters pre-trained on large-scaled datasets. The top half of the figure shows the original normalized within-class variance, while the bottom half displays the relative change in variance before and after model aggregation.

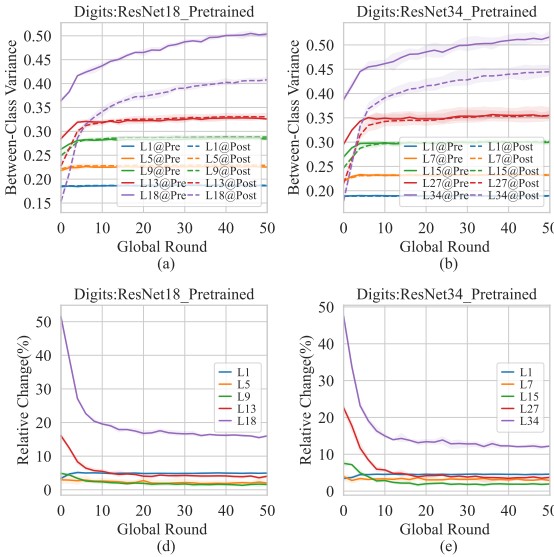

Figure 77: Changes in the normalized between-class variance of features across FL training at specific model layers. The model is trained on Digit-Five with multiple models that are initialized by parameters pre-trained on large-scaled datasets. The top half of the figure shows the normalized within-class variance, while the bottom half displays the relative change in variance before and after model aggregation.

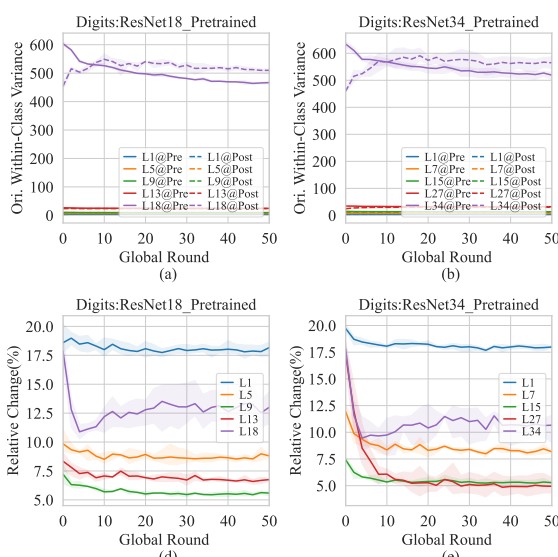

Figure 78: Changes in the original unnormalized within-class variance of features across FL training at specific model layers. The model is trained on Digit-Five with multiple models that are initialized by parameters pre-trained on large-scaled datasets. The top half of the figure shows the original unnormalized within-class variance, while the bottom half displays the relative change in variance before and after model aggregation.

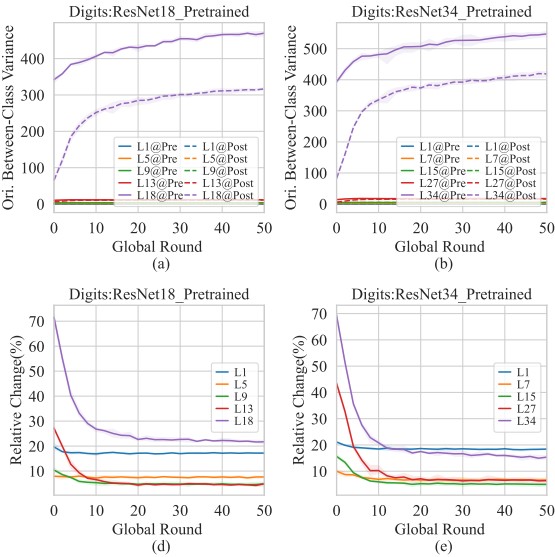

Figure 79: Changes in the original unnormalized between-class variance of features across FL training at specific model layers. The model is trained on Digit-Five with multiple models that are initialized by parameters pre-trained on large-scaled datasets. The top half of the figure shows the original unnormalized between-class variance, while the bottom half displays the relative change in variance before and after model aggregation.

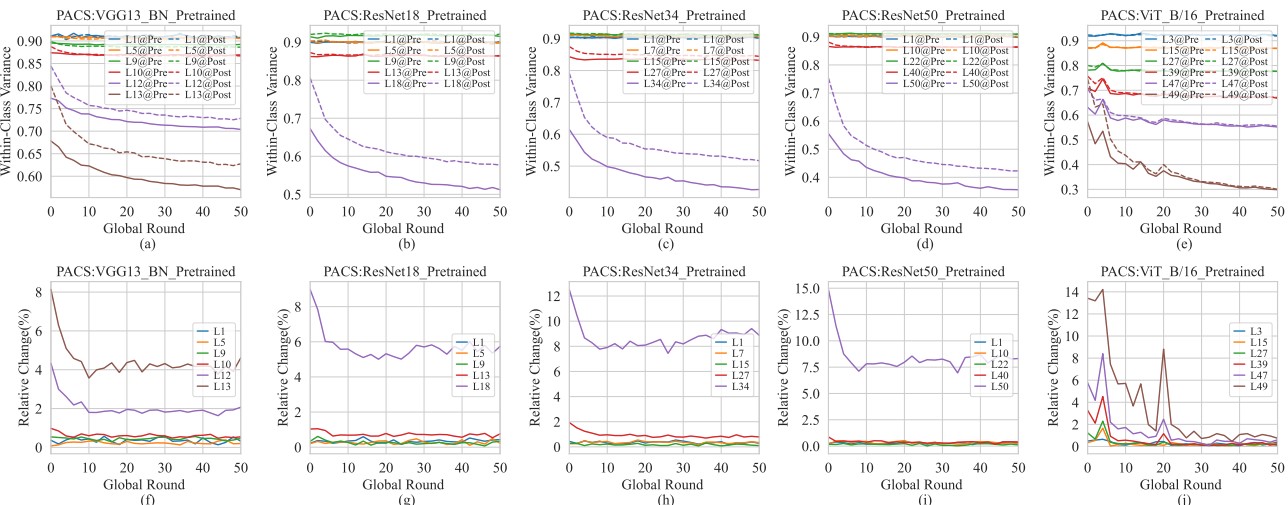

Figure 80: Changes in the normalized within-class variance of features across FL training at specific model layers. The model is trained on PACS with multiple models that are initialized by parameters pre-trained on large-scaled datasets. The top half of the figure shows the normalized within-class variance, while the bottom half displays the relative change in variance before and after model aggregation.

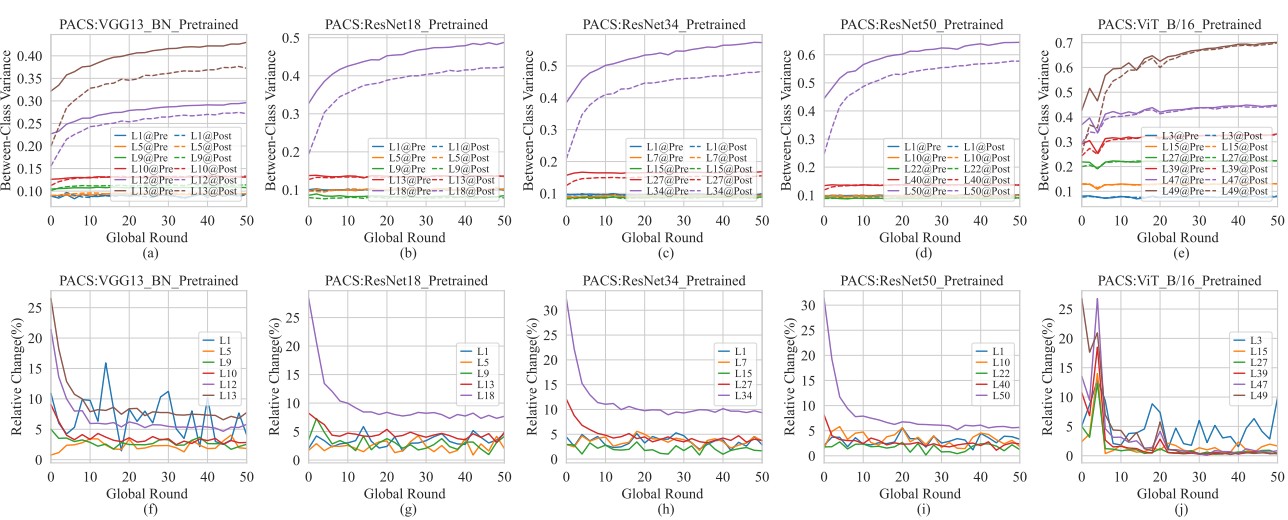

Figure 81: Changes in the normalized between-class variance of features across FL training at specific model layers. The model is trained on PACS with multiple models that are initialized by parameters pre-trained on large-scaled datasets. The top half of the figure shows the normalized between-class variance, while the bottom half displays the relative change in variance before and after model aggregation.

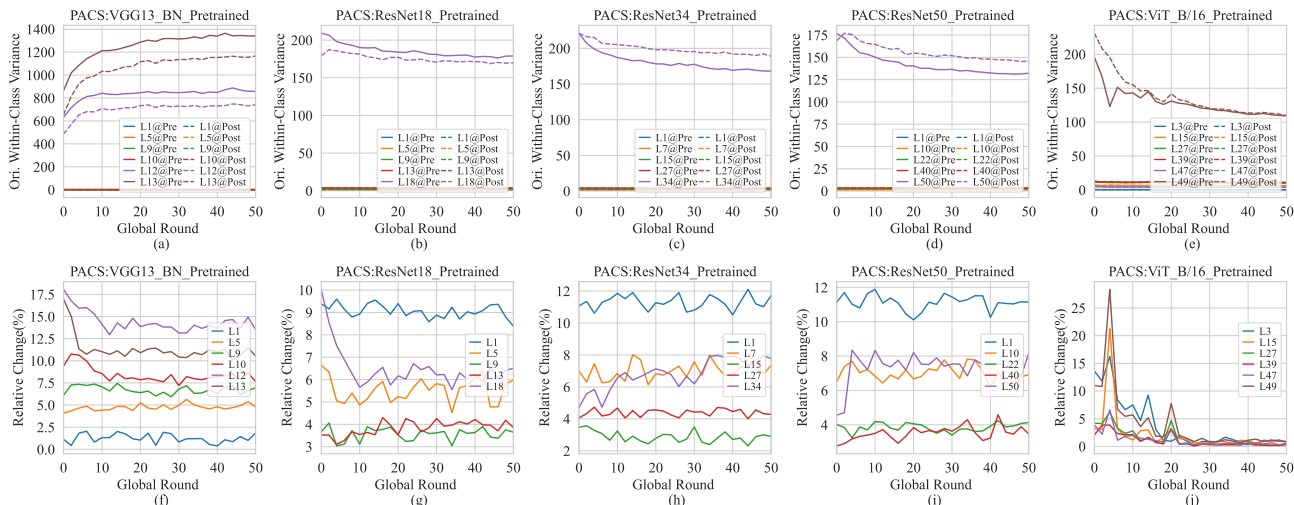

Figure 82: Changes in the original unnormalized within-class variance of features across FL training at specific model layers. The model is trained on PACS with multiple models that are initialized by parameters pre-trained on large-scaled datasets. The top half of the figure shows the original unnormalized within-class variance, while the bottom half displays the relative change in variance before and after model aggregation.

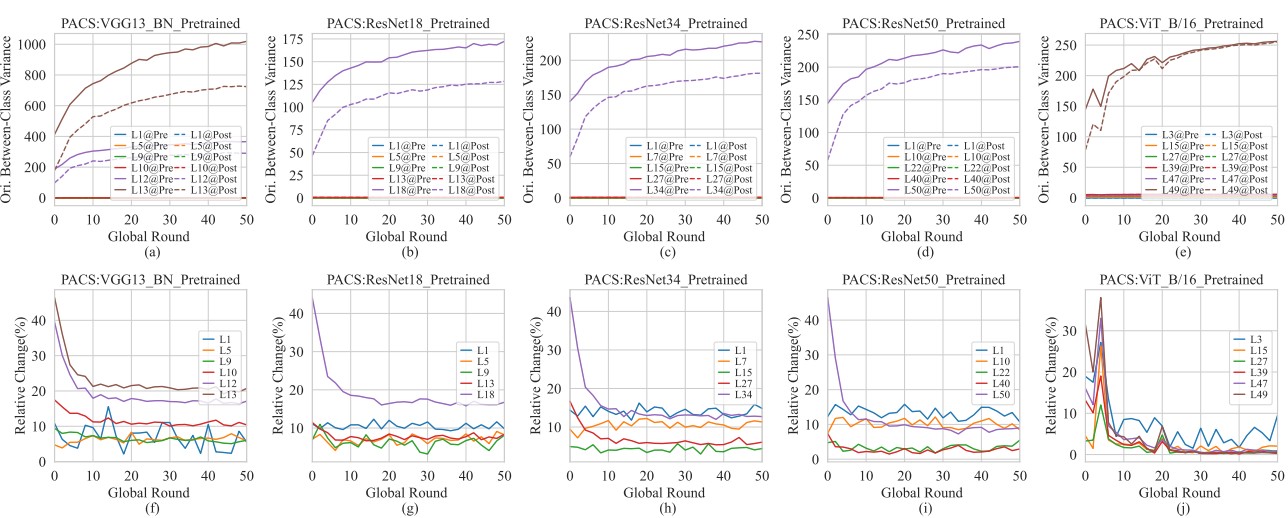

Figure 83: Changes in the original unnormalized between-class variance of features across FL training at specific model layers. The model is trained on PACS with multiple models that are initialized by parameters pre-trained on large-scaled datasets. The top half of the figure shows the original unnormalized between-class variance, while the bottom half displays the relative change in variance before and after model aggregation.

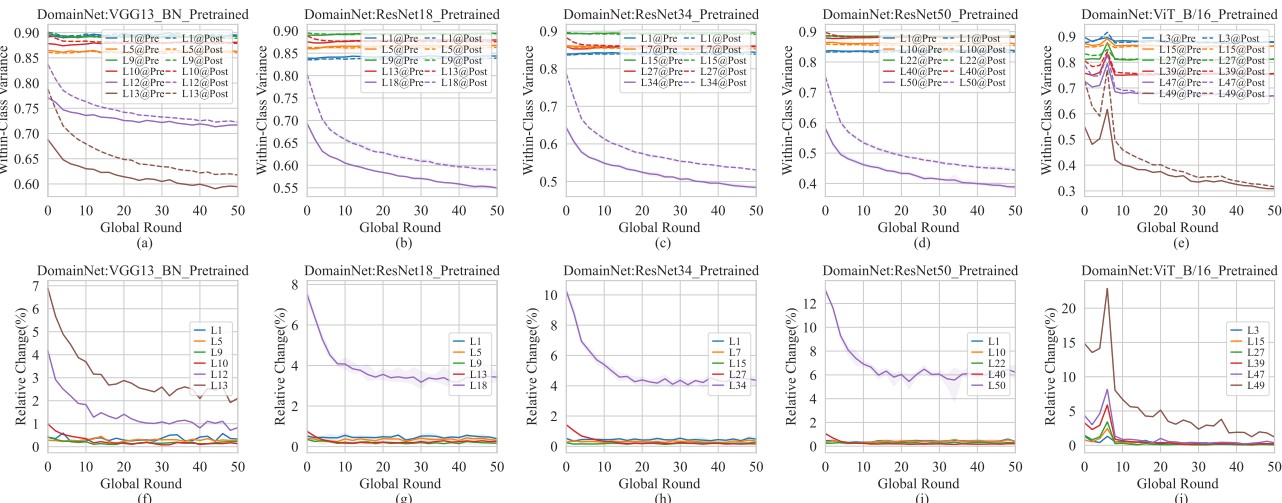

Figure 84: Changes in the normalized within-class variance of features across FL training at specific model layers. The model is trained on DomainNet with multiple models that are initialized by parameters pre-trained on large-scaled datasets. The top half of the figure shows the normalized within-class variance, while the bottom half displays the relative change in variance before and after model aggregation.

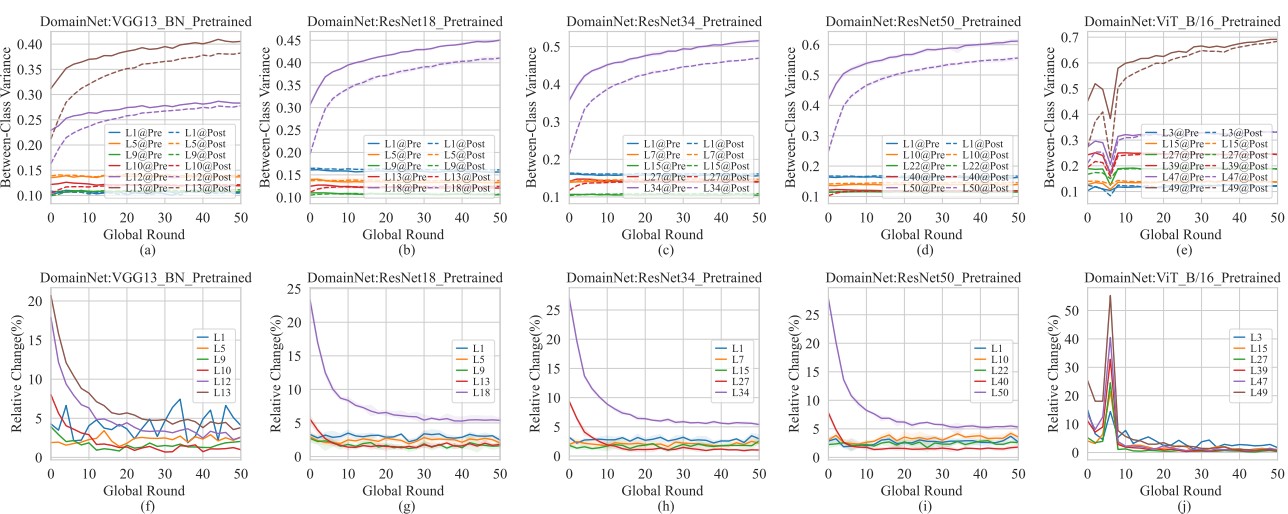

Figure 85: Changes in the normalized between-class variance of features across FL training at specific model layers. The model is trained on DomainNet with multiple models that are initialized by parameters pre-trained on large-scaled datasets. The top half of the figure shows the normalized between-class variance, while the bottom half displays the relative change in variance before and after model aggregation.

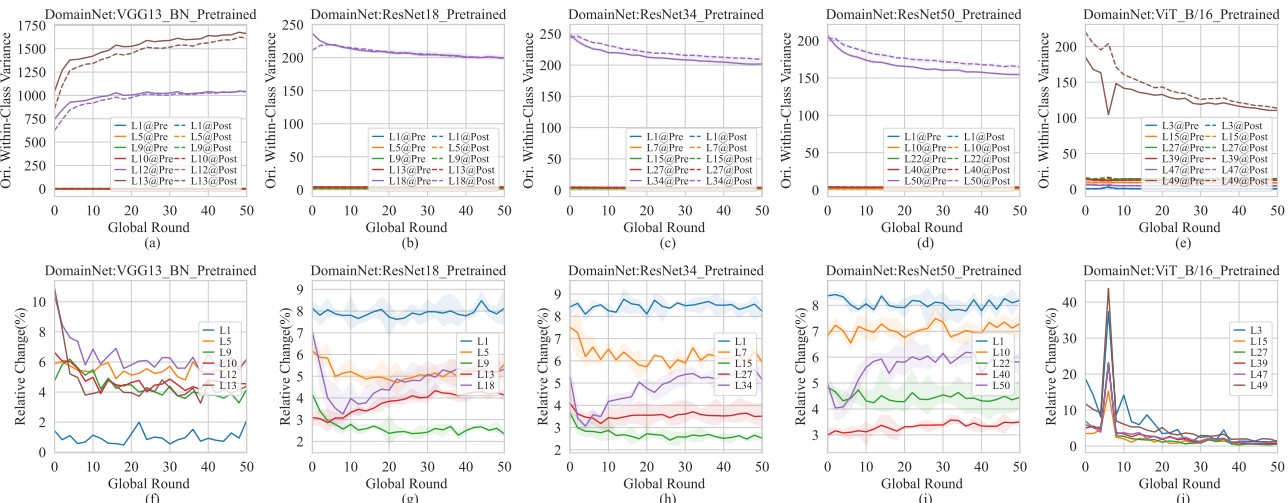

Figure 86: Changes in the original unnormalized within-class variance of features across FL training at specific model layers. The model is trained on DomainNet with multiple models that are initialized by parameters pre-trained on large-scaled datasets. The top half of the figure shows the original unnormalized within-class variance, while the bottom half displays the relative change in variance before and after model aggregation.

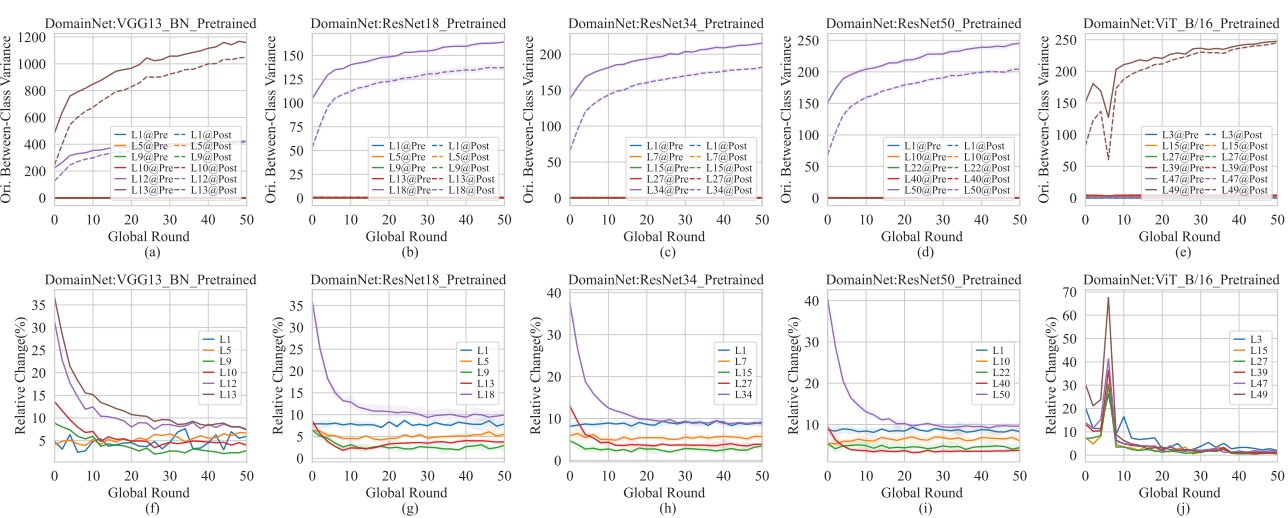

Figure 87: Changes in the original unnormalized between-class variance of features across FL training at specific model layers. The model is trained on DomainNet with multiple models that are initialized by parameters pre-trained on large-scaled datasets. The top half of the figure shows the original unnormalized between-class variance, while the bottom half displays the relative change in variance before and after model aggregation.

## R.4 Changes of Alignment Across Layers

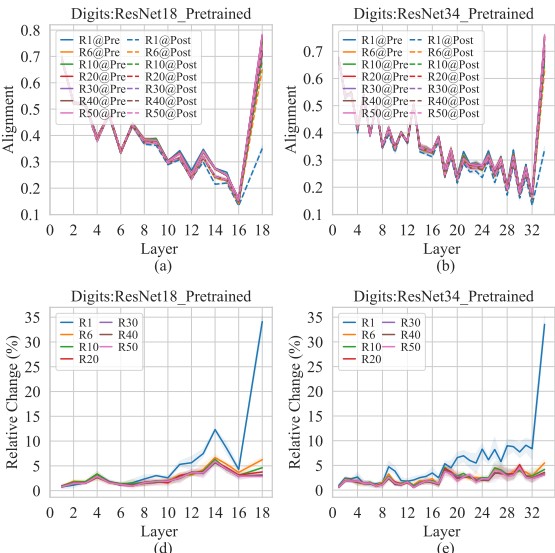

Figure 88: Changes in the alignment between features and parameters across model layers for specific global rounds, with larger X-axis values indicating deeper layers. The model is trained on Digit-Five with multiple models that are initialized by parameters pre-trained on large-scaled datasets. The top half of the figure shows the original alignment values between features and parameters, while the bottom half displays the relative change in alignment values before and after model aggregation.

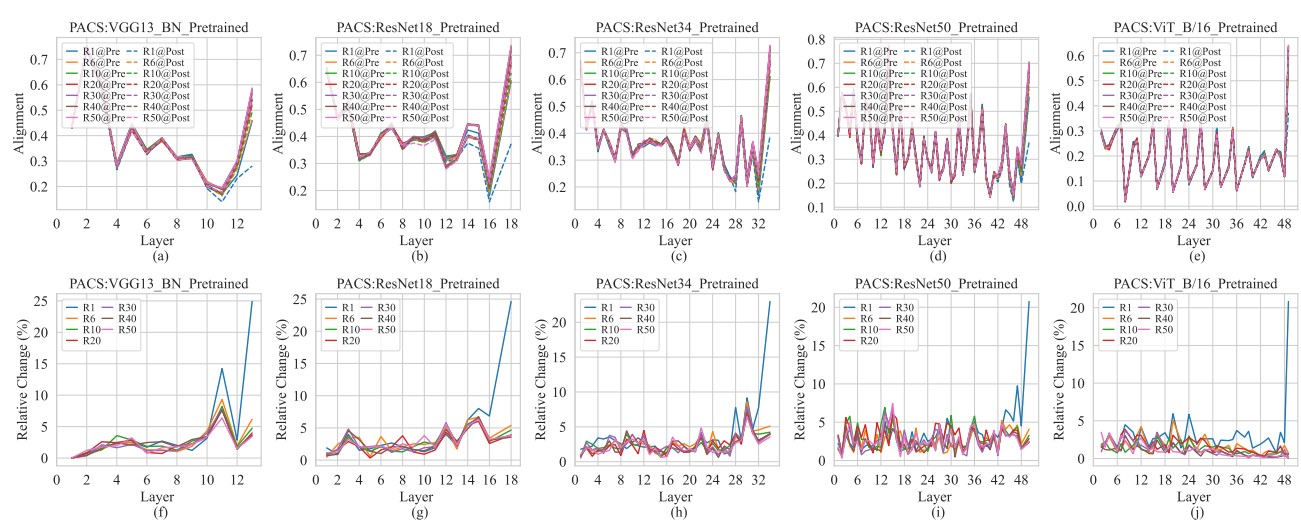

Figure 89: Changes in the alignment between features and parameters across model layers for specific global rounds, with larger X-axis values indicating deeper layers. The model is trained on PACS with multiple models that are initialized by parameters pre-trained on large-scaled datasets. The top half of the figure shows the original alignment values between features and parameters, while the bottom half displays the relative change in alignment before and after model aggregation.

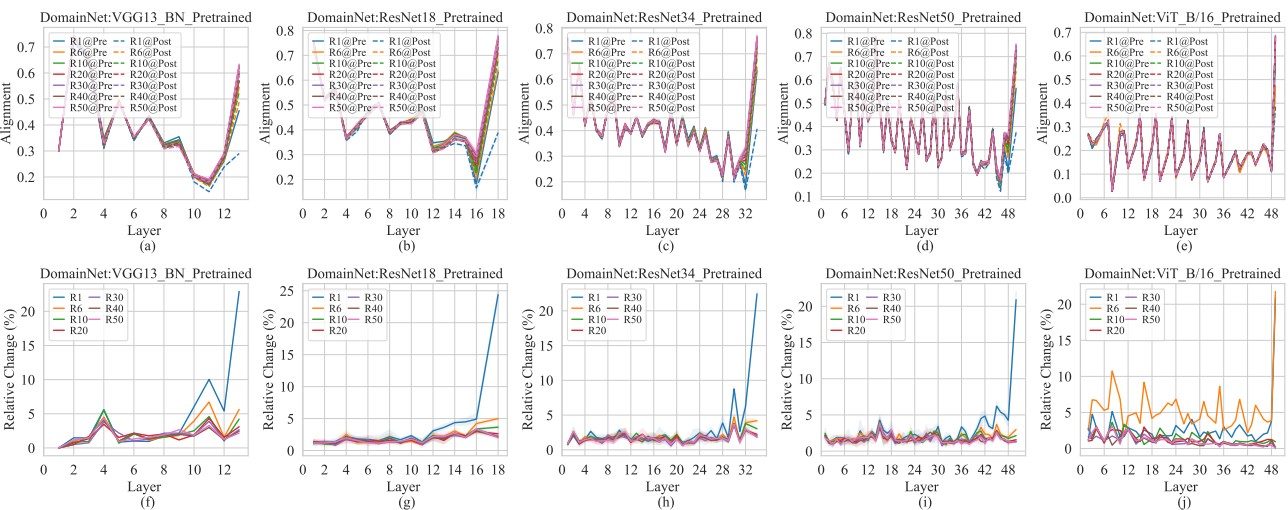

Figure 90: Changes in the alignment between features and parameters across model layers for specific global rounds, with larger X-axis values indicating deeper layers. The model is trained on DomainNet with multiple models that are initialized by parameters pre-trained on large-scaled datasets. The top half of the figure shows the original alignment values between features and parameters, while the bottom half displays the relative change in alignment before and after model aggregation.

## R.5 CHANGES OF ALIGNMENT ACROSS TRAINING ROUNDS

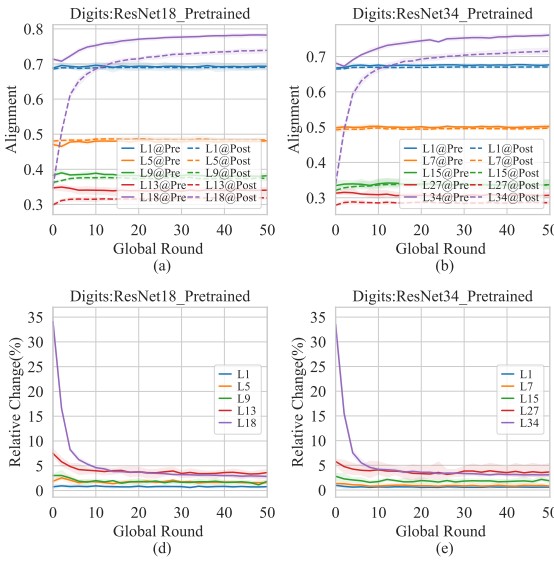

Figure 91: Changes in the alignment between features and parameters across FL training at specific model layers. The model is trained on Digit-Five with multiple models that are initialized by parameters pre-trained on large-scaled datasets. The top half of the figure shows the original alignment values between features and parameters, while the bottom half displays the relative change in alignment before and after model aggregation.

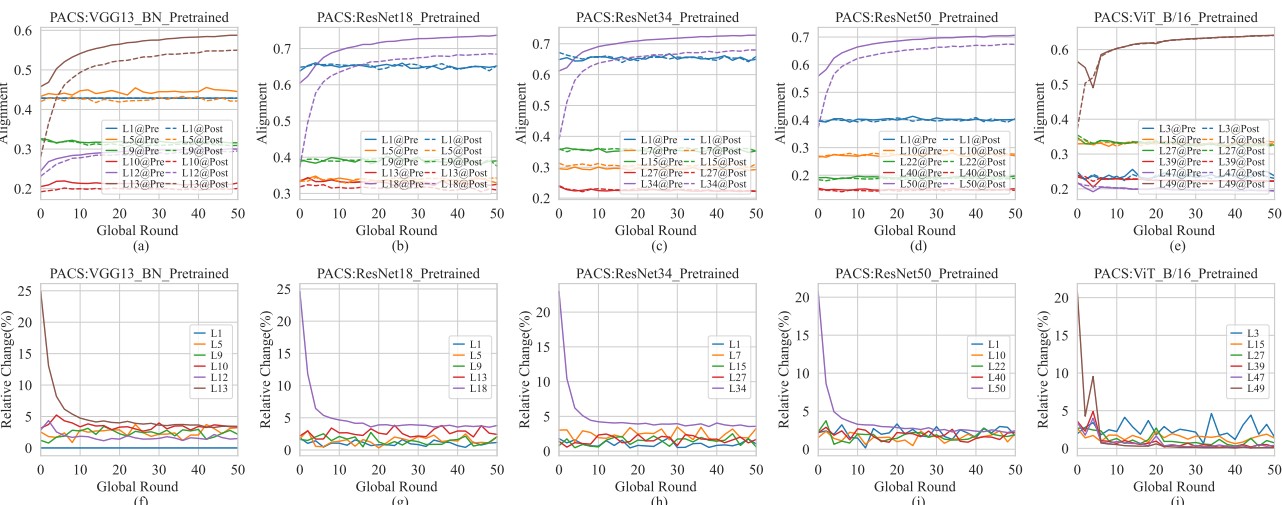

Figure 92: Changes in the alignment between features and parameters across FL training at specific model layers. The model is trained on PACS with multiple models that are initialized by parameters pre-trained on large-scaled datasets. The top half of the figure shows the original alignment values between features and parameters, while the bottom half displays the relative change in alignment before and after model aggregation.

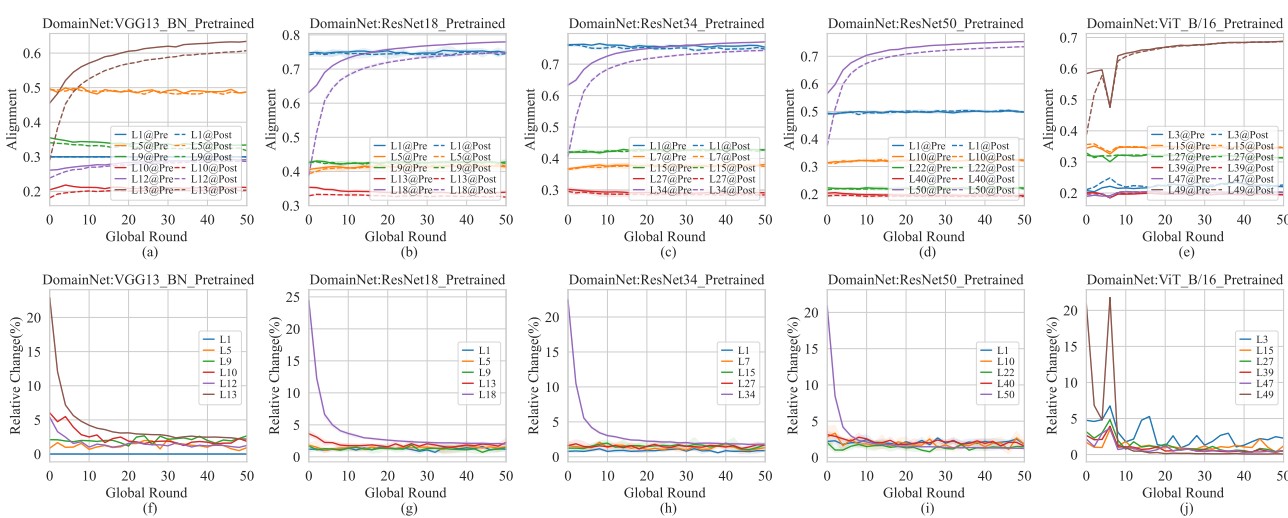

Figure 93: Changes in the alignment between features and parameters across FL training at specific model layers. The model is trained on DomainNet with multiple models that are initialized by parameters pre-trained on large-scaled datasets. The top half of the figure shows the original alignment values between features and parameters, while the bottom half displays the relative change in alignment before and after model aggregation.

R.6  VISUALIZATION OF PRE-AGGREGATED AND POST-AGGREGATED FEATURES

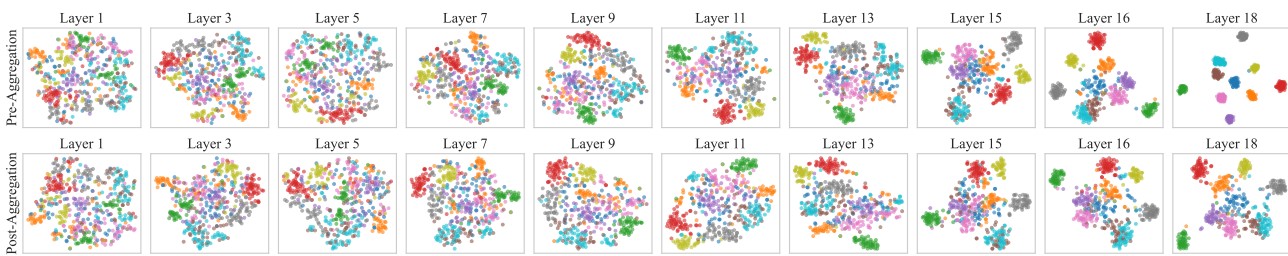

Figure 94: T-SNE visualization of features at different layers on the 'Quickdraw' domain of Domain-Net before and after aggregation. The features are extracted from ResNet18 in the final global round of FL training, whose parameters are initialized by the parameters pre-trained on large-scaled datasets at the beginning.

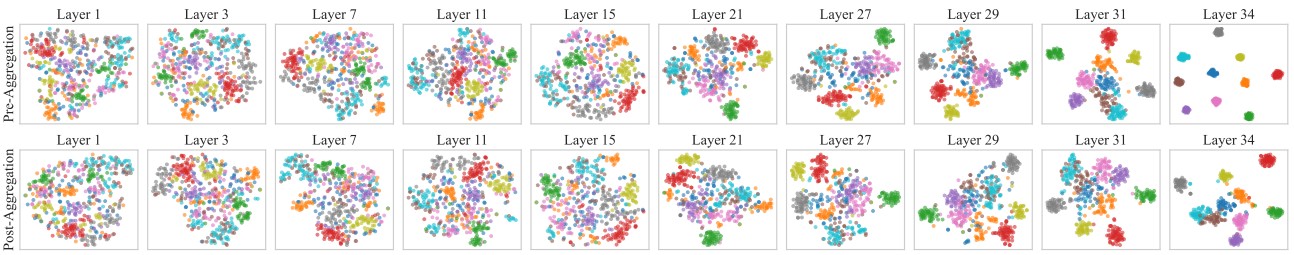

Figure 95: T-SNE visualization of features at different layers on the 'Quickdraw' domain of Domain-Net before and after aggregation. The features are extracted from ResNet34 in the final global round of FL training, whose parameters are initialized by the parameters pre-trained on large-scaled datasets at the beginning.

## S  FUTURE WORK

Our analysis reveals that CFD is fundamentally driven by the recursive nature of feature extraction, where the degradation at shallower layers propagates and amplifies through the network depth. This insight suggests that future FL optimization should shift from treating the model as a flat parameter vector to explicitly modeling the inter-layer feature relationships. Promising directions based on our findings include:

1. **Depth-Adaptive Aggregation Strategies:** Future algorithms should account for the varying sensitivity of layers at different depths. By decoupling the aggregation frequency or intensity of deeper layers from shallower ones, it is possible to block the error propagation path. For example, freezing the aggregation of shallower layers once they stabilize could prevent them from introducing noise to the deeper, more task-specific layers.

2. **In-Network Feature Correction:** Inspired by the layer-peeled perspective, future work could explore mechanisms to rectify features during the forward pass. Incorporating constraints or auxiliary modules that enforce feature alignment at intermediate layers would ensure that the input to subsequent layers remains clean, thereby preventing the "domino effect" of feature degradation.

## LLM USAGE STATEMENT

In this research, we used a Large Language Model (LLM) as a tool for language refinement, including the enhancement of clarity, fluency, and style in the manuscript. The LLM assisted with rephrasing sentences, improving readability, and ensuring consistency in terminology. However, all scientific content, including research design, analysis, and conclusions, was independently generated by the authors. We fully acknowledge our responsibility for the final content and the accuracy of the research presented. The LLM was not involved in the ideation, methodology, or data analysis aspects of this work.

