# OpenReview forum: "The Other Side of the Coin: Unveiling the Downsides of Model Aggregation in Federated Learning from a Layer-peeled Perspective"
_ICLR.cc/2026/Conference — Submitted to ICLR 2026_

### Official Review · Reviewer_4CgH · 2025-10-29

**Soundness:** 2
**Presentation:** 3
**Contribution:** 3
**Rating:** 6
**Confidence:** 4

**Summary:**

This paper investigates model aggregation in FL from a layer-wise feature perspective by proposing a layer-peeled analysis framework for a more interpretable lens to understand the internal dynamics of FL. The analysis reveals a key phenomenon termed Cumulative Feature Degradation (CFD), and the study further examines how different FL settings influence this degradation during model aggregation.

**Strengths:**

- The topic is relevant and important. The construction of a layer-peeled feature analysis framework is helpful.
- The findings of CFD help explain why the performance drop is so pronounced and why it is a fundamental challenge in aggregating deep models.
- The analysis covers multiple datasets and model architectures to support the findings.

**Weaknesses:**

- The current analysis is primarily empirical, relying on experimental metrics. The paper would be strengthened by incorporating theoretical analysis to support or generalize the empirical findings.
- While the abstract and introduction highlight aggregation frequency as a key factor, the corresponding analysis is put into appendix. Including it in the main text along with a more detailed discussion would be better.
- Although the paper successfully diagnoses a key issue (CFD), it does not propose concrete solutions or algorithmic adjustments inspired by the insights. The claim in the abstract that the work “potentially paves the way” for better FL algorithms would be more convincing if accompanied by specific, testable hypotheses or design principles.
- The figures and captions in the supplementary section can be further elaborated.

**Questions:**

Please see the weakness part.

---

> ### Author Response · Authors · 2025-12-03
>
> **Response to Weakness 1:** We appreciate this constructive suggestion. To bridge the gap between empirical observation and theoretical understanding, we have explicitly incorporated a theoretical analysis in Appendix G of the revised manuscript.
> Adopting the deep linear network assumptions used in prior studies, our theoretical derivation proves that model aggregation mathematically leads to the above specific negative effects we empirically observed in our paper:
>
> - Model aggregation increases the within-class variance ($\Sigma_W$) and decreases the between-class variance ($\Sigma_B$) of features on local data, as shown by:
>   $$\mathrm{Tr}(\tilde{\Sigma} _ W^\ell) > \mathrm{Tr}(\Sigma _ {W, m}^\ell) \quad \text{and} \quad \mathrm{Tr}(\tilde{\Sigma} _ B^\ell) < \mathrm{Tr}(\Sigma _ {B, m}^\ell)$$
>   where $\mathrm{Tr}(\tilde{\Sigma} _ W^\ell)$ and $\mathrm{Tr}(\Sigma _ {W, m}^\ell)$ represent the within-class feature variance for the aggregated model and the local model, $\mathrm{Tr}(\tilde{\Sigma} _ B^\ell)$ and $\mathrm{Tr}(\Sigma _ {B, m}^\ell)$ represent the between-class feature variance for the aggregated model and the local model, respectively.
>
> - The feature-parameter alignment score of the aggregated model is strictly lower than that of the optimal local model:
>   $$\mathcal{A}(\tilde{\mathbf{Z}}_m^\ell, \tilde{\mathbf{W}}^{\ell+1}) < \mathcal{A}(\mathbf{Z}_m^\ell, \mathbf{W}_m^{\ell+1})$$
>   where $\mathcal{A}(\tilde{\mathbf{Z}}_m^\ell, \tilde{\mathbf{W}}^{\ell+1})$ and $\mathcal{A}(\mathbf{Z}_m^\ell, \mathbf{W}_m^{\ell+1})$ represent the feature-parameter alignment for the aggregated and local models, respectively.
>
> - Let $E^\ell = \|\tilde{\mathbf{z}}_m^\ell - \mathbf{z}_m^\ell\|$ be the feature error at layer $\ell$ for client $m$. Assuming the spectral norm of aggregated weights is bounded by $\lambda$ ($\|\tilde{\mathbf{W}}^\ell\|_2 \le \lambda$) and the layer-wise heterogeneity noise is bounded by $\gamma$ ($\|\mathbf{\Delta}_m^\ell \mathbf{z}_m^{\ell-1}\| \le \gamma$), the feature error accumulates exponentially with depth $\ell$:
>   $$E^\ell \le \gamma \frac{\lambda^\ell - 1}{\lambda - 1} \quad (\text{for } \lambda > 1)$$
>
>
> These theoretical results provide a rigorous justification for feature degradation, and the CFD phenomenon in deep networks, confirming that our findings generalize beyond specific empirical settings.
>
> **Response to Weakness 2:** Thanks for your comments. We have moved the corresponding analysis regarding aggregation frequency to the main text and provided a more detailed discussion as suggested.
>
> **Response to Weakness 3:** We appreciate the constructive feedback. While our primary contribution in this work is identifying and characterizing the CFD mechanism, we agree that these insights directly point to specific algorithmic innovations. In the revised manuscript, we have outlined concrete design principles inspired by our finding that feature degradation accumulates via the layer-wise extraction process in Appendix S.
>
> Specifically, future algorithms can address CFD by "breaking the chain" of error propagation:
>
> - **Depth-Aware Aggregation:** Since we proved that degradation amplifies towards deeper layers, a uniform averaging strategy (like FedAvg) is suboptimal. A potential solution is to apply layer-wise adaptive aggregation rates. By assigning lower aggregation weights or performing less frequent aggregation for deeper layers (which are highly sensitive to noisy inputs from previous layers), we can mitigate the accumulation effect.
>
> - **Layer-wise Feature Rectification:** To prevent the error at layer $l$ from becoming a noisy input for layer $l+1$, future methods could introduce intermediate feature calibration. For instance, lightweight adapter modules could be inserted between layers to realign the statistics (e.g., variance) of the aggregated features before they are fed into the next layer, effectively resetting the accumulation counter.
>
> We believe these directions, grounded in our layer-peeled analysis, demonstrate how our diagnostic work paves the way for next-generation FL algorithms.
>
> Additionally, we will expand the figure captions to provide more detailed explanations of the experimental setups and to explicitly interpret the trends shown in the plots.

---

### Official Review · Reviewer_jURL · 2025-10-31

**Soundness:** 2
**Presentation:** 3
**Contribution:** 2
**Rating:** 4
**Confidence:** 3

**Summary:**

This paper presents a layer-peeled analysis framework to investigate performance degradation after model aggregation in federated learning (FL). The study shows that feature variance and feature-parameter alignment deteriorate as network depth increases, a phenomenon the paper refers to as cumulative feature degradation (CFD). The paper further demonstrates that model aggregation can improve feature generalization across clients. Finally, it analyzes how existing FL strategies mitigate the effects of CFD to achieve improved performance.

**Strengths:**

- Overall, the paper is well-written, and the figures and explanations are clear and easy to follow.

- The paper provides a systematic set of metrics for analyzing the dynamics of features and model parameters in FL settings.

**Weaknesses:**

- The analysis mainly focuses on the proposed analytical metrics without presenting accompanying accuracy trends to support the findings. While the paper suggests that model aggregation may degrade performance, it does not clearly demonstrate how the performance drops would correlate with the reported feature and parameter metrics and their dynamics.

- At Lines 273-279, the paper briefly introduces and defines CFD as the larger relative changes in the metrics as network depths increase. However, this definition is somewhat vague and lacks direct evidence that the degradation is indeed cumulative, since the observed relative changes may be influenced by various factors. For example, if performance is already low at earlier layers, the relative change may appear smaller simply because there is limited room for further degradation.

- The paper introduces CFD and uses it to analyze feature and parameter dynamics in FL, as well as to interpret the behavior of existing FL approaches. However, it is not entirely clear what concrete insights CFD offers that would meaningfully guide the design of future FL methods or lead to further advances in the field.

**Questions:**

Besides the weakness shown in the above section, please also see the following questions:

Q1: In Figures 2 and 3, the relative changes in feature variance increase with network depth. However, the feature variances in the shallow layers are already not performing well (e.g., large within-class variance and small between-class variance for “L1” in Figures 2(a) and 2(c)). In this case, can we still conclude that feature degradation becomes more severe in deeper layers? The smaller relative change observed in shallow layers may simply be due to their initially poor performance, rather than indicating less degradation.

Q2: In Figure 8, it seems that the personalization method FedBN still exhibits larger relative changes in the deeper layers than FedAvg. Does this imply that FedBN is less effective in mitigating CFD? Additionally, what is the accuracy comparison between FedAvg and FedBN?

Q3: At Lines 14-16 in the abstract, the paper states that performance drops after aggregation can potentially slow down the convergence of FL. However, in Section 4.4, the results indicate that model aggregation improves generalization. Why would improved model generalization hinder convergence? Does this imply that without aggregation, the model would converge more quickly but to an overfitted local minimum? If so, would slower convergence in this case actually be preferable?

---

> ### Author Response · Authors · 2025-12-03
>
> **Response to Weakness 1:** Thanks for your comments. We have included a quantitative analysis of the correlation between the degree of feature degradation and the accuracy drop.
> The experimental results are shown below, which demonstrate a strong correlation between these two metrics. This confirms that severe feature degradation is indeed a primary factor contributing to the performance decline in global models.
>
> | Feature Metric| E1    | E20   | E30   | E40   | E50   |
> |------------------------------|-------|-------|-------|-------|-------|
> | Within-Class Variance| -0.43 | -0.61 | -0.32 | -0.69 | -0.21 |
> | Alignment Change | -0.72 | -0.66 | -0.50 | -0.66 | -0.43 |
>
> **Response to Weakness 2 and Question 1:** Thank you for this insightful comment. To clarify the definition and verify the "cumulative" nature of CFD, we conducted an ablation study using a partial aggregation strategy. Specifically, during FedAvg training, we selectively excluded a varying number of shallow layers from aggregation (keeping them local) while aggregating the remaining deeper layers.
> The table below shows the metrics of final layer features under different aggregation strategies, where C1 refers to the first convolutional layer of ResNet, and S* denotes the block in ResNet. The parameters aggregated from left to right in the table gradually decrease.
> This demonstrates a clear causal pattern: the more shallow layers we keep unaffected by aggregation, the less feature degradation is observed in the final layer. These results suggest that the degradation is not merely due to layer-specific sensitivity or low performance baselines. Instead, it confirms that the degradation originates in shallower layers and propagates/accumulates to the deeper layers.
>
> | Metric | FedAvg | Per_C1 | Per_C1_S1 | Per_C1_S1-2 | Per_C1_S1-3 | Per_C1_S1-4 |
> |---|---|---|---|---|---|---|
> | Within-class Variance | 17.08±7.12 | 15.25±4.25 | 11.33±3.38 | 6.84±2.53 | 0.85±0.80 | 0.57±0.55 |
> | Alignment | 3.04±1.99 | 3.11±1.63 | 1.66±0.73 | 0.74±0.40 | 0.19±0.17 | 0.05±0.06 |
>
> **Response to Weakness 3:** We appreciate the reviewer's comment on the practical implications of our work. While our primary contribution is identifying and characterizing the CFD mechanism, we agree that these findings should translate into actionable design principles. In the revised manuscript (Appendix S), we have added a discussion outlining concrete algorithmic innovations inspired by our insight that feature degradation accumulates via the layer-wise extraction process.
>
> Specifically, future algorithms can address CFD by "breaking the chain" of error propagation in two ways:
>
> - **Depth-Aware Aggregation:** Since our analysis proves that degradation amplifies towards deeper layers, a uniform averaging strategy (like FedAvg) is suboptimal. A concrete design principle is to apply layer-wise adaptive aggregation. For instance, assigning lower aggregation weights or performing less frequent aggregation for deeper layers (which are highly sensitive to noisy inputs) can effectively mitigate the accumulation effect.
>
> - **Layer-wise Feature Rectification:** To prevent the error at layer $l$
>  from becoming a noisy input for layer $l+1$
> , future methods could introduce intermediate feature calibration. For example, lightweight adapter modules could be inserted between layers to realign the statistics (e.g., variance) of the aggregated features before they are fed into the next layer, effectively "resetting" the accumulation counter.
> We believe these specific directions, grounded in our layer-peeled analysis, demonstrate how our work paves the way for next-generation FL algorithms.

---

> ### Author Response · Authors · 2025-12-03
>
> **Response to Question 2:** Thank you for the detailed observation. We response your questions as below.
>   - **Why FedBN exhibits high CFD:**
> Although FedBN keeps Batch Normalization (BN) layers localized to handle statistical heterogeneity, it still performs aggregation on all other weight parameters (e.g., Convolutional/Linear layers) across the entire network depth. Consequently, the "path length" for error propagation is not reduced. The larger relative changes observed in FedBN imply that the shared weights undergo significant shifts to adapt to the diverse local BN statistics, which ironically can exacerbate the feature-parameter misalignment during aggregation. This confirms that simply fixing local statistics (via FedBN) does not resolve the structural accumulation of feature degradation.
>
>   - **Accuracy Comparison:**
> Below is the accuracy at the epoch used for plotting the figure. The overall mean accuracy of FedAvg and FedBN remains relatively consistent, though there are differences in accuracy across domains.
>
>
> | Method | Real | Sketch | Clipart | Infograph | Painting | Quickdraw |
> |---|---|---|---|---|---|---|
> | FedAvg | 69.47±2.70 | 54.07±4.01 | 77.33±2.54 | 59.73±4.74 | 60.73±2.96 | 38.33±1.23 |
> | FedBN | 65.67±2.08 | 55.67±6.70 | 76.20±1.61 | 58.53±2.75 | 64.07±2.31 | 39.07±1.68 |
>
> **Response to Question 3:**
> Thank you for this insightful question. We appreciate the opportunity to clarify the relationship between convergence speed and generalization in the context of CFD.
>   - **Clarification of Definitions:**
>     **Convergence:** In our context, this refers to the efficiency of the training process—specifically, how many communication rounds are required for the global model to minimize the training loss on participating clients.
>     **Generalization:** This refers to the final model's ability to transfer to unseen data or other clients (measured by linear probing).
>
>   - **Why improved generalization can coexist with slower convergence:**
> Ideally, should achieve good generalization efficiently by aggregating the knowledge learned during training. However, our findings show that CFD acts as a disturbance. At each aggregation step, feature degradation occurs, and subsequent local training must first "repair" these degraded features before learning new information on local clients. This "damage-repair" cycle hinders the convergence rate.
>
>   - **Is slower convergence preferable?**
> We agree with your opinion that without aggregation, the model would converge more quickly but to an overfitted local minimum.
> However, this does mean that this slow convergence is preferable. Specifically, The slowed convergence is caused by feature destruction, not by beneficial regularization that prevents overfitting. If we could eliminate CFD, the model would leverage the diverse data for generalization without wasting rounds on repairing features, thus achieving the same (or better) generalization more quickly.

---

### Official Review · Reviewer_cMUs · 2025-10-31

**Soundness:** 2
**Presentation:** 2
**Contribution:** 1
**Rating:** 2
**Confidence:** 4

**Summary:**

This paper investigates why model averaging in federated learning (FL) often causes a temporary drop in clients’ local performance after aggregation. The authors identify that existing work treats this post-aggregation degradation as an inherent cost without explaining its internal mechanisms. To address this, they propose a layer-peeled analysis framework that examines how aggregation alters feature representations and their alignment with subsequent layers, introducing the concept of Cumulative Feature Degradation (CFD), a depth-accumulating degradation of feature quality and feature-parameter alignment. Through empirical analysis, they show that aggregation increases within-class variance, decreases between-class variance, and disrupts alignment between penultimate features and classifiers, while also improving out-of-distribution generalization.

**Strengths:**

1. The figure illustrating the layer-wise performance trend is well-presented and effectively supports the analysis.

2. The experimental setup is described with sufficient clarity and detail to ensure reproducibility.

**Weaknesses:**

1. Limited novelty compared to prior layer-wise/feature-alignment analyses. Prior work already diagnoses aggregation-induced feature/layer misalignment and layer-dependent behavior, and studies when layer-wise averaging or alignment helps (e.g., Fed2 [1] aligns features across clients; pFedLA [2] learns layer-wise aggregation analysis in personalized FL setting; FedFA provides detailed analysis of latent feature statistics and provide a feature alignment method; Layer-wise Linear Mode Connectivity [4] shows layers often admit linear connectivity with thorough analysis of the layer-wise parameter dynamics in model aggregation). I don’t clearly see what new insight this paper adds beyond those feature alignment analyses.

2. CFD seems like a correlate, not a fundamental driver. The experiments mainly establish correlations between CFD metrics and accuracy without causal interventions; the “cumulative” phrasing is also puzzling because aggregation is a weight-averaging step (no depth-wise propagation), and the depth trend likely reflects local training signals rather than averaging per se.

3. Explanation is generic and widely known. The argument reduces to “within-class variance rises and between-class separation falls after averaging,” which mirrors established neural-collapse/feature-separation results [5, 6] in standard deep nets; please clarify what is federation-specific beyond these generic patterns or provide theory linking heterogeneity of federated learning setting.

[1] Yu, Fuxun, et al. "Fed2: Feature-aligned federated learning." Proceedings of the 27th ACM SIGKDD conference on knowledge discovery & data mining. 2021.
[2] Ma, Xiaosong, et al. "Layer-wised model aggregation for personalized federated learning." Proceedings of the IEEE/CVF conference on computer vision and pattern recognition. 2022.
[3] Zhou, Tianfei, and Ender Konukoglu. "FedFA: Federated Feature Augmentation." The Eleventh International Conference on Learning Representations.
[4] Adilova, Linara, et al. "Layer-wise linear mode connectivity." The Twelfth International Conference on Learning Representations.
[5] Papyan, Vardan, X. Y. Han, and David L. Donoho. "Prevalence of neural collapse during the terminal phase of deep learning training." Proceedings of the National Academy of Sciences 117.40 (2020): 24652-24663.
[6] Parker, Liam, et al. "Neural collapse in the intermediate hidden layers of classification neural networks." arXiv preprint arXiv:2308.02760 (2023).

**Questions:**

See weakness

---

> ### Author Response · Authors · 2025-12-03
>
> **Response to Weakness 1:** Thank you for your comments. We would like to clarify that our paper significantly differs from the listed works in your references.
>
> The provided feature-alignment approaches focus on aligning features across different clients [1], whereas our paper centers on the alignment between features and parameters within the internal layers of the models. Regarding the layer-wise analysis works [2, 4], these studies typically operate within the parameter space and do not consider the layer-by-layer feature extraction, where features extracted by earlier layers can influence subsequent layers. In contrast, our paper reveals the effects of model aggregation in data-heterogeneous FL from a layer-by-layer feature extraction perspective, which helps to better understand the internal feature dynamics caused by model aggregation. Below is a more detailed discussion:
>
> - First, we would like to emphasize that work [4], which we consider the most relevant to our study and has been discussed in our manuscript, also performs a layer-wise analysis in FL. However, this work uses loss or error metrics to assess changes when aggregating partial or full model parameters in a layer-wise manner. This approach overlooks the dynamics of intermediate features during the feature extraction process across the model’s depth, which is a central focus of our paper.
>
> - Second, we have compared our work with studies focused on aligning features across clients, such as FedProto [7] and FPL [8], which are similar to work [1] in your references. However, these studies typically rely on measuring the distance between features across clients. This distance may not accurately reflect the quality of features extracted from raw data. In contrast, our work focuses on the inherent feature structure within local clients, measuring the quality of features on each client. Furthermore, we believe there may be a misunderstanding regarding the use of the term "alignment" in the referenced studies versus our paper. As discussed earlier, most current studies focus on misalignment between features or layers across clients, whereas our work addresses the alignment between extracted features and their subsequent parameters, computed within a single model on one client.
>
> - Third, compared to layer-wise solutions in FL, whether in the parameter space (work [2,4]) or feature space (work [3]), our study offers distinct insights. Specifically, while these studies explore layer-wise aggregation or feature augmentation, they do not address the progressive feature extraction in deep models—the interdependencies between layers. In contrast, our work analyzes how model aggregation progressively affects feature extraction, layer by layer, from raw data to high-level representations.
>
> We appreciate the references you provided. We have included this discussion in our revised manuscript to further clarify the unique contributions of our work. Please refer to the Introduction and Related Works sections for more details.

---

> ### Author Response · Authors · 2025-12-03
>
> **Response to Weakness 2:**
> Thanks for your comments. We address your concerns regarding the "cumulative" nature of CFD and causality below.
> - **Clarification on "Cumulative" feature degradation.**
> we believe there is a misunderstanding regarding the core contribution and perspective of our work. Unlike previous studies that focus on the parameter space, our analysis is fundamentally rooted in a layer-peeled feature extraction perspective, specifically examining the representation degradation before and after aggregation. In this context, "cumulative" characterizes the propagation of degradation along the network depth during the forward feature extraction process, rather than analyzing the dynamics of local training or the formation of local models. Since deep networks are sequential, the feature error induced by aggregation at a shallower layer becomes noisy input for the subsequent layer. Therefore, the degradation accumulates layer-by-layer. This is a structural consequence of aggregating hierarchical representations, distinct from the properties of the local training process.
>
> - **Causal link between CFD and accuracy.**
> Proving causality is challenging, and we acknowledge that we have not theoretically proven causality between CFD and accuracy drop. However, we have made the following efforts to establish a causal relationship between the two.
> First, as noted in our experiments (see **Section 6**), methods that explicitly mitigate CFD (e.g., by improving feature-parameter alignment or personalizing shallow parameters) directly lead to better convergence. This suggests that CFD is not merely a correlation of performance degradation.
> Moreover, our theoretical analysis (see **Appendix G**) proves that aggregation mathematically causes the increase in within-class variance and misalignment. Since classification accuracy fundamentally depends on feature separability, this theoretical link establishes a causal pathway from aggregation $\rightarrow$ CFD $\rightarrow$Performance Drop.

---

> ### Author Response · Authors · 2025-12-03
>
> **Response to Weakness 3:** Thank you for your comments. We would like to clarify that there has been a misunderstanding regarding our contribution. While our findings show that the total training objective mirrors the established Neural Collapse (NC) results [5, 6] in centralized learning, this is not the core contribution of our paper.
>
> In fact, one of the major contributions of our paper is that we reveal how model aggregation in FL increases within-class variance and decreases between-class variance, which is exactly the opposite of the NC phenomenon in centralized settings. In addition to the degradation in feature variance, we also observe decreases in feature-parameter alignment caused by aggregation. More importantly, we show that this degradation can accumulate across network depth, causing significant degradation of features in the final layers and a substantial disruption of feature-parameter alignment.
>
> Furthermore, we have conducted theoretical analysis to correspond to the observed feature degradation phenomenon discussed above. The theoretical conclusions are shown below:
>
> - Model aggregation increases the within-class variance ($\Sigma_W$) and decreases the between-class variance ($\Sigma_B$) of features on local data, as shown by:
>   $$\mathrm{Tr}(\tilde{\Sigma} _ W^\ell) > \mathrm{Tr}(\Sigma _ {W, m}^\ell) \quad \text{and} \quad \mathrm{Tr}(\tilde{\Sigma} _ B^\ell) < \mathrm{Tr}(\Sigma _ {B, m}^\ell)$$
>   where $\mathrm{Tr}(\tilde{\Sigma} _ W^\ell)$ and $\mathrm{Tr}(\Sigma _ {W, m}^\ell)$ represent the within-class feature variance for the aggregated model and the local model, $\mathrm{Tr}(\tilde{\Sigma} _ B^\ell)$ and $\mathrm{Tr}(\Sigma _ {B, m}^\ell)$ represent the between-class feature variance for the aggregated model and the local model, respectively.
>
> - The feature-parameter alignment score of the aggregated model is strictly lower than that of the optimal local model:
>   $$\mathcal{A}(\tilde{\mathbf{Z}}_m^\ell, \tilde{\mathbf{W}}^{\ell+1}) < \mathcal{A}(\mathbf{Z}_m^\ell, \mathbf{W}_m^{\ell+1})$$
>   where $\mathcal{A}(\tilde{\mathbf{Z}}_m^\ell, \tilde{\mathbf{W}}^{\ell+1})$ and $\mathcal{A}(\mathbf{Z}_m^\ell, \mathbf{W}_m^{\ell+1})$ represent the feature-parameter alignment for the aggregated and local models, respectively.
>
> - Let $E^\ell = \|\tilde{\mathbf{z}}_m^\ell - \mathbf{z}_m^\ell\|$ be the feature error at layer $\ell$ for client $m$. Assuming the spectral norm of aggregated weights is bounded by $\lambda$ ($\|\tilde{\mathbf{W}}^\ell\|_2 \le \lambda$) and the layer-wise heterogeneity noise is bounded by $\gamma$ ($\|\mathbf{\Delta}_m^\ell \mathbf{z}_m^{\ell-1}\| \le \gamma$), the feature error accumulates exponentially with depth $\ell$:
>   $$E^\ell \le \gamma \frac{\lambda^\ell - 1}{\lambda - 1} \quad (\text{for } \lambda > 1)$$
>
> [1] Yu, Fuxun, et al. "Fed2: Feature-aligned federated learning." Proceedings of the 27th ACM SIGKDD conference on knowledge discovery \& data mining. 2021.
>
> [2] Ma, Xiaosong, et al. "Layer-wised model aggregation for personalized federated learning." Proceedings of the IEEE/CVF conference on computer vision and pattern recognition. 2022.
>
> [3] Zhou, Tianfei, and Ender Konukoglu. "FedFA: Federated Feature Augmentation." The Eleventh International Conference on Learning Representations.
>
> [4] Adilova, Linara, et al. "Layer-wise linear mode connectivity." The Twelfth International Conference on Learning Representations.
>
> [5] Papyan, Vardan, X. Y. Han, and David L. Donoho. "Prevalence of neural collapse during the terminal phase of deep learning training." Proceedings of the National Academy of Sciences 117.40 (2020): 24652-24663.
>
> [6] Parker, Liam, et al. "Neural collapse in the intermediate hidden layers of classification neural networks." arXiv preprint arXiv:2308.02760 (2023).
>
> [7] Tan, Yue, et al. FedProto: Federated Prototype Learning across Heterogeneous Clients, AAAI Conference on Artificial Intelligence, 2022.
>
> [8] Huang, Wenke, et al. "Rethinking Federated Learning With Domain Shift: A Prototype View", Proceedings of the IEEE/CVF Conference on Computer Vision and Pattern Recognition (CVPR), 2023.

---

### Official Review · Reviewer_FUHN · 2025-11-05

**Soundness:** 3
**Presentation:** 3
**Contribution:** 3
**Rating:** 6
**Confidence:** 3

**Summary:**

This paper investigates the temporary performance drop seen in Federated Learning (FL) after models are aggregated, using a novel "layer-peeled" analysis framework to understand the root causes. The authors identify a phenomenon called Cumulative Feature Degradation (CFD), where aggregation progressively degrades feature quality and disrupts the alignment between features and parameters as network depth increases. This degradation, especially the mismatch between the final features and the classifier, is pinpointed as the main cause of the performance drop. Despite this downside, the study confirms that aggregation is vital for improving model generalization and preventing overfitting to local client data. The paper also uses this framework to explain why common FL solutions work, showing that methods like parameter personalization, pre-trained initialization, and classifier fine-tuning are effective because they successfully mitigate the CFD effect

**Strengths:**

1. The authors rigorously demonstrate that the negative impact of aggregation is not a uniform hit but a compounding problem that progressively accumulates with network depth.

2. The study offers a balanced perspective. It not only identifies the downsides of aggregation (CFD) but also validates its crucial upside, showing that aggregation is what enables the model to create more generalizable features and mitigate local overfitting.

3. The paper introduces a "layer-peeled" analysis framework  that moves beyond standard accuracy or loss metrics.

**Weaknesses:**

1. The experimental setup involves a very small number of clients (e.g., 4 clients for PACS, 6 for DomainNet). This is not representative of typical cross-device FL scenarios, which can involve hundreds, thousands, or even millions of clients. The dynamics of averaging four or six models may be very different from averaging thousands, and it remains an open question whether the severity and behavior of CFD would scale, diminish, or change entirely in a massively federated setting.

2. The paper's conclusions about "model aggregation" are almost exclusively based on analyzing the FedAvg algorithm, which uses simple parameter-wise averaging. While FedAvg is a foundational baseline, the paper does not investigate whether the Cumulative Feature Degradation (CFD) phenomenon persists in more advanced FL algorithms designed specifically to combat aggregation problems (like FedProx, SCAFFOLD, or FedDyn). It's possible that CFD is a specific artifact of the naive FedAvg approach rather than an unavoidable downside of all model aggregation in FL

3. All experiments are conducted on image classification datasets (Digit-Five, PACS, and DomainNet) using standard vision architectures (CNNs and ViT) . The findings, while significant for computer vision, cannot be assumed to generalize to other major applications of FL. It is unknown if CFD manifests similarly in fundamentally different tasks, such as Regression problem, classification on text datasets etc.

**Questions:**

1. Can the author suggest that this analysis still holds for strong FL algorithms like SCAFFOLD, FedDyn, pFedMe, etc.?

2. Does the authors have any theoretical justification to explain why the simple averaging of model parameters fundamentally leads to this progressive, layer-by-layer degradation in feature quality and alignment?

---

> ### Author Response · Authors · 2025-12-03
>
> **Response to Weakness 1:**
> Thanks for your comments. We acknowledge the importance of verifying our findings in larger-scale settings. To address this, we have extended our layer-peeled feature evaluation to include a larger number of clients (e.g., [6, 12, 30] clients) on DomainNet dataset. The results demonstrate that the observed CFD phenomenon persists even as the number of participating clients increases. Furthermore, we contend that the CFD phenomenon is unlikely to diminish in massive federated settings due to the linear nature of aggregation. Mathematically, the aggregation of a large number of clients can be decomposed into a hierarchical aggregation of smaller subsets. Since CFD is prevalent within these smaller subsets, the phenomenon is propagated to the global level rather than being averaged out. Therefore, both our empirical results and intuition suggest that CFD remains significant in large-scale scenarios.
> Below is the relative change in feature metrics for specific layers, with detailed results provided in Figure 13 of the main text.
> - Relative change of within-class feature variance at specific layer.
>
>     | Layer | 6 Clients | 12Clients | 30Clients |
>     |---|---|---|---|
>     | L1 | 0.32±0.23 | 0.55±0.52 | 0.74±0.57 |
>     | L7 | 0.72±0.50 | 0.54±0.44 | 1.76±1.52 |
>     | L15 | 1.63±1.06 | 1.37±1.28 | 4.48±4.81 |
>     | L27 | 2.29±1.98 | 2.63±1.95 | 5.20±4.63 |
>     | L50 | 5.26±3.73 | 4.58±3.55 | 8.46±5.86 |
>
> - Relative change of feature-parameter alignment at specific layer.
>     | Layer | 6 Clients | 12Clients | 30Clients |
>     |---|---|---|---|
>     | L1 | 1.49±1.14 | 1.76±1.38 | 2.48±1.68 |
>     | L7 | 1.86±1.60 | 2.31±1.53 | 2.64±1.67 |
>     | L15 | 2.48±2.36 | 3.44±2.11 | 2.32±1.52 |
>     | L27 | 2.26±2.02 | 2.18±1.98 | 1.83±1.05 |
>     | L50 | 6.04±3.77 | 4.86±3.19 | 6.69±6.53 |
>
> **Response to Weakness 2 and Question 1:**
> Thanks for your comments. In response, we have extended our analysis to include advanced FL algorithms, specifically FedProx, SCAFFOLD, FedDyn, and pFedMe. The experimental results demonstrate that the CFD phenomenon persists across all these methods. Although these algorithms introduce mechanisms (e.g., proximal terms or control variates) to mitigate client drift or heterogeneity in the parameter space, our analysis reveals that they do not fundamentally resolve the feature degradation issue arising from aggregation. This confirms that CFD is not specific to the naive FedAvg, but rather a pervasive challenge inherent to model aggregation in non-I.I.D. settings across various FL algorithms.
> Below is the relative change in feature metrics for specific layers.Detailed results are provided in Figure 10 of the main text.
>
> - Relative change of within-class feature variance at specific layer.
>
>     | Layer | FedAvg | FedProx | SCAFFOLD | FedDyn | pFedMe |
>     |---|---|---|---|---|---|
>     | L1 | 0.60±0.51 | 0.63±0.50 | 0.59±0.51 | 0.36±0.37 | 1.74±1.31 |
>     | L7 | 0.67±0.50 | 0.64±0.76 | 0.73±0.68 | 0.50±0.45 | 1.93±1.43 |
>     | L15 | 2.64±2.19 | 1.95±1.24 | 2.78±2.27 | 1.96±1.64 | 2.16±1.77 |
>     | L27 | 4.83±3.62 | 2.93±2.55 | 3.78±3.43 | 2.32±2.15 | 1.37±1.11 |
>     | L50 | 9.68±7.25 | 4.49±4.25 | 5.91±6.00 | 5.89±7.86 | 2.08±2.25 |
>
> - Relative change of feature-parameter alignment at specific layer.
>     | Layer | FedAvg | FedProx | SCAFFOLD | FedDyn | pFedMe |
>     |---|---|---|---|---|---|
>     | L1 | 3.12±2.14 | 1.17±0.93 | 1.91±2.23 | 1.43±0.88 | 3.56±2.93 |
>     | L7 | 3.76±2.51 | 2.04±1.17 | 3.80±1.92 | 2.12±1.59 | 3.42±2.02 |
>     | L15 | 2.48±2.05 | 1.36±1.54 | 3.85±2.82 | 2.22±1.33 | 4.19±2.43 |
>     | L27 | 2.93±2.00 | 2.24±1.55 | 2.11±1.38 | 2.59±1.85 | 4.30±2.30 |
>     | L50 | 6.85±3.65 | 6.28±3.77 | 8.47±9.21 | 13.94±12.07 | 4.00±2.44 |

---

> ### Author Response · Authors · 2025-12-03
>
> **Response to Weakness 3:** Thank you for the suggestion. To verify the cross-domain generalizability of our findings, we have conducted additional experiments on a text classification task using the AmazonReviews dataset with MLP. As shown below, the experimental results confirm that the CFD phenomenon also exists in the NLP domain, suggesting that it is not specific to vision tasks. Regarding the regression task, we focused on classification in this paper because our analysis relies on metrics like feature variance and class separability, which are explicitly defined for classification problems. Extending these layer-peeled metrics to regression requires a different set of analytical tools, which we plan to explore in future work.
>
> - Relative change of within-class feature variance at specific layer.
>
>     | Epoch | L1  | L2 | L3  | L4 |
>     |-------|---------------|----------------|----------------|----------------|
>     | E10   | 0.00±0.00     | 0.24±0.10      | 2.54±1.11      | 3.95±1.14      |
>     | E20   | 0.00±0.00     | 0.07±0.04      | 0.60±0.16      | 1.07±0.43      |
>     | E30   | 0.00±0.00     | 0.04±0.02      | 0.34±0.11      | 0.60±0.26      |
>     | E40   | 0.00±0.00     | 0.03±0.02      | 0.24±0.08      | 0.41±0.21      |
>     | E50   | 0.00±0.00     | 0.05±0.05      | 0.32±0.16      | 0.43±0.36      |
>
> **Response to Question 2:** Thank you for your comments. We have provided a theoretical justification in **Appendix G** to support our empirical findings. Based on the assumptions of deep linear networks, our theoretical analysis demonstrates that model aggregation can lead to the following negative effects:
>
> - Model aggregation increases the within-class variance ($\Sigma_W$) and decreases the between-class variance ($\Sigma_B$) of features on local data, as shown by:
>   $$\mathrm{Tr}(\tilde{\Sigma} _ W^\ell) > \mathrm{Tr}(\Sigma _ {W, m}^\ell) \quad \text{and} \quad \mathrm{Tr}(\tilde{\Sigma} _ B^\ell) < \mathrm{Tr}(\Sigma _ {B, m}^\ell)$$
>   where $\mathrm{Tr}(\tilde{\Sigma} _ W^\ell)$ and $\mathrm{Tr}(\Sigma _ {W, m}^\ell)$ represent the within-class feature variance for the aggregated model and the local model, $\mathrm{Tr}(\tilde{\Sigma} _ B^\ell)$ and $\mathrm{Tr}(\Sigma _ {B, m}^\ell)$ represent the between-class feature variance for the aggregated model and the local model, respectively.
>
> - The feature-parameter alignment score of the aggregated model is strictly lower than that of the optimal local model:
>   $$\mathcal{A}(\tilde{\mathbf{Z}}_m^\ell, \tilde{\mathbf{W}}^{\ell+1}) < \mathcal{A}(\mathbf{Z}_m^\ell, \mathbf{W}_m^{\ell+1})$$
>   where $\mathcal{A}(\tilde{\mathbf{Z}}_m^\ell, \tilde{\mathbf{W}}^{\ell+1})$ and $\mathcal{A}(\mathbf{Z}_m^\ell, \mathbf{W}_m^{\ell+1})$ represent the feature-parameter alignment for the aggregated and local models, respectively.
>
> - Let $E^\ell = \|\tilde{\mathbf{z}}_m^\ell - \mathbf{z}_m^\ell\|$ be the feature error at layer $\ell$ for client $m$. Assuming the spectral norm of aggregated weights is bounded by $\lambda$ ($\|\tilde{\mathbf{W}}^\ell\|_2 \le \lambda$) and the layer-wise heterogeneity noise is bounded by $\gamma$ ($\|\mathbf{\Delta}_m^\ell \mathbf{z}_m^{\ell-1}\| \le \gamma$), the feature error accumulates exponentially with depth $\ell$:
>   $$E^\ell \le \gamma \frac{\lambda^\ell - 1}{\lambda - 1} \quad (\text{for } \lambda > 1)$$
>
> These theoretical results confirm the feature degradation (including feature variance and feature-parameter alignment) and the CFD phenomenon observed in our paper.

---

### Author Response · Authors · 2025-12-03

Dear reviewers and chair,

Thank you for your insightful comments on our paper. We have made revisions based on your suggestions, which are highlighted in red in the paper. The main changes are as follows:

- We have added a **theoretical justification** for our experimental findings, providing theoretical backing for our empirical observations.

- We have validated the phenomenon discovered in our experiments in **more scenarios**, including more modalities, additional federated learning algorithms, and varying numbers of clients.

- We have added an ablation study to validate the **accumulation of feature degradation**, further confirming that the degradation of high-level representations results from the propagation of degradation in low-level representations.

- We have included a **correlation analysis** between accuracy degradation and feature degradation, further clarifying their relationship.

Once again, we thank you for your valuable feedback, which has greatly helped improve our paper.

---

### Meta-Review · Area_Chair_mzvQ · 2026-01-06

**Summary:**

The paper presents a systematic study of model aggregation in federated learning from a layer‑peeled feature‑extraction perspective. Multiple reviewers appreciate that the proposed framework provides a systematic set of metrics for analyzing the dynamics of features and model parameters, moving beyond standard accuracy or loss metrics. Some concerns have been well addressed by the authors; however, the responses to several questions surrounding the key proposed concept of Cumulative Feature Degradation could be further strengthened.

**Reviewer Concerns:**

Some concerns raised by the reviewers have been well addressed. For example, reviewers noted that only a very small number of clients were used (e.g., 4 clients for PACS and 6 for DomainNet), the conclusion about “model aggregation” were drawn almost exclusively from FedAvg, and  all experiments were conducted on image classification datasets, etc. In response, the authors added new results using a larger number of clients (e.g., 6, 12, and 30 clients) on the DomainNet dataset, extended the analysis to additional FL algorithms including FedProx, SCAFFOLD, FedDyn, and pFedMe, and incorporated a text classification task. Two reviewers who gave “Reject” and “Marginally Below” ratings were concerned about the role and claims surrounding the proposed concept of Cumulative Feature Degradation (CFD); for instance, one questioned whether “CFD seems like a correlate, not a fundamental driver.” The AC appreciates the authors’ substantial effort to address these concerns. However, the AC still finds that the paper could be significantly strengthened by integrating these newly added results and analyses into a future submission to more convincingly position CFD as a guiding principle for analyzing or designing FL methods.

**Reviewer Scores:**

This paper receives the following ratings: Marginally Above, Reject, Marginally Below, and Marginally Above. If the reviewers had been able to participate fully in the discussion, the AC would expect some negative ratings to still remain. The AC recommends not accepting the paper in its present form.

---

### Decision · Program_Chairs · 2026-01-26

Reject